# Intellectual disability-associated factor Zbtb11 cooperates with NRF-2/GABP to control mitochondrial function

Brooke C. Wilson [1,3], Lena Boehme [1,3], Ambra Annibali[1,3], Alan Hodgkinson [1], Thomas S. Carroll[2], Rebecca J. Oakey [1] & Vlad C. Seitan [1✉]

Zbtb11 is a conserved transcription factor mutated in families with hereditary intellectual disability. Its precise molecular and cellular functions are currently unknown, precluding our understanding of the aetiology of this disease. Using a combination of functional genomics, genetic and biochemical approaches, here we show that Zbtb11 plays essential roles in maintaining the homeostasis of mitochondrial function. Mechanistically, we find Zbtb11 facilitates the recruitment of nuclear respiratory factor 2 (NRF-2) to its target promoters, activating a subset of nuclear genes with roles in the biogenesis of respiratory complex I and the mitoribosome. Genetic inactivation of *Zbtb11* resulted in a severe complex I assembly defect, impaired mitochondrial respiration, mitochondrial depolarisation, and ultimately proliferation arrest and cell death. Experimental modelling of the pathogenic human mutations showed these have a destabilising effect on the protein, resulting in reduced Zbtb11 dosage, downregulation of its target genes, and impaired complex I biogenesis. Our study establishes Zbtb11 as an essential mitochondrial regulator, improves our understanding of the transcriptional mechanisms of nuclear control over mitochondria, and may help to understand the aetiology of Zbtb11-associated intellectual disability.

[1] Department of Medical and Molecular Genetics, King's College London, London SE1 9RT, UK. [2] Bioinformatics Resource Centre, The Rockefeller University, New York, NY 10065, USA. [3]These authors contributed equally: Brooke C. Wilson, Lena Boehme, Ambra Annibali. ✉email: vlad.seitan@kcl.ac.uk

The biochemical energy contained within nutrients is captured through the process of cellular respiration, which culminates in the production of ATP through oxidative phosphorylation (OXPHOS) in mitochondria[1]. OXPHOS is fuelled by a chain of redox reactions in which electrons are transferred along an electron transport chain (ETC) that comprises four complexes (I–IV) localised to the mitochondrial inner membrane[2]. Three of the ETC complexes (I, III and IV) couple substrate oxidation with proton extrusion from the mitochondrial matrix into the intermembrane space, thus creating an electrochemical gradient across the mitochondrial inner membrane, which forces protons back into the matrix mainly through complex V (ATP synthase), driving the synthesis of ATP[3]. Tissues with high-energy demands, such as brain and muscle, are particularly reliant on the activity of the ETC, and these are usually disproportionally affected in mitochondrial diseases. Nevertheless, because the activity of the ETC is also vital for regenerating the cellular pool of redox co-factors that catalyse a number of other metabolic reactions[4,5], a functioning ETC is essential for all proliferating cells, irrespective of their ATP consumption[6].

Mitochondria possess their own genome (mtDNA) and transcription machinery, as well as a specialised organellar ribosome (the mitoribosome). While the mtDNA encodes a small number of core subunits of the OXPHOS complexes, the majority of the mitochondrial proteome is encoded in the nuclear genome (nDNA). Mitochondrial biogenesis and function are therefore controlled to a significant extent through transcriptional regulation of nuclear genes. A number or transcription factors have been implicated in the regulation of nuclear-encoded mitochondrial genes, some of which are essential while others have modulatory roles important mainly in specific cell types or in conditions of increased energy demand[7,8]. The nuclear respiratory factors (NRFs) NRF-1 and NRF-2 (also known as GABP (GA-binding protein)) are essential transcriptional activators, and the most prominent regulators of nuclear-encoded mitochondrial genes, driving the expression of the majority of nuclear-encoded OXPHOS subunits and mitoribosomal proteins, as well as mtDNA replication and transcription factors[9]. NRF-1 and NRF-2/GABP play overlapping but non-redundant roles in activating these genes, and they are found associated with different DNA sequence motifs. It is not currently completely understood how the activity of NRFs is regulated, how their recruitment to target genes is controlled or whether they engage all their genomic targets in the same way or through locus-specific mechanisms.

Zinc-finger and BTB domain-containing (ZBTB) proteins are a family of structurally related transcription factors, which function by binding DNA through C-terminal zinc-finger motifs and recruiting co-factors via protein–protein interactions mediated by an N-terminal BTB domain[10]. Zbtb11 is a poorly characterised ubiquitously expressed ZBTB protein that is conserved in vertebrates. Recent genetic studies of consanguineous families with recessive intellectual disability (ID) identified homozygous missense mutations in human ZBTB11 as causal variants[11,12]. Two pathogenic mutations were identified, both of which are predicted to disrupt individual Zbtb11 zinc-finger motifs. Affected patients display morphological defects in the brain, including ventriculomegaly and cerebellar atrophy, and also show neuromuscular defects, such as ataxia and facial hypotonia[12]. The molecular and cellular functions of Zbtb11 have hitherto remained unknown, so the aetiology of this disease is not currently understood. Zbtb11 is listed as a housekeeping protein by the Human Protein Atlas[13] (https://www.proteinatlas.org/humanproteome/tissue/housekeeping), and two separate genome-wide genetic screens (gene trap and CRISPR) recently conducted in several different human cell lines (KBM7, HAP1 and K562)[14,15] have identified ZBTB11 as an essential gene, indicating that this factor plays fundamental cellular roles. Consistent with this notion, all ZBTB11 mutations reported so far have been missense mutations that are not predicted to cause complete loss of function, suggesting that the disease may be the result of a partial impairment of otherwise essential functions carried out by Zbtb11. However, what these functions might be has remained unknown.

Here, we investigate the cellular and molecular functions of Zbtb11 and find that it is a key regulator of mitochondrial function. We show that Zbtb11 controls the locus-specific recruitment of NRF-2/GABP, but not NRF-1, to activate a subset of nuclear-encoded genes with roles in the biogenesis of OXPHOS complex I and the mitoribosome. Genetic inactivation of Zbtb11 leads to a severe complex I assembly defect and loss of complex I activity, reduced respiration, mitochondrial depolarisation, and consequently to proliferation arrest and cell death. Zbtb11 therefore cooperates with NRF-2/GABP to maintain the homeostasis of mitochondrial function. We provide evidence that mutations associated with hereditary ID disrupt this function by destabilising the Zbtb11 protein, leading to its reduced dosage, downregulation of its target genes and impaired complex I biogenesis. Our study establishes Zbtb11 as an essential regulator of mitochondrial function, reveals a previously unanticipated mechanism of locus-specific regulation of NRF-2/GABP activity, and indicates that ID associated with mutations in ZBTB11 may be at least in part the manifestation of a mitochondrial disease.

## Results

**Zbtb11 targets nuclear-encoded mitochondrial genes**. We initially set out to determine the potential functions of Zbtb11 by identifying the exact locations where it binds in the genome, and to this end, we carried out chromatin immunoprecipitation coupled with high-throughput sequencing (ChIP-seq) experiments. Due to the lack of existing validated anti-Zbtb11 antibodies, we used CRISPR/Cas9 to 'knock-in' a 3×FLAG tag into the Zbtb11 locus of the E14 mouse embryonic stem cell line (ESC) line[16] (Fig. 1a), and isolated a homozygous line in which all endogenous Zbtb11 protein is N-terminally tagged with 3×FLAG (FLAG-Zbtb11) (Fig. 1b and Supplementary Fig. 1a). We then performed anti-FLAG ChIP-seq in Zbtb11[FLAG/FLAG] E14 cells and control E14 wild-type (WT) cells, identifying 9350 FLAG-Zbtb11 consensus peaks from three biological replicates (irreproducibility discovery rate (IDR) <0.02, see Methods), from which we removed 219 peaks that that were also identified by FLAG ChIP-seq in control E14 WT cells.

We used this reliable set of FLAG-Zbtb11 binding sites to validate an anti-Zbtb11 antibody, which, although not suitable for immunoblotting, reproducibly showed strong ChIP enrichment at the same sites as FLAG-Zbtb11 (Supplementary Fig. 1b). ChIP-seq with this Zbtb11 antibody in E14 WT cells detected 8957 Zbtb11 peaks (IDR < 0.02), of which 7500 overlapped with the FLAG-Zbtb11 peaks in Zbtb11[FLAG/FLAG] E14 cells (Fig. 1c). The ChIP-seq signal was very well correlated between Zbtb11 and FLAG-Zbtb11 datasets (Fig. 1d and Supplementary Fig. 1c–e). We therefore generated a high-confidence Zbtb11 peak set that comprises the 7500 peaks common to both Zbtb11 and FLAG-Zbtb11, which we subsequently used in the rest of the analyses in this study.

We found that Zbtb11 binds predominantly (56%) at gene promoters, the vast majority (85.6 %) of which are CpG island promoters (Fig. 1e) — a promoter type mainly associated with ubiquitously expressed housekeeping genes[17,18]. Analysing the distribution of Zbtb11 peak signal values revealed that the top 858 peaks are significantly stronger than the rest of the peaks in the dataset, to the extent that they are outliers in the distribution

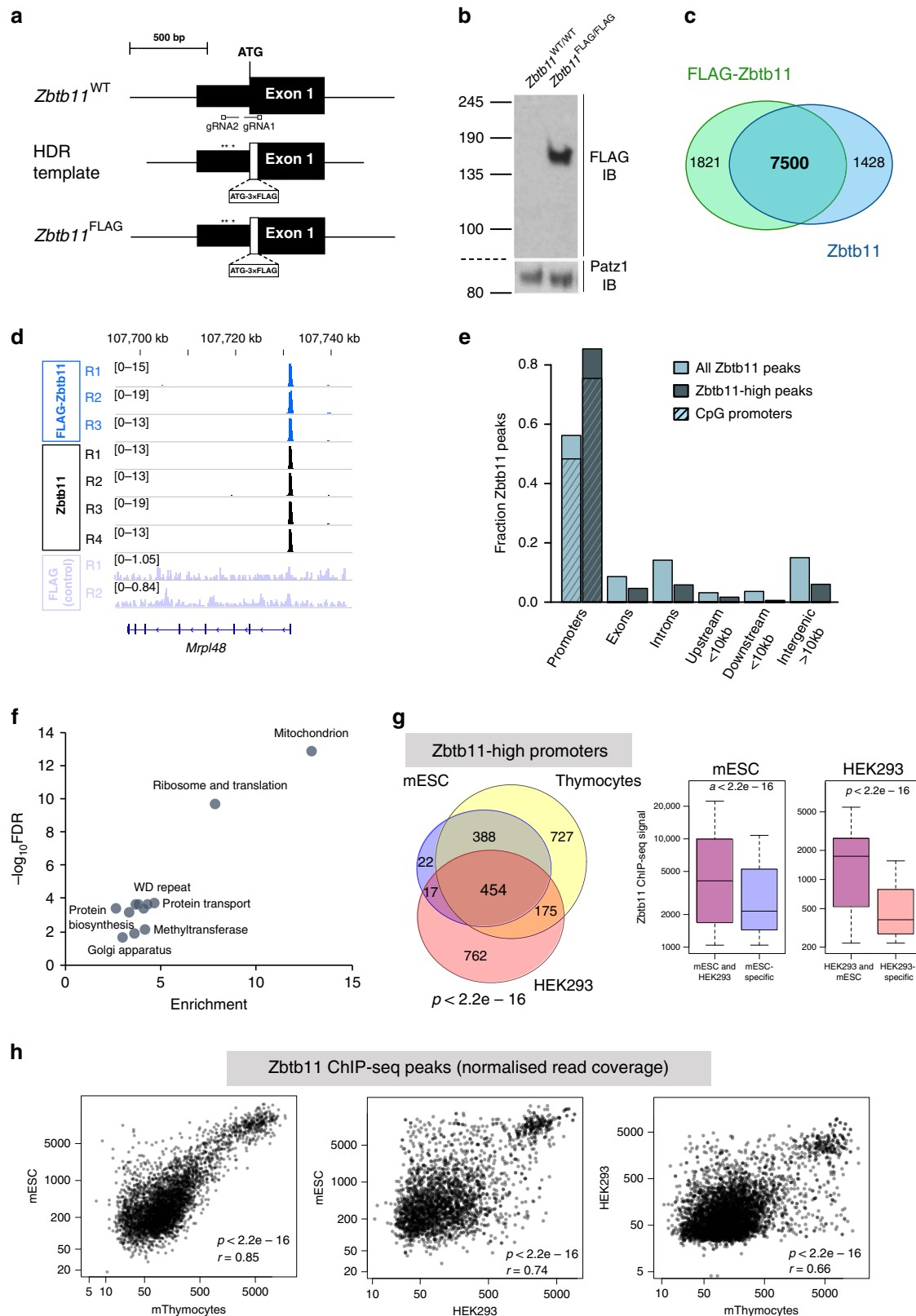

(values are >1.5 times the interquartile range over the third quartile) (Supplementary Fig. 1f). These Zbtb11-high peaks represent the top 10% of the peak set, and their signal is overall 8.7 times stronger than the rest of the peaks. Zbtb11-high peaks show increased association with promoters compared to the entire Zbtb11 peak dataset (86.2% vs. 56%) (Fig. 1e).

To identify pathways and cellular processes potentially controlled by Zbtb11, we carried out functional enrichment analyses using genes with the most prominent Zbtb11 binding sites (the 858 Zbtb11-high peaks). The most highly enriched annotation term cluster was related to mitochondria (13-fold enrichment), 15% of tested genes being included in this category

**Fig. 1 Zbtb11 is enriched at promoters of nuclear-encoded mitochondrial genes. a** CRISPR/Cas9 'knock-in' approach used to generate the Zbtb11[FLAG] allele. Two gRNAs targeting the 5′ end of the coding region in the Zbtb11 gene were used in combination with the Cas9 nickase, and a donor template for homology-directed repair (HDR). The donor template contained the 3×FLAG tag immediately downstream of the translation initiating methionine and was protected from Cas9 activity through the substitution of poorly conserved non-coding nucleotides at the gRNA2 site (*), and because the gRNA1 site is disrupted by the 3×FLAG insertion. **b** Anti-FLAG immunoblot of whole-cell lysates from a Zbtb11[FLAG/FLAG] homozygous cell line, and the parental line E14 (Zbtb11[WT/WT]). Patz1 was used as a loading control. Molecular weight marker unit is kDa. Source data are provided as a Source Data file. **c** Overlap between ChIP-seq peaks for FLAG ChIP in Zbtb11[FLAG/FLAG] E14 cells, and for ChIP with anti-Zbtb11 antibodies in the parental line E14 (Supplementary Fig. 1c, d). **d** Example of read coverage at a Zbtb11 target locus (Mrpl48). FLAG ChIP in E14 WT (FLAG control) is shown as background control (Supplementary Fig. 1c, d). **e** Genome-wide distribution of Zbtb11 peaks with respect to gene features. The distribution of the entire Zbtb11 ChIP-seq dataset is shown alongside that of Zbtb11-high peaks. Within the fraction of peaks found at the promoters, the hatched area indicates the proportion found at CpG island promoters. Source data are provided as a Source Data file. **f** DAVID functional annotation terms associated with genes that have Zbtb11-high peaks present at their promoters (FDR < 0.05). Plot shows enrichment values vs. significance ($-\log_{10}$ FDR) when compared to all genes in the mouse genome. Source data are provided as a Source Data file. **g** Venn diagram: conservation of Zbtb11-high peaks at promoters in mESCs, mouse thymocytes and HEK293. The p value for each pairwise hypergeometric test was $p < 2.2e-16$. Box plots: distribution of Zbtb11 ChIP-seq signal at conserved and cell-type/species-specific peaks, showing that Zbtb11 preferentially binds to conserved peaks. P values are for Wilcoxon's rank-sum tests. Box plots show the interquartile range (box outline) and median value (horizontal line), with the whiskers delineating the lower and upper limits of the data. Source data are provided as a Source Data file. **h** Conservation of Zbtb11 binding preference at target promoters, across tissues (left panel, mouse ESCs vs. thymocytes) and species (right panels, mESCs vs. human HEK293 cells, and mouse thymocytes vs. HEK293). Pearson's correlation coefficients (r) and p values (p) for two-sided correlation tests are shown. Source data are provided as a Source Data file.

(Fig. 1f). This was followed at some distance by ribosome and translation (7.8-fold enrichment); however, 52% of genes in this category function in mitochondrial translation and were also listed in the mitochondria annotation cluster. We did not detect any enrichment of terms related to development or ESC differentiation. Zbtb11 therefore seems to mainly associate with promoters of a subset of housekeeping genes, among which genes with mitochondrial functions are highly enriched.

To determine whether the genomic distribution of Zbtb11 is conserved between cell types, we performed ChIP-seq with Zbtb11 antibodies in a pure population of fluorescence-activated cell sorting (FACS)-sorted mouse thymocytes, as well as in the human cell line HEK293. The strength of Zbtb11 ChIP-seq peaks in mouse thymocytes and human HEK293 cells showed a very similar distribution to mESC, revealing a select group of Zbtb11-high peaks (Supplementary Fig. 1f). The binding of Zbtb11 to its target promoters was remarkably well conserved across cell types and species, showing a high degree of conservation not only in the location of binding sites but also in affinity. Eighty-seven percent of Zbtb11-high peaks identified in mESCs were conserved in thymocytes, and 53% were conserved in human cells (Fig. 1g). Zbtb11 showed a significant preference for conserved binding sites, ChIP-seq peaks at these sites being considerably stronger than the cell-type- and species-specific ones (Fig. 1g). Moreover, among conserved sites binding affinity was highly correlated between cell types (Pearson's $r = 0.85$), and between species ($r = 0.75$) (Fig. 1h). Functional enrichment analysis of Zbtb11 target genes in human cells revealed that just as in mice these are also highly enriched in genes with mitochondrial function (Supplementary Fig. 1g).

Altogether, the genomic distribution of Zbtb11 is consistent with a conserved essential role of this factor in regulating a subset of housekeeping genes, in particular genes with mitochondrial function.

**Zbtb11 is essential for proliferation and cell viability.** To establish an experimental system that would allow us to interrogate the functions of Zbtb11, we first attempted to generate constitutive Zbtb11 knockout (KO) ESCs by targeting exon 3 using CRISPR/Cas9-mediated genome editing. However, although we were able to efficiently isolate and expand heterozygous WT/KO clones, we were never able to isolate homozygous Zbtb11 KO lines, which suggested that just as in human cells[14,15] Zbtb11 is also essential for cell viability in mouse ESCs. We therefore took a different approach, generating an inducible

Zbtb11 KO ESC line. Using CRISPR/Cas9-mediated 'knock-in', we flanked exon 3 of Zbtb11 with loxP sites, and also inserted the ERt2-Cre transgene[19] in the Rosa26 (R26) locus[20] (Fig. 2a and Supplementary Fig. 2a). In these cells, ERt2-Cre can be induced by addition of 4-hydroxytamoxifen (4OHT) to delete Zbtb11 exon 3, which is predicted to induce a frame shift and truncate the Zbtb11 protein to an 185-amino-acid N-terminal fragment that lacks the BTB domain and all of the zinc-fingers motifs.

Treatment of Zbtb11[lox/lox] R26[ERt2-Cre] cells with 4OHT, but not with the carrier ethanol (EtOH), resulted in efficient deletion of Zbtb11 exon 3 and depletion of functional Zbtb11 mRNA transcripts, which was essentially complete 24 h after 4OHT treatment (Fig. 2b). ChIP-qPCR experiments with anti-Zbtb11 antibodies showed the signal was decreased 24 h after treatment and became virtually undetectable from 48 h post treatment onwards (Fig. 2b and Supplementary Fig. 2b). This further validated the specificity of the anti-Zbtb11 ChIP antibody, and showed that Zbtb11 is completely depleted at the protein level 48 h after Zbtb11 KO is induced. Acute depletion of Zbtb11 transcripts and protein is maintained at 72 and 96 h post KO induction (Fig. 2b), indicating that there is no selection for cells that may escape deletion and have a growth advantage. These results establish our inducible KO cell line as a robust experimental system that allows rapid and acute depletion of Zbtb11 to study its cellular functions.

Zbtb11[lox/lox] R26[ERt2-Cre] cells treated with EtOH continued to grow exponentially as long as they were regularly passaged. By contrast, the proliferation of 4OHT-treated cells slowed down after Zbtb11 was completely depleted from the chromatin, being completely arrested 96 h after treatment (Fig. 2b). Cell cycle analyses by flow cytometry revealed that Zbtb11 depletion resulted in a reduction of cells in S phase and a relative increase of cells in G1, while the proportion of cells in G2/M did not appear to change (Fig. 2c). This indicates that the slowdown in proliferation caused by Zbtb11 depletion is mainly underpinned by cell cycle arrest in G1. Concomitant with proliferation arrest, cell death was also noticeable in 4OHT-treated cultures starting at day 4 (96 h), becoming extensive a day later (120 h) so that very few cells were left in culture beyond 5 days post treatment (Fig. 2d). Flow cytometry measurements of cleaved caspase-3 failed to detect any increase in apoptosis rates (Supplementary Fig. 2c). However, the increase in cell death was measurable by Annexin V-mediated detection of phosphatidylserine externalisation (Fig. 2e and Supplementary Fig. 2d), which was significant from 96 h post-4OHT

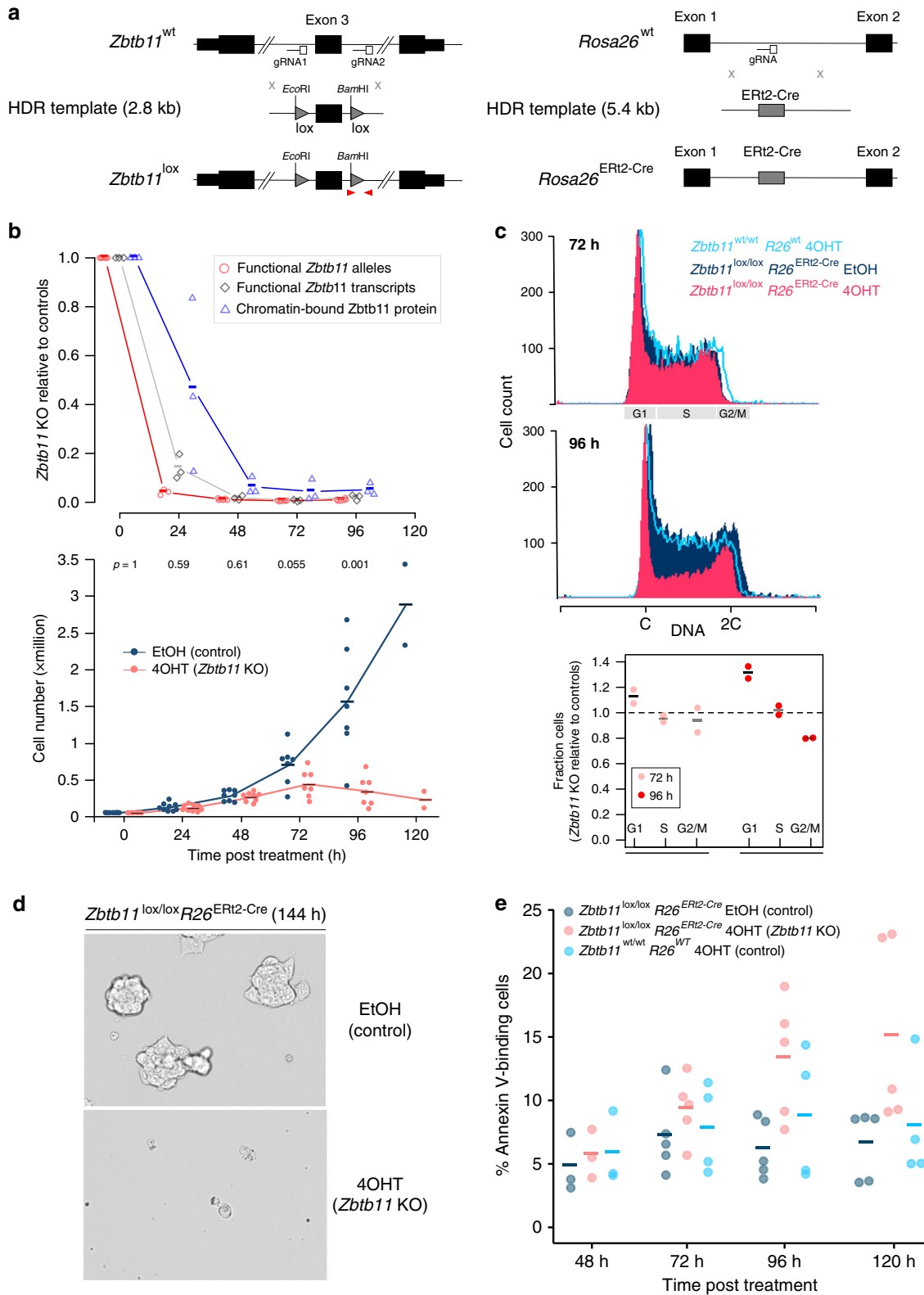

treatment onwards. No significant changes in proliferation or cell death were discernible in 4OHT-treated $Zbtb11^{wt/wt}$ $R26^{wt}$ cells (Fig. 2c, e), confirming that the observed effects were not artefacts of 4OHT treatment. These results establish that Zbtb11 is essential for proliferation and cell viability, and when its functions are blocked, cells undergo cell cycle arrest in G1 and subsequently suffer caspase-independent cell death.

**Zbtb11 functions as a transcriptional activator**. To determine what processes are disrupted by the depletion of Zbtb11, we carried out whole-transcriptome analyses by directional RNA-sequencing (RNA-seq). We compared $Zbtb11^{lox/lox}$ $R26^{ERt2-Cre}$ cells treated with either 4OHT (*Zbtb11* KO) or EtOH (control). To control for non-specific effects of 4OHT, we also analysed $Zbtb11^{wt/wt}$ $R26^{wt}$ cells treated with 4OHT. We analysed cells

**Fig. 2 Zbtb11 is essential for proliferation and cell viability. a** CRISPR 'knock-in' strategies used to generate the conditional *Zbtb11* KO allele (*Zbtb11*$^{lox}$) (left), and to insert the ERt2-Cre transgene in the *Rosa26* locus (right). The loxP sites (lox) flanking *Zbtb11* exon 3 include ectopic restriction sites (Supplementary Fig. 2a). Red arrowheads mark qPCR primers used to measure the deletion efficiency once ERt2-Cre is activated—recombination between the two loxP sites leads to excision of the left primer binding site along with exon 3, so PCR products are no longer generated. **b** Time-course experiment describing the dynamics of Zbtb11 inactivation in our inducible *Zbtb11* KO ESC line (*Zbtb11*$^{lox/lox}$ *Rosa26*$^{ERt2-Cre}$). Upper panel—decay of functional *Zbtb11* alleles (red circles, $n = 6$ independent experiments, except 24 h timepoint for which $n = 3$), *Zbtb11* transcripts (grey diamonds, $n = 3$) and protein (blue triangles, $n = 3$) in 4OHT-treated cells relative to their EtOH-treated control cells following KO induction. Lower panel—growth curves, with $p$ values (two-tailed paired $t$ test) above each timepoint. Source data are provided as a Source Data file. **c** Propidium iodide stain for cell cycle analysis of control (*Zbtb11*$^{lox/lox}$ EtOH) and *Zbtb11* KO (*Zbtb11*$^{lox/lox}$ 4OHT) cells. 4OHT-treated *Zbtb11*$^{wt/wt}$ cells were included to control for potential non-specific effects of 4OHT. Top two panels—representative flow cytometry histograms showing the distribution of DNA content in the sampled cell populations. G1, S or G2/M phases of the cell cycle are indicated on the horizontal axis. Bottom panel—mean and SD of the fractions of 4OHT-treated cells present in each cell cycle phase (normalised to EtOH-treated controls, $n = 2$). Source data are provided as a Source Data file. **d** Bright field microscopy images of 144 h *Zbtb11* KO and control cells, illustrating the collapse of the culture 6 days after inducing *Zbtb11* KO. **e** Quantification of cell death after *Zbtb11* KO induction with FITC-coupled Annexin V. The fraction of Annexin V-positive cells is shown for $n = 3$ (48 h), $n = 5$ (72, 96 and 120 h) independent experiments (Supplementary Figs. 2d, 9). Source data are provided as a Source Data file.

treated for 48 h because at this time point Zbtb11 protein is completely depleted in 4OHT-treated cells (Fig. 2b and Supplementary Fig.2b), but there are no detectable changes in proliferation and cell survival (Fig. 2c, e). We therefore expected to detect mainly transcriptional changes at genes directly regulated by Zbtb11.

When comparing 4OHT- and EtOH-treated *Zbtb11*$^{lox/lox}$ *R26*$^{ERt2-Cre}$ cells, we found 154 differentially expressed (DE) genes (false discovery rate (FDR) < 0.05), 145 of which were downregulated (Fig. 3a and Supplementary Data), which indicates that Zbtb11 functions as a transcriptional activator. None of these genes were DE when comparing 4OHT-treated *Zbtb11*$^{wt/wt}$ *R26*$^{wt}$ cells with EtOH-treated cells, indicating that their deregulation was not a result of the 4OHT treatment. To verify our RNA-seq results through a different method, we performed quantitative reverse transcription-PCR (qRT-PCR) for a panel of 20 genes spanning the entire range of effect sizes, and the results of the two methods were strongly correlated (Supplementary Fig. 3a).

To verify that the transcriptional changes are specific to Zbtb11 depletion, we performed experiments in which we rescued the expression of Zbtb11 in KO cells by ectopically expressing its cDNA. We transfected *Zbtb11*$^{lox/lox}$ *R26*$^{ERt2-Cre}$ cells with a plasmid expressing either FLAG-Zbtb11-IRES-GFP, or GFP alone as control, and treated them for 48 h with either EtOH or 4OHT. We subsequently FACS-sorted GFP-positive cells to isolate successfully transfected cells, confirmed specific expression of the FLAG-Zbtb11 protein by immunoblotting (Fig. 3b), and subsequently performed qRT-PCR to measure the expression of a panel of genes found to be downregulated in the RNA-seq dataset. As seen in Fig. 3b, the expression of these genes was specifically rescued in the 4OHT-treated cells transfected with FLAG-Zbtb11 but not in 4OHT-treated cells transfected with GFP only. These results confirm the specificity of our inducible *Zbtb11* KO ESC line and show that the transcriptional changes detected by RNA-seq are indeed the result of Zbtb11 depletion.

Integration of the RNA-seq and ChIP-seq datasets revealed that 147 of the 154 DE genes (95%) have Zbtb11 peaks at their promoters, and that these binding sites are the strongest in the dataset (Supplementary Fig. 3b). Moreover, Zbtb11 binding at the promoters of DE genes is remarkably well conserved across cell types and mammalian species, these binding sites being also the strongest in mouse thymocytes and human HEK293 cells (Fig. 3c). These results firmly indicate that Zbtb11 is directly regulating these genes, and that strength of binding is a good indicator of whether a Zbtb11 binding event is critical for transcriptional regulation.

In agreement with the results of the ChIP-seq analyses showing that Zbtb11 targets housekeeping genes, the Zbtb11-dependent genes were constitutively expressed at levels that were very well correlated between mouse ESCs, thymocytes and HEK293 cells (Supplementary Fig. 3c). To determine which processes are controlled by Zbtb11, we performed functional enrichment analyses using the list of DE genes. This identified only one significant cluster of annotation terms (FDR < 0.05), which is defined by mitochondria and had an enrichment score of 6.12 (Fig. 3d and Supplementary Table 1). To investigate more thoroughly the association between Zbtb11 and genes with mitochondrial functions, we cross-referenced the DE genes with the MitoMiner database[21] and found that 27% encode proteins with mitochondrial function, which represents an enrichment of 4.5 relative to the genes tested. All DE genes were nuclear-encoded, and they did not include any of the factors previously implicated in the regulation mitochondrial biogenesis and function, such as NRF-1, NRF-2/GABP or members of the PGC-1, PPAR or ERR families.

Finally, to verify that the changes observed at the mRNA level are reflected at the protein level, we carried out immunoblots on whole-cell lysates from *Zbtb11*$^{lox/lox}$ *R26*$^{ERt2-Cre}$ and *Zbtb11*$^{wt/wt}$ *R26*$^{wt}$ cells treated with either EtOH or 4OHT for 48 and 72 h. As seen in Fig. 3e, the mitochondrial proteins Mrpl48 and Ndufc2, encoded by DE genes, were already strongly downregulated 48 h post treatment. These changes were not detected in 4OHT-treated *Zbtb11*$^{wt/wt}$ *R26*$^{wt}$ cells, indicating they are not caused by 4OHT treatment alone. Altogether, these results show that Zbtb11 has fundamental housekeeping roles in directly promoting the expression of a subset of nuclear-encoded mitochondrial proteins.

**Zbtb11 controls recruitment of NRF-2 to its target promoters.** To understand how Zbtb11 controls its target genes, we sought to identify other factors that Zbtb11 might functionally interact with. To this end, we performed de novo motif discovery using the Zbtb11 ChIP-seq peak set, and separately the promoter sequences of the 154 Zbtb11-dependent genes. Both mouse and human ChIP-seq datasets identified as the top hit an identical motif, which closely matched that recognised by the ETS-domain protein GABPa (Fig. 4a and Supplementary Fig. 4a). In addition, the most highly enriched motif at the promoters of the Zbtb11-dependent genes was the same GABPa motif, in both the mouse and the human genome, and this was firmly centred around the transcription start sites (TSS) (Fig. 4b). The consensus motif for NRF-1 binding was also found to be enriched at the promoters of DE genes, but to a lesser extent and without a clear preference for the TSS (Fig. 4b). GABPa is the DNA binding subunit of the multimeric transcription factor GABP also known as NRF-2, which alongside NRF-1 plays essential roles in regulating mitochondrial biogenesis and function. Our motif analyses therefore

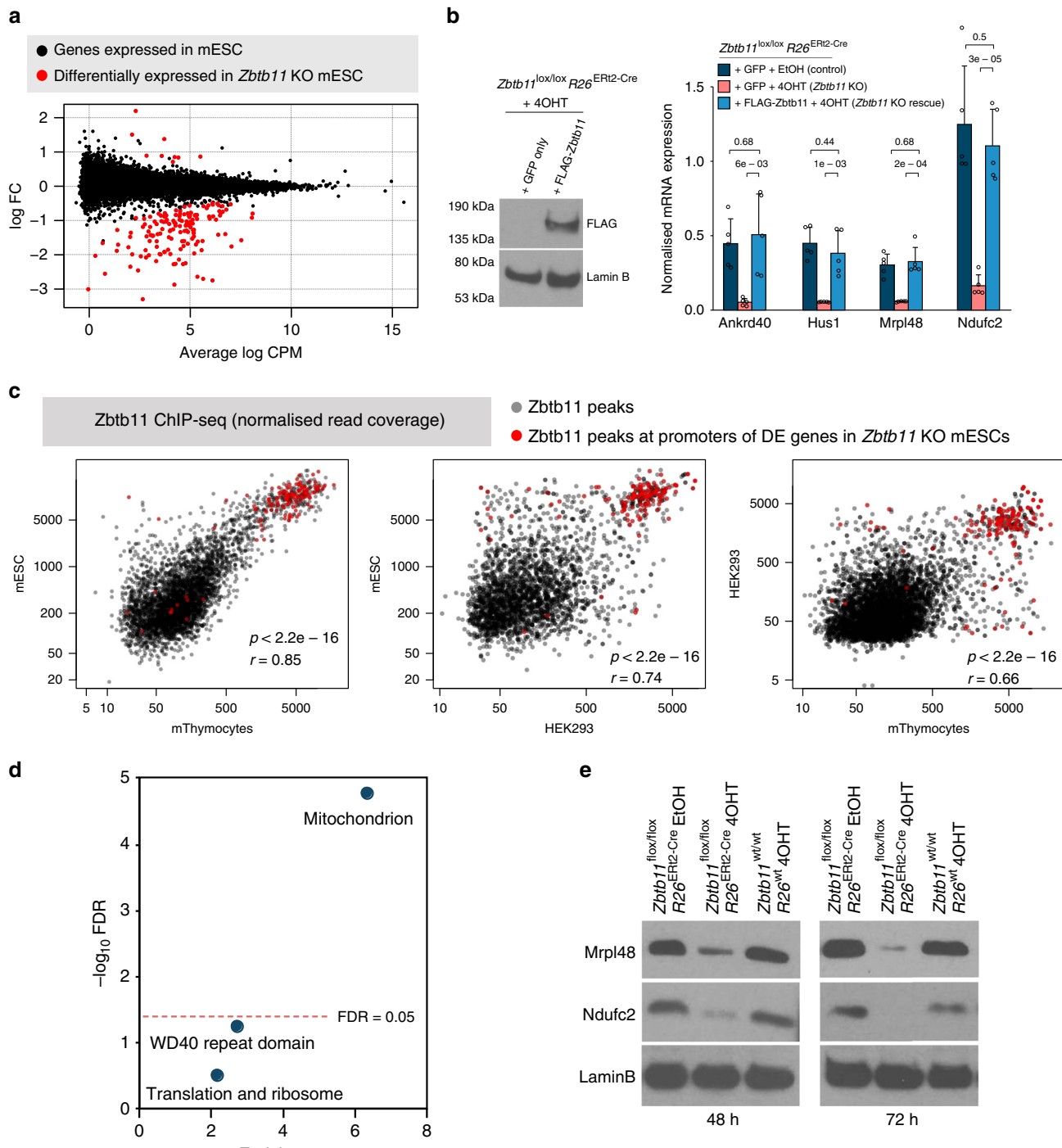

**Fig. 3 Zbtb11 supports the expression of a subset of nuclear-encoded mitochondrial proteins. a** Whole-transcriptome analysis by RNA-seq comparing *Zbtb11*^lox/lox^ *Rosa26*^ERt2-Cre^ cells treated with either EtOH (control) or 4OHT (*Zbtb11* KO) for 48 h. Plot shows log$_2$-transformed fold-change (log FC) against transcript abundance (mean counts per million, log CPM) for all genes expressed in ESCs. Differentially expressed (DE) genes (FDR < 0.05) are shown in red. **b** Zbtb11 expression rescue experiment. *Zbtb11*^lox/lox^ *Rosa26*^ERt2-Cre^ cells were transfected with a plasmid expressing either FLAG-Zbtb11-IRES-GFP or GFP alone as control, and treated with either EtOH (Control) or 4OHT (*Zbtb11* KO). GFP-positive cells were FACS-sorted 48 h post treatment. Left panel—immunoblot of whole-cell lysates showing the expression of Zbtb11 protein was specifically restored. Right panel—qRT-PCR measuring the expression of a panel of genes downregulated in *Zbtb11* KO cells. Barplot shows mean and standard deviation of *n* = 5 independent experiments. Source data are provided as a Source Data file. **c** Conservation of Zbtb11 binding at promoters of Zbtb11-dependent genes. Plots show strength of Zbtb11 binding at conserved target promoters in mESC, thymocytes and HEK293 cells (Fig. 1f). Binding sites at promoters of genes DE in *Zbtb11* KO cells are marked in red—note these are the strongest sites in the dataset and very well conserved across cell types and species. *r* = Pearson's correlation coefficient; *p* = *p* value for two-sided correlation test based on Pearson's product–moment correlation coefficient. **d** DAVID functional annotation analysis of the genes differentially expressed in *Zbtb11* KO indicates one significant cluster of annotation terms (FDR < 0.05), which is related to mitochondria. Significance (−log$_{10}$ FDR) is plotted against enrichment of the annotation terms associated with the genes. **e** Immunoblots for two mitochondrial proteins (Mrpl48 and Ndufc2) for which transcripts were found to be downregulated in *Zbtb11* KO cells. Lamin B was used as a loading control.

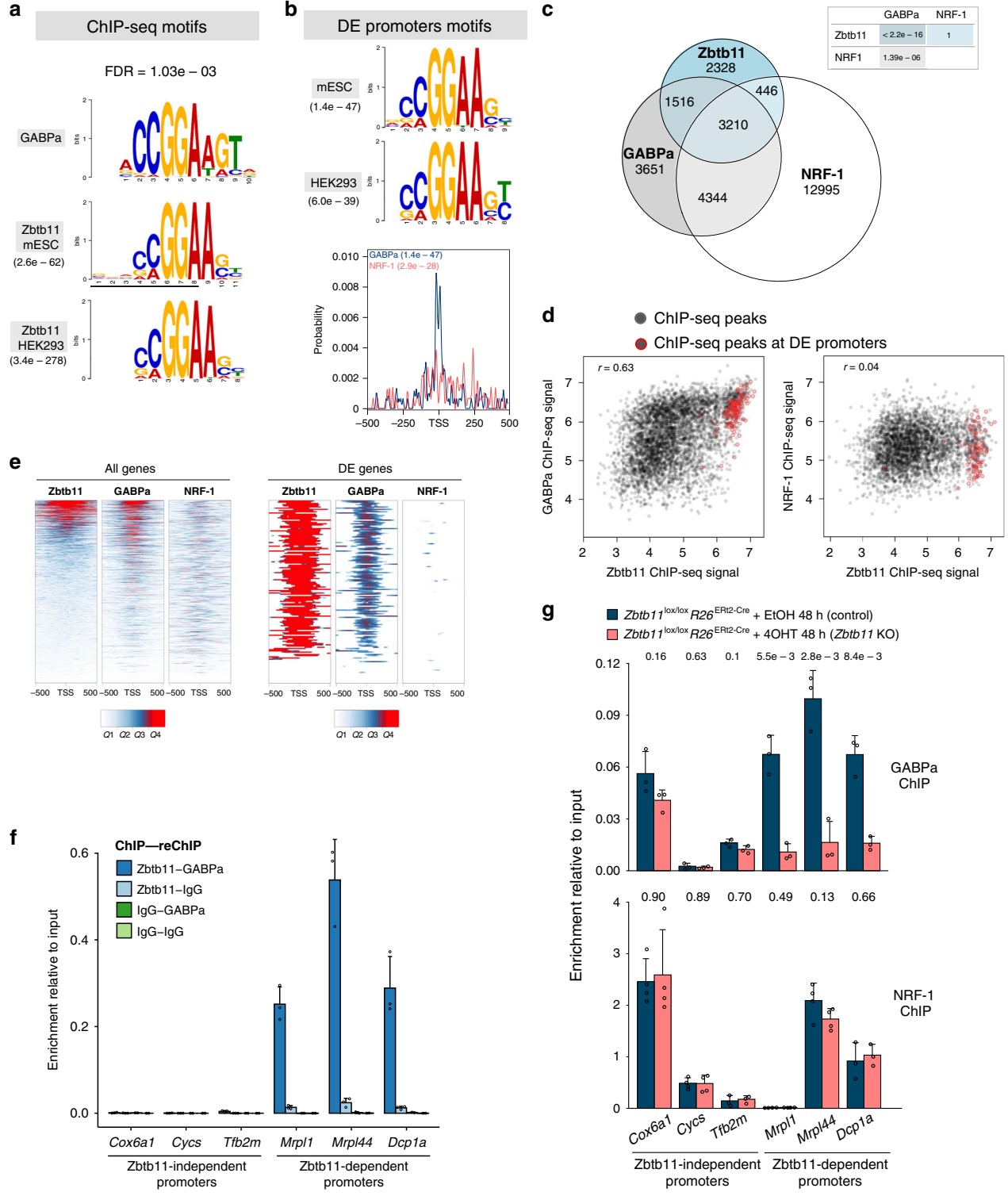

suggested that Zbtb11 may regulate its target genes and mitochondrial activity through functional interactions with NRF-1 and/or NRF-2/GABP.

To test whether these motif analyses were corroborated by experimental evidence, we integrated our Zbtb11 ChIP-seq data with previously published ChIP-seq datasets for GABPa[22] and NRF-1 (ref. [23]) in mouse ESCs. This revealed significant genome-wide overlap between Zbtb11 and GABPa peaks (p value < 2.2e −16), but not between Zbtb11 and NRF-1 peaks (Fig. 4c). Sixty-three percent of Zbtb11 peaks overlap with GABPa, but this

fraction increases to 97% among Zbtb11-high peaks. While Zbtb11 overlaps with NRF-1 mainly at sites that are also occupied by GABPa, there is extensive NRF-1-independent overlap between Zbtb11 and GABPa. In agreement with previous studies showing that NRF-1 and NRF-2 share some common target genes, we find significant genome-wide overlap between NRF-1 and GABPa peaks, including at many Zbtb11-independent sites (Fig. 4c). Illustrative of the different locus-specific associations between Zbtb11, NRF-2 and NRF-1, these factors co-occupy the promoters of some mitoribosome subunit genes, but Zbtb11 does

**Fig. 4 Zbtb11 controls the locus-specific recruitment of NRF-2. a** Zbtb11 top motifs identified de novo from Zbtb11-high peaks in mESCs and HEK293 cells, respectively (MEME-ChIP *E* values indicated), aligned to previously determined GABPa motif (upper panel). FDR for the Zbtb11 motif match to GABPa is indicated above. **b** Top motifs identified de novo from the promoter sequences of Zbtb11-dependent genes in mouse (mESC) and human (HEK293), respectively. Lower panel—the probability distribution for GABPa and NRF-1 motifs in the promoter regions of Zbtb11-dependent genes. Enrichment *p* values are given for each motif. **c** Genome-wide overlap between ChIP-seq peaks identified in mouse ES cell for Zbtb11, NRF-1 and GABPa. *P* values for pairwise hypergeometric tests (see Methods) are shown in the table. **d** Correlation of ChIP-seq signal (normalised read coverage) at overlapping peaks. Peaks at promoters of Zbtb11-dependent genes are circled in red. **e** Zbtb11, NRF-1 and GABPa ChIP-seq signal (quantiles of normalised read coverage) at the promoters of all genes expressed in mouse ESCs (left panels, $n = 14,362$), and at the promoters of Zbtb11-dependent genes (right panels, $n = 154$). **f** Sequential ChIP enrichment values at a panel of Zbtb11-dependent and -independent promoters in wild-type mouse ESCs, using the indicated combinations of ChIP and reChIP antibodies (mean and SD of three replicates). Source data are provided as a Source Data file. **g** GABPa and NRF-1 ChIP enrichment values at a panel of Zbtb11-dependent and -independent promoters, in control and *Zbtb11* KO cells. Bars represent means and standard deviations of $n = 3$ (GABPa) and $n = 4$ (NRF-1) independent experiments (shown as open circles). *P* values for two-tailed *t* tests comparing control to *Zbtb11* KO samples are shown above each binding site. Source data are provided as a Source Data file.

not bind or regulate classical NRF-1 and NRF-2 targets such as the promoters of cytochrome *c*, mitochondrial transcription factors TFB or cytochrome *c* oxidase subunit genes (Supplementary Fig. 4b, c). At the promoters of DE genes, virtually all (98%) Zbtb11 peaks are also bound by GABPa, but only half of them are bound by NRF-1.

Analysing the strength of binding at shared sites, we found strong positive correlation genome-wide between Zbtb11 and GABPa ($r = 0.63$, *p* value $= 2.516e-16$), with the peaks at DE genes overwhelmingly showing strong affinity for both Zbtb11 and GABPa (Fig. 4d, left panel). By contrast, we found no correlation between Zbtb11 and NRF-1 ChIP-seq signal ($r = 0.04$, *p* value $= 0.012$) either genome-wide or at DE genes (Fig. 4d, right panel).

A promoter-centric analysis focusing on the ChIP-seq signal across the 1-kb region around the TSS of all genes expressed in mouse ESCs ($n = 14,362$) revealed a positive correlation between Zbtb11 and GABPa ($r = 0.49$) and between GABPa and NRF-1 ($r = 0.34$), but not between Zbtb11 and NRF-1 ($r = 0.10$) (Fig. 4e, left panel). When considering only the promoters of the DE genes, Zbtb11 is very strongly correlated with GABPa ($r = 0.74$), but only weakly correlated with NRF-1 ($r = 0.27$) (Fig. 4e, right panel). Our analyses therefore strongly indicate that Zbtb11 functionally interacts with NRF-2, but do not support an interaction between Zbtb11 and NRF-1.

The strong positive correlation between Zbtb11 and GABPa binding suggested that these factors may bind cooperatively at their shared targets. Alternatively, Zbtb11 and GABPa could bind independently to the same genomic sites, which could take place on the same or on different alleles in the cell population, their positive correlation potentially being driven by their ability to recognise the same DNA binding motif. To distinguish between these possibilities, we performed sequential ChIP with antibodies to Zbtb11 and GABPa. As seen in Fig. 4f, chromatin fragments bound by Zbtb11 are also strongly bound by GABPa, showing that Zbtb11 and GABPa bind simultaneously to their shared genomic sites. To explore the potential cooperative binding of these factors, we performed GABPa and NRF-1 ChIP in control and *Zbtb11* KO cells. This revealed that Zbtb11 depletion strongly impairs GABPa recruitment to shared Zbtb11/GABPa sites, but not to Zbtb11-independent sites (Fig. 4g). By comparison, NRF-1 recruitment was not affected by Zbtb11 depletion, either at shared Zbtb11/NRF-1 sites or at Zbtb11-independent sites (Fig. 4g). Zbtb11 therefore specifically controls the locus-specific recruitment of NRF-2 to its target promoters. Although Zbtb11 also shares some target genes with NRF-1, their association with these promoters appears to take place through independent mechanisms.

Altogether, our results strongly indicated that Zbtb11 cooperates with NRF-2 to regulate mitochondrial function.

**Zbtb11 depletion impairs mitochondrial respiration.** To test whether mitochondrial function is affected in *Zbtb11* KO cells, we first investigated the status of the mitochondrial membrane potential (MMP) by staining live cells with TMRE (tetramethylrhodamine, ethyl ester), a cationic fluorophore that permeates the plasma membrane and accumulates in the mitochondrial matrix in MMP-dependent manner. As can be seen in Fig. 5a (left panel), the TMRE staining decreased progressively after Zbtb11 depletion. This was not the result of fewer mitochondria, as the ratio of mitochondrial-to-nuclear DNA was the same in control and *Zbtb11* KO cells (Supplementary Fig. 5a). These results indicate that mitochondria gradually depolarise in *Zbtb11* KO cells, and show that Zbtb11 is required for the maintenance of functional mitochondria but not for mitochondrial replication.

To test the ability of *Zbtb11* KO cells to maintain their MMP in more stressful circumstances than the resting state, for example, in situations when ATP demands suddenly increase, we challenged them with increasing concentrations of the mitochondrial uncoupler FCCP (carbonyl cyanide 4-(trifluoromethoxy) phenylhydrazone). FCCP allows protons to flow back into the mitochondrial matrix, short-circuiting the MMP and stimulating the ETC complexes, which respond by increasing their activity to restore the cross-membrane proton gradient[24]. We stained control and *Zbtb11* KO cells with TMRE in the presence of increasing concentrations of FCCP and subsequently analysed them by flow cytometry. As seen in Fig. 5a (right panel) and Supplementary Fig. 4b (left panel), the MMP in *Zbtb11* KO cells showed increased sensitivity to FCCP, which gradually intensified following Zbtb11 depletion. These effects were not observed in $Zbtb11^{wt/wt} R26^{wt}$ cells treated with 4OHT (Supplementary Fig. 5b, right panel).

The inability of *Zbtb11* KO cells to maintain polarised mitochondria could be caused either by decreased respiration or by an increase in proton leak or ATP demand. To determine whether respiration is impaired in Zbtb11-deficient cells, we used the Seahorse platform to measure oxygen consumption rates (OCRs), which in combination with sequential addition of specific mitochondrial inhibitors (oligomycin, FCCP and a mixture of rotenone and antimycin A) allows the derivation of several mitochondrial respiration parameters, as previously described[24] (see Methods). This revealed that respiration is already significantly impaired 48 h post KO induction, and it further deteriorates 24 h later (72 h post KO induction) (Fig. 5b). Basal respiration was the most affected, both its constituent parameters (leak respiration and ATP synthesis) showing the largest decrease of all parameters measured, which shows that mitochondrial depolarisation in *Zbtb11* KO cells is caused by deficiencies in the ETC and not by increased proton leak or ATP turnover. The coupling efficiency between respiration and ATP

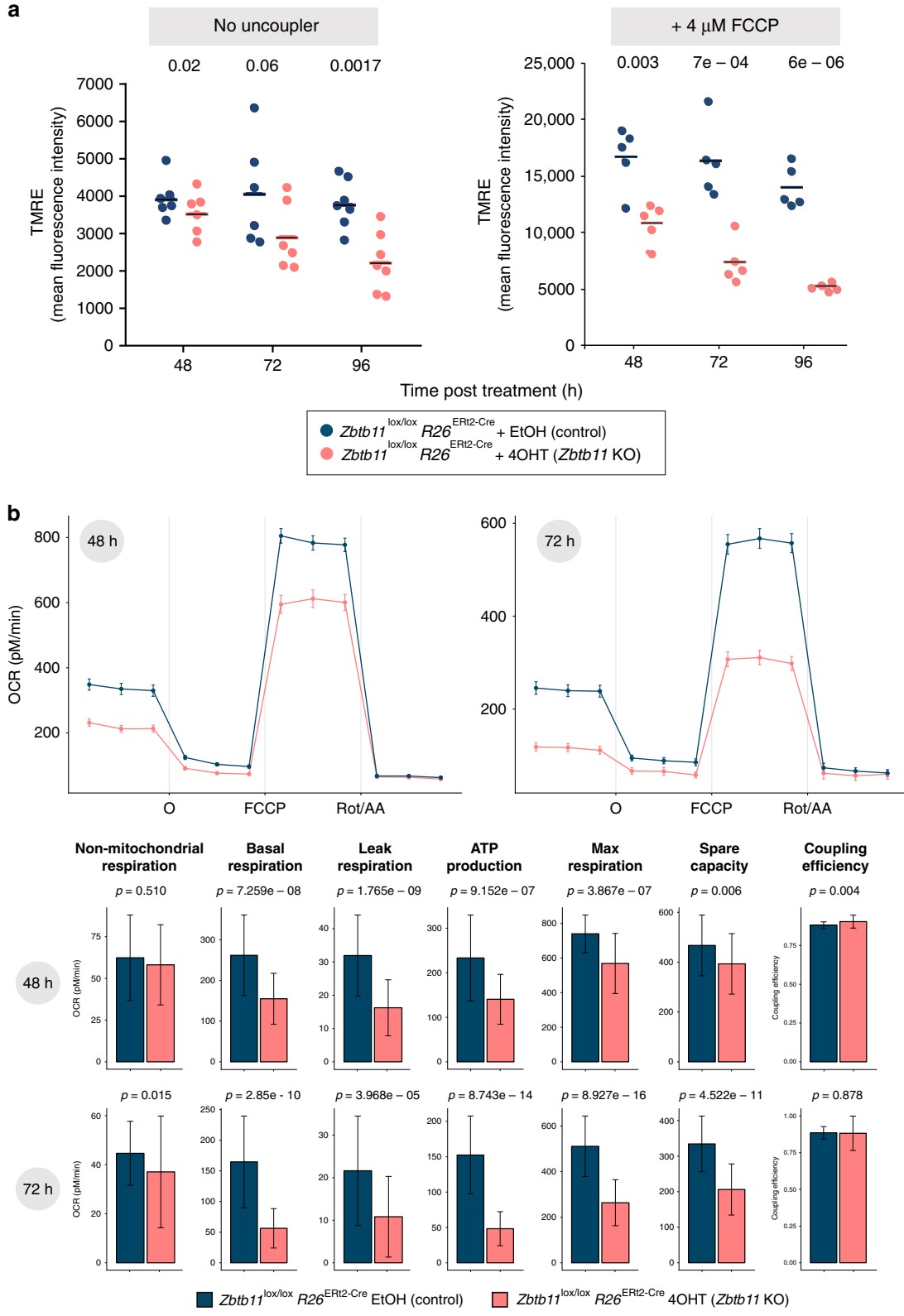

synthesis did not change in *Zbtb11* KO cells at any of the time points assessed, which indicates that Zbtb11 depletion has no direct effect on the proton leak, and that the reduction in leak respiration is just a consequence of a reduced proton flux through the system. The maximal respiratory capacity was also decreased in *Zbtb11* KO cells — although less affected than basal respiration to begin with, it subsequently deteriorated significantly, in

agreement with the observed increasing sensitivity of the MMP to the uncoupling agent FCCP (Fig. 5a, right panel).

Altogether, these results show that *Zbtb11* KO cells have a deficient OXPHOS system, which fails to maintain the MMP and thus to supply an adequate proton-motive force, both at resting state and when forced to function at maximum capacity. Importantly, the decline of mitochondrial functions in our

**Fig. 5 Zbtb11 depletion results in impaired mitochondrial respiration. a** Changes in mitochondrial membrane potential (MMP) caused by Zbtb11 depletion. Control and Zbtb11 KO cells were stained with TMRE (tetramethylrhodamine, ethyl ester) at the indicated time following KO induction. Mean fluorescence intensity obtained by flow cytometry and their means are shown in samples without (left) and with 4 μM FCCP (right). Individual data points and means are shown for $n = 6$ (no FCCP) and $n = 5$ (FCCP-treated) independent experiments, respectively. P values for paired two-tailed t tests are shown above. See also Supplementary Fig. 5b for full FCCP titration curves. **b** Decreased mitochondrial respiration in Zbtb11 KO cells. Zbtb11[lox/lox] Rosa26[ERt2-Cre] cells were treated with either EtOH or 4OHT for 48 or 72 h, and oxygen consumption rates (OCR) were measured using the Seahorse platform before and after the sequential addition of the ATP synthase inhibitor oligomycin (O), the mitochondrial uncoupler FCCP, and a mix of rotenone (complex I inhibitor) and antimycin A (complex III inhibitor) (Rot/AA). Upper panels—OCR values adjusted for gDNA content (measured by qPCR and used as a proxy for cell number), showing mean ± SEM of 8 (48 h) and 7 (72 h) biological replicates, and a total of 44–48 measurements (technical × biological). Lower panels—mitochondrial respiration parameters (mean ± SD) in control and Zbtb11 KO cells, calculated from OCR measurements (see Methods). P values (Wilcoxon's rank-sum test) are indicated above each pair.

experimental system closely followed the downregulation of Zbtb11-dependent mitochondrial proteins, but preceded any overt signs of cell death or cell cycle arrest.

**Zbtb11 controls complex I biogenesis**. To determine more specifically how Zbtb11 depletion affects mitochondrial respiration, we performed pathway mapping and over-representation analysis using the list of Zbtb11-dependent genes with known or predicted mitochondrial functions. This identified 'complex I biogenesis' and 'mitochondrial translation' as the most likely affected processes (FDR < 0.05, Fig. 6a and Supplementary Table 3). There were four genes involved in the biogenesis of respiratory complex I, encoding one core subunit (Ndufs7), two accessory subunits (Ndufa12 and Ndufc2), and one assembly factor (Ndufaf1) (Fig. 6a and Supplementary Table 3). There were eight DE genes involved in mitochondrial translation, all of which encode subunits of the mitochondrial ribosome (Fig. 6a and Supplementary Table 3). All DE genes that mapped to mitochondrial pathways were downregulated, and the effect size was particularly pronounced at Ndufc2 and Ndufaf1 (5.5- and 4-fold, respectively, see Fig. 6a). This acute depletion was also reflected at the protein level, as Ndufc2 was virtually undetectable by immunoblotting 72 h after Zbtb11 KO induction (Fig. 3e). These results suggested that Zbtb11 KO cells may be defective in the synthesis and assembly of respiratory complex I. In addition, these cells may also have a compromised mitoribosome, which would cause a general defect in mitochondrial translation, and therefore have a knock-on effect on the synthesis of all OXPHOS complexes with mtDNA-encoded subunits (I, III, IV and V).

To assess the integrity of respiratory complexes in Zbtb11 KO cells, we carried out immunoblotting with antibodies against specific subunits that become degraded when their respective OXPHOS complex is unstable or fails to assemble[25–27]. We found a striking downregulation of complex I integrity marker Ndufb8 (Fig. 6b), despite the fact its transcript is expressed at normal level (fold-change 1.006, adjusted p value = 0.99). This indicated that the downregulation of the complex I subunits Ndufc2, Ndufaf1, Ndufa12 and Ndufs7, resulting from Zbtb11 depletion, was enough to disrupt the biogenesis of complex I, which in turn lead to the degradation of the stability marker Ndufb8. The integrity markers for complex III (Uqcrc2) and complex IV (Mtco1) were also downregulated, but to a lesser extent and only after prolonged Zbtb11 depletion (Fig. 6b). By comparison, the expression of complex II and V markers remained unchanged at all times (Fig. 6b). These results indicated that complex I biogenesis is specifically affected in the immediate aftermath of Zbtb11 depletion, in agreement with the prediction by the differential gene expression (DGE) analysis.

Further immunoblotting experiments showed that not all complex I subunits are degraded (Fig. 6c), indicative of an assembly defect rather than degradation of the entire holocomplex, which would be consistent with the strong downregulation of the accessory subunit Ndufc2 and assembly factor Ndufaf1. To

assess this, we separated mitochondrial extracts by blue native-polyacrylamide gel electrophoresis (BN-PAGE) and subsequently performed immunoblotting with antibodies against Ndufb11 — a subunit that is not degraded as a result of Zbtb11 depletion (Fig. 6c). This showed that 48 h post Zbtb11 KO induction, the holocomplex was already reduced, while a smaller Ndufb11-containing subcomplex was increased, consistent with an assembly defect (Fig. 6d). Subsequently, the amount of complex I holocomplex continued to decrease, and at 96 h both the holocomplex and the subcomplex were barely detectable with Ndufb11 antibodies (Fig. 6d), indicating the subcomplex also becomes unstable or fails to assemble. Altogether, these data show that Zbtb11 depletion leads to a severe complex I assembly defect.

Consistent with this biogenesis defect, spectrophotometric quantification of complex I activity in mitochondrial extracts (see Methods) revealed this was gradually lost in Zbtb11 KO cells (Fig. 6e), strongly indicating that the reduction in mitochondrial respiration that follows Zbtb11 depletion (Fig. 5b) is underpinned to a significant extent by complex I insufficiency. The impairment of complex I activity also provides a potential explanation for the proliferation arrest in Zbtb11 KO cells. NADH oxidation by complex I maintains the cellular pool of $NAD^+$, which functions as an electron acceptor in several biosynthetic pathways required for cellular growth and proliferation[4,5]. A critical consequence of ETC disruption and $NAD^+$ depletion is aspartate insufficiency, which becomes restrictive for proliferation[4,5]. Consistent with aspartate insufficiency taking place in Zbtb11 KO cells, cell cycle and proliferation arrest only become evident (Fig. 2b, c) once complex I activity deteriorates significantly 96 h post KO induction (Fig. 6e). Proliferation arrest induced by aspartate insufficiency can be rescued in several somatic cell lines by supplementing the media with either aspartate directly, or pyruvate as surrogate electron acceptor[4,5]; however, it is not yet known whether this is also the case in ESCs. To investigate this, we cultured control and Zbtb11 KO cells in either standard ESC media, or with supraphysiological levels of pyruvate or aspartate (see Methods). In the immediate aftermath of Zbtb11 depletion, when Zbtb11 KO cells in standard media are still able to proliferate, supplementation of the media did not have a significant effect on cellular proliferation (Fig. 6f, upper panel). However, at 96 h post KO induction when complex I activity is dramatically reduced, Zbtb11-deficient cells can only keep proliferating if pyruvate or aspartate is added to the media (Fig. 6f, lower panel). These results indicate that aspartate insufficiency caused by disruption of complex I is an important contributor to proliferation arrest in Zbtb11 KO cells.

Finally, to understand whether the impaired biogenesis of respiratory complexes in Zbtb11 KO cells is underpinned by a mitoribosome defect, we first tested if pharmacological inhibition of the mitoribosome mimics the respiratory complexes defect that follows Zbtb11 depletion. Blocking mitochondrial translation with chloramphenicol in otherwise untreated Zbtb11[lox/lox]

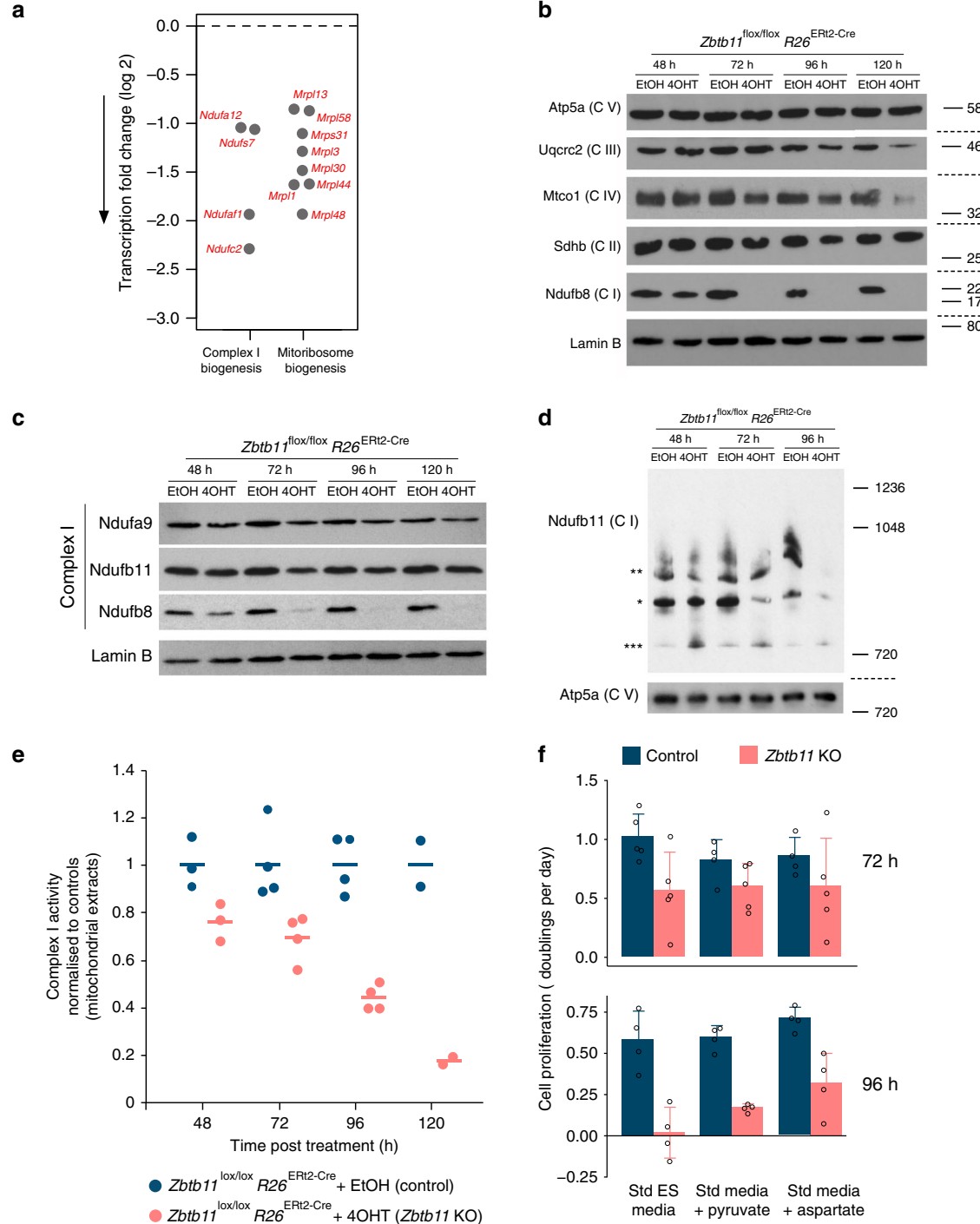

**Fig. 6 Zbtb11 controls assembly and stability of respiratory complex I. a** Differentially expressed genes mapping to enriched mitochondrial pathways (see also downregulation of Ndufc2 at the protein level in Fig. 3e). **b** Immunoblots of whole-cell lysates from control and *Zbtb11* KO cells, probing stability markers of all five OXPHOS complexes. Molecular weight marker unit is kDa. Note that none of the stability markers probed in the blot are affected at the transcriptional level by Zbtb11 depletion; the downregulation of Ndufb8 is a consequence of complex I instability caused by downregulation of other complex I subunits that are under direct Zbtb11 control, as listed in **a** (Fig. 3e). Source data are provided as a Source Data file. **c** SDS-PAGE and immunoblotting of complex I subunits in whole-cell lysates of control and *Zbtb11* KO cells showing that not all complex I subunits are downregulated. Lamin B is shown as a loading control. Source data are provided as a Source Data file. **d** Immunoblot of native mitochondrial extracts separated by BN-PAGE. Complex I was detected with antibodies to the Ndufb11 subunit (which was not affected by Zbtb11 depletion, **c**). Note the reduction in complex I holocomplexes (*) and supercomplexes (**), and the increase in a smaller Ndufb11-containing assembly intermediate (***). Complex V detected with anti-Atp5a antibodies is shown as a loading reference. Molecular weight marker unit is kDa. Source data are provided as a Source Data file. **e** Spectrophotometric quantification of complex I activity in mitochondrial extracts from control and *Zbtb11* KO cells. Individual data points and means are shown for $n = 3$ (48 h), $n = 4$ (72 and 96 h) and $n = 2$ (120 h) independent experiments. **f** Rates of cell proliferation in control and *Zbtb11* KO cells grown in standard ES media (Std ES), and in media supplemented with supraphysiological amounts of pyruvate (Pyr) or aspartate (Asp). Bars represent mean and standard deviation of $n = 5$ (72 h) and $n = 4$ (96 h) independent experiments (shown as open circles). Source data are provided as a Source Data file.

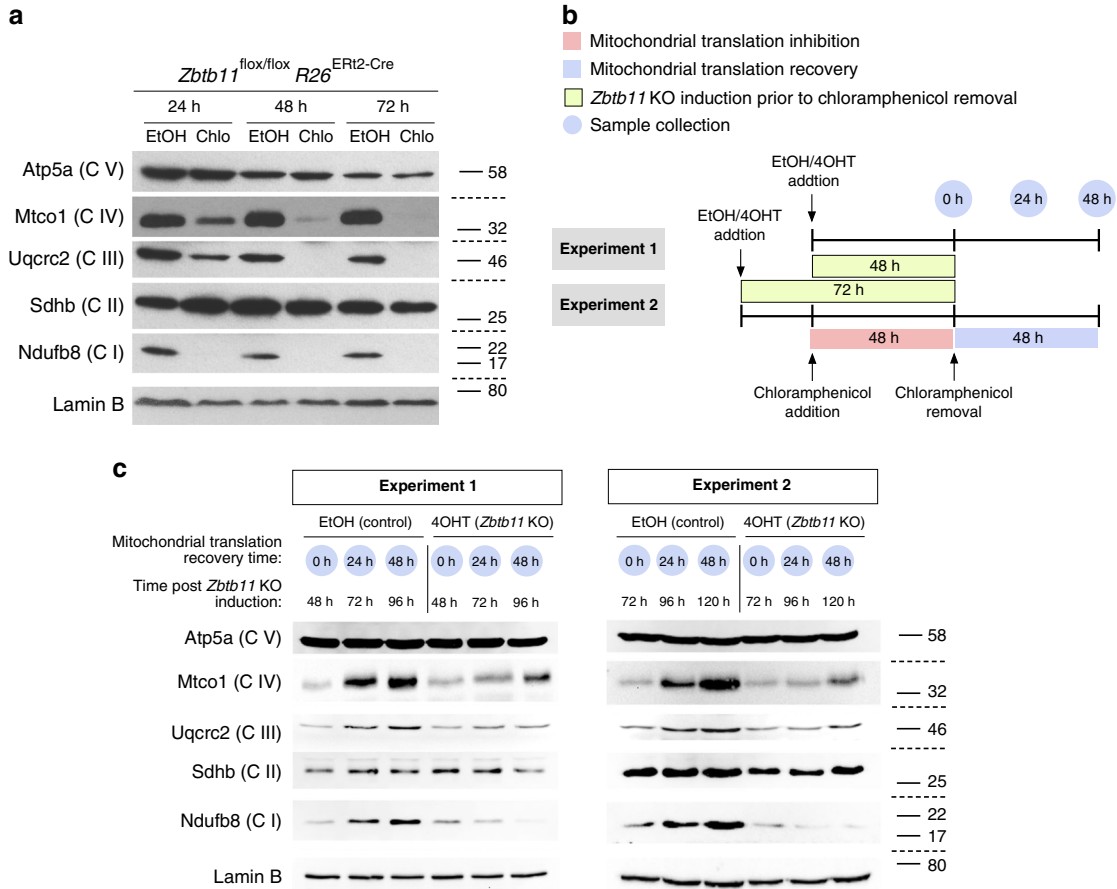

**Fig. 7 Zbtb11 is required for the biogenesis of respiratory complexes I, III and IV. a** SDS-PAGE and immunoblotting of whole-cell extracts following inhibition of mitochondrial translation with chloramphenicol. Samples from cells treated with ethanol (carrier) were used as reference. Note that complexes I, III and IV are sensitive to the mitochondrial translation block. Lamin B is shown as a loading control. Molecular weight marker unit is kDa. Source data are provided as a Source Data file. **b** Diagram of the experimental design used to compare de novo synthesis of respiratory complexes in control and *Zbtb11* KO cells. Fully assembled complexes I, III and IV were depleted from *Zbtb11*^lox/lox *Rosa26*^ERt2-Cre cells by inhibiting mitochondrial translation for 48 h (as in **a**). Mitochondrial translation was then allowed to resume by removing the chloramphenicol treatment, and samples for immunoblotting were collected at 24 h intervals in order to monitor the regeneration of respiratory complexes. The chloramphenicol treatment was synchronised with the addition of 4OHT (or EtOH as control), so that translation was allowed to resume either 48 h (experiment version 1) or 72 h (experiment version 2) post *Zbtb11* KO induction, thus allowing to compare de novo synthesis of respiratory complexes in control and *Zbtb11* KO cells. **c** Immunoblotting analyses of representative experiments comparing de novo synthesis of respiratory complexes I, III and IV, in control and *Zbtb11* KO cells, as outlined in **b**. Molecular weight marker unit is kDa. Note that the synthesis of complexes III and IV is only partly impaired in *Zbtb11* KO cells, while the synthesis of complex I is completely blocked. Source data are provided as a Source Data file.

*R26*^ERt2-Cre cells resulted in complete depletion of complexes I, III and IV within 48 h of treatment, while complexes II and V remained unaffected (Fig. 7a). This shows that mitochondria-encoded subunits of complexes I, III and IV are rapidly turned over, leading to the collapse of these complexes 48 h after their synthesis is blocked. This phenotype is largely recapitulated in 120 h *Zbtb11* KO cells (Fig. 6b and Supplementary Fig. 6), when a milder downregulation of complexes III and IV is also apparent, despite the fact that Zbtb11 does not control the expression of any of their subunits; therefore, suggesting that mitochondrial translation may become impaired 72 h post *Zbtb11* KO induction. To test this possibility, we compared the dynamics of mitoribosome-dependent synthesis of respiratory complexes in control and *Zbtb11* KO cells. We first depleted complexes I, III and IV by treating cells with chloramphenicol for 48 h, after which we removed the mitochondrial translation block, allowing it to resume, and subsequently monitored the regeneration of the respiratory complexes (Fig. 7b). By synchronising the chloramphenicol and EtOH/4OHT treatments in *Zbtb11*^lox/lox *R26*^ERt2-Cre

cells, we allowed mitochondrial translation to resume either 48 or 72 h post *Zbtb11* KO induction (Fig. 7b). Upon release from the mitochondrial translation block, control cells resumed synthesis of mitochondrially encoded proteins and regenerated all respiratory complexes within 48 h (Fig. 7c). By contrast, in *Zbtb11* KO cells complex I biogenesis was completely blocked and never recovered after chloramphenicol removal, while synthesis of complex III and IV was partly impaired but not fully arrested (Fig. 7c). These results are consistent with a partial impairment of mitochondrial translation, leading to reduced complex III and IV synthesis, but which in the case of complex I is compounded or fully masked by the specific and more severe biogenesis defect caused by the downregulation of complex I subunits directly controlled by Zbtb11 (Supplementary Table 2).

In conclusion, Zbtb11 specifically controls complex I biogenesis by promoting the expression of key accessory and assembly subunits, while also exerting — albeit to a lesser extent — broader control over the synthesis of respiratory complexes, likely through its role in mitoribosome biogenesis.

**Pathogenic mutations destabilise the Zbtb11 protein**. In order to investigate the relevance of our findings to the aetiology of *ZBTB11*-associated ID, we modelled the mutations associated with this disease in our experimental system. Two *ZBTB11* missense mutations have been identified in families with hereditary ID, both of which lead to single amino acid substitutions of key histidine residues (H729Y and H880Q)[12] in C2H2 zinc-finger motifs that are conserved between mouse and humans. We induced *Zbtb11* KO and rescued its expression by ectopically expressing either WT or mutant FLAG-Zbtb11 cDNA (H729Y or H880Q). The cDNA was linked to an IRES-GFP reporter, which allowed us to use flow cytometry to sort successfully transfected cells with the same level of cDNA expression. As seen in Fig. 8a, although the WT and the two mutant FLAG-Zbtb11 cDNA were expressed at comparable levels, only WT Zbtb11 was able to fully rescue the expression of complex I and mitoribosome biogenesis genes. Complex I biogenesis genes *Ndufc2* and *Ndufaf1* were particularly affected by the Zbtb11 mutations, their transcripts being downregulated 2-fold when Zbtb11 expression was rescued with H880Q and H729Y mutants, respectively. By comparison, the maximum effect size among the mitoribosome genes was 1.6-fold downregulation as a result of mutating Zbtb11.

Immunoblotting with antibodies to the complex I stability marker Ndufb8 showed that the inability of mutant Zbtb11 to fully support transcription of complex I biogenesis genes results in reduced amounts of complex I (Fig. 8b). Surprisingly, immunoblotting with anti-FLAG antibodies revealed that mutant FLAG-Zbtb11 protein levels were also reduced ~2-fold for both mutants (Fig. 8b), despite their respective cDNA transcripts being at least as abundant as WT FLAG-Zbtb11 (Fig. 8a). This shows that the pathogenic mutations in the zinc-finger motifs of Zbtb11 cause the protein to be unstable or mark it for degradation.

To frame our findings in the context of human tissue-specific gene expression, we integrated our results with the Genotype-Tissue Expression (GTEx) dataset (V8), which contains whole-transcriptome data from 948 individuals across 52 different primary tissues. This revealed that while *ZBTB11* is ubiquitously expressed, it is most highly expressed in samples from the cerebellum (Fig. 8c) — in agreement with an important function for Zbtb11 in this tissue, also indicated by the fact that patients with *ZBTB11* mutations have cerebellar atrophy[12]. Of the 154 Zbtb11-dependent genes we identified in mouse ES cells, 149 were successfully matched to human orthologues. Their expression was highly correlated with *ZBTB11* (Fig. 8c), supporting the notion that the regulatory relationship of Zbtb11 with its target genes is conserved in mammals.

Altogether, our results indicate that the effect of human *ZBTB11* mutations is mediated at least in part by a reduction in the amount of Zbtb11 protein available to activate its target genes. This is likely to be particularly detrimental in tissues requiring higher Zbtb11 expression, such as the cerebellum, and may be further compounded by a defect in the ability of mutant Zbtb11 to recognise its cognate targets through the zinc-finger motifs. Importantly, these mutations impair complex I biogenesis, indicating that the phenotype in affected patients may be, at least partly, the manifestation of a mitochondrial disease. This is supported by the phenotypical overlap with other mitochondrial diseases, which are also commonly characterised by cerebellar atrophy, ataxia and hypotonia[28].

## Discussion

There are currently approximately 700 genes known to be involved in the genetics of ID[29], but it is estimated the total number may be >2500 (ref. [30]). Large-scale studies of consanguineous families with inherited ID have identified new candidate loci, including several genes encoding proteins with unknown function[11,31]. Determining the molecular and cellular functions of these proteins is a crucial step towards understanding the aetiology of the different forms of ID. Zbtb11 is a ubiquitously expressed transcription factor recently found to be mutated in two families with inherited ID[11,12], but which had remained largely uncharacterised. In this study, we performed a systematic analysis of its cellular functions.

Consistent with the findings of previous genome-wide genetic screens, we found Zbtb11 is an essential factor required for cell proliferation and survival, which explains why only missense mutations, but no homozygous loss of function variants, have been identified so far in ID patients or in the Genome Aggregation Database[32]. Using an inducible KO system allowed us to perform timed deletion experiments and detect the early transcriptional changes at genes under direct Zbtb11 control. Zbtb11 appears to function as a transcriptional activator, contrasting with the majority of previously characterised ZBTB proteins, most of which have been reported to function as transcriptional repressors[10].

A recent study reported that in zebrafish, Zbtb11 directly regulates p53 by binding to its gene promoter and repressing transcription[33]. Inspection of our ChIP-seq datasets also revealed binding of Zbtb11 at the promoter of p53 gene (*Trp53*), but *Trp53* was not deregulated in our RNA-seq experiment (fold-change = 0.93, adjusted *p* value = 0.99) (Supplementary Fig. 7a). qRT-PCR analyses confirmed that *Trp53* transcripts do not change in *Zbtb11* KO cells, even after prolonged Zbtb11 depletion 72 h post KO induction (Supplementary Fig. 7b), leading us to conclude that Zbtb11 does not directly regulate the expression of p53 in mouse ESCs. The discrepancy between our results and the previous study may be due to species-specific differences. Alternatively, the zebrafish *Zbtb11* missense mutation found to upregulate p53 may act as a gain-of-function mutation that potentiates the transcriptional activating function of Zbtb11.

We find instead that in mammalian cells Zbtb11 promotes the expression of a select group of housekeeping genes that are significantly enriched in genes encoding mitochondrial proteins. Our data strongly indicate that its target genes and mechanisms of activation by Zbtb11 are conserved between cell types and across mammalian species. Promoting the expression of this subset of mitochondrial proteins has functional importance, as Zbtb11 depletion significantly impairs respiration and leads to mitochondrial depolarisation. These changes precede detectable increases in cell death, indicating they are not a consequence of apoptosis. Maintenance of mitochondrial function is therefore a key role of Zbtb11.

Mechanistically, we show that Zbtb11 is an integral part of the network of nuclear transcription factors, which controls mitochondrial activity and biogenesis, being responsible for the recruitment of NRF-2/GABP to a subset of its target sites. Zbtb11 shares a large proportion of its genomic binding sites with NRF-2/GABP and our experiments established that these two factors bind simultaneously at their common targets, where NRF-2/GABP recruitment is heavily dependent on Zbtb11. The association between Zbtb11 and NRF-2/GABP appears to be conserved in mammals, and importantly, it is a signature of virtually all Zbtb11-high sites and promoters of Zbtb11-dependent genes, indicating that recruitment of NRF-2/GABP is part of the mechanism through which Zbtb11 controls its targets genes. NRF-2/GABP is a potent promoter activator, as recently illustrated by the finding that in a large proportion of glioblastomas the telomerase gene is reactivated by promoter mutations that generate novel NRF-2/GABP binding sites[34]. The transcriptional activating role of Zbtb11 can therefore be explained by its ability to facilitate the recruitment of NRF-2/GABP to promoters.

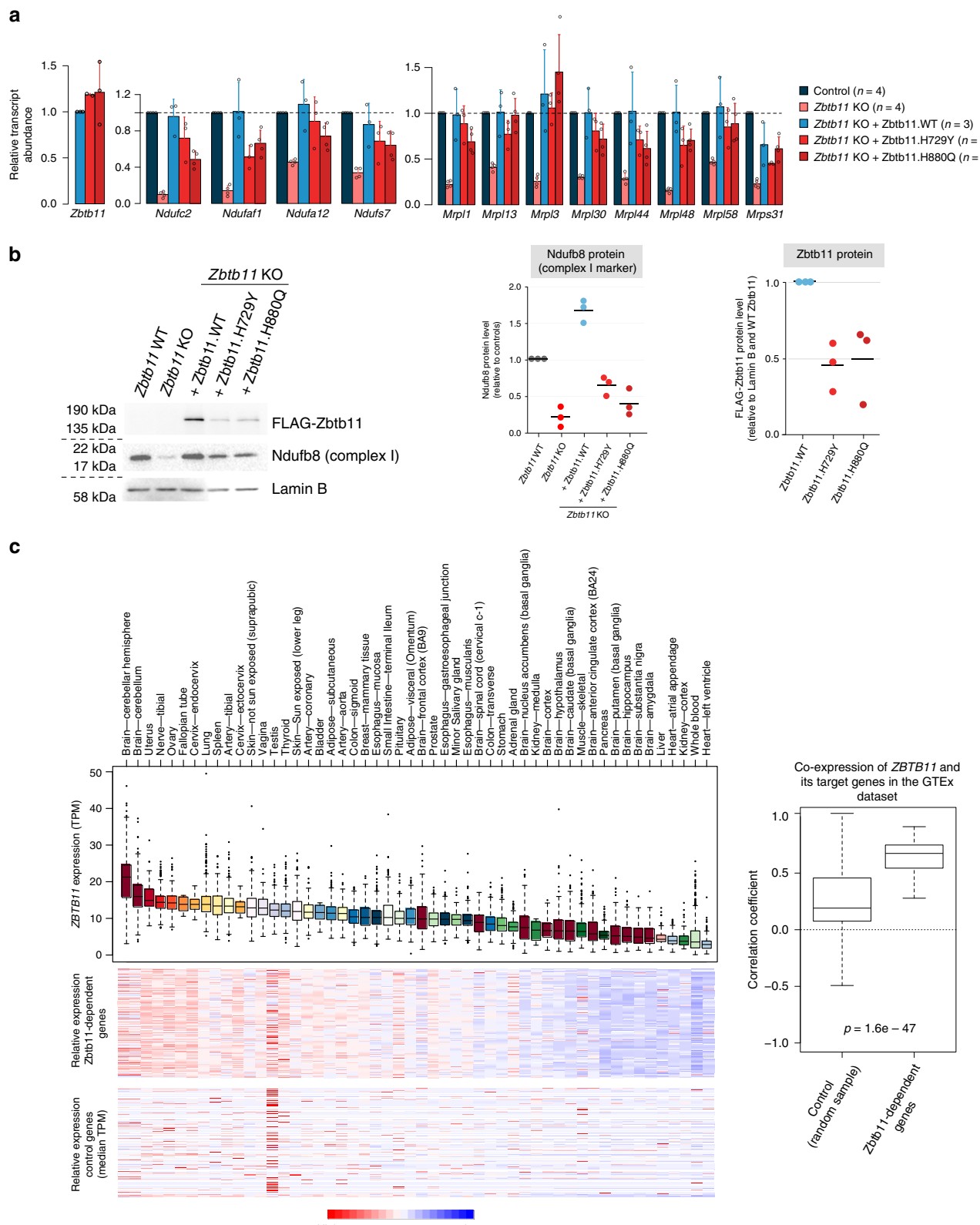

Interestingly, despite their extensive genome-wide association, NRF-2/GABP also binds to a number of promoters, including some of its known classical targets, where its binding is not Zbtb11-dependent. This shows that NRF-2/GABP does not bind its targets through a uniform mechanism, but its recruitment is instead regulated in locus-specific manner. This may reflect the existence of locus-specific complexes in which NRF-2/GABP

associates with different factors. Alternatively, Zbtb11 may facilitate promoter activation by regulating the local chromatin structure to increase TSS accessibility for NRF-2/GABP.

Zbtb11 was found to co-localise with NRF-1 at the promoters of several genes, the majority of which, however, are also bound by NRF-2/GABP. At these sites, Zbtb11 specifically controls the recruitment of NRF-2/GABP, but not NRF-1, showing that

**Fig. 8 Mutations associated with intellectual disability destabilise the Zbtb11 protein and impair complex I biogenesis. a** qPCR quantification of transcripts in control and *Zbtb11* KO cells rescued with either wild-type (WT) or mutant Zbtb11 cDNA. *Zbtb11*[lox/lox] *Rosa26*[ERt2-Cre] cells were treated with either EtOH (Control) or 4OHT (*Zbtb11* KO), and concomitantly transfected with a plasmid expressing either WT or the indicated mutant FLAG-Zbtb11 cDNA fused to an IRES-GFP reporter. Forty-eight hours later, successfully transfected cells were sorted based on GFP expression, gating on cells with intermediate fluorescence, thus selecting cells expected to have similar levels of cDNA expression. Bars represent mean and standard deviation. Source data are provided as a Source Data file. **b** Immunoblot of whole-cell lysates of *Zbtb11* KO cells rescued with either WT or mutant Zbtb11 cDNA. Cells treated and sorted as in **a**, except collected 72 h post *Zbtb11* KO induction, were analysed by immunoblotting for the complex I stability marker (Ndufb8) and FLAG-Zbtb11 expression. Lamin B was used as a loading reference. Left panel shows a representative image of the results, while the two panels on the right show the quantification of the blots from three biological replicates. Source data are provided as a Source Data file. **c** Correlation of expression between Zbtb11 and its target genes in primary human tissues. Box plot to the left shows normalised expression of human *ZBTB11* across primary tissues sampled in the GTEx dataset. The heatmaps below represent the same tissues as in the box plot and median expression values for the human orthologues of Zbtb11-dependent genes (upper panel), and for an equal number of randomly selected genes as control (lower panel). Box plot to the right shows distribution of correlation coefficients for expression of *ZBTB11* and that of individual target genes across the GTEx tissues. $P = p$ value for two-sided Wilcoxon's rank-sum test. Box plots show the interquartile range (box outline) and median value (horizontal line), with the whiskers delineating the lower and upper limits of the data. Number of individual samples (*n*) in the GTEx database varies for each individual tissue, as follows: brain—cerebellar hemisphere = 215; brain—cerebellum = 241; uterus = 142; nerve—tibial = 619; ovary = 180; fallopian tube = 9; cervix—endocervix = 10; lung = 578; spleen = 241; artery—tibial = 663; cervix—ectocervix = 9; skin—not sun exposed (suprapubic) = 604; vagina = 56; testis = 361; thyroid = 653; skin—sun exposed (lower leg) = 701; artery—coronary = 240; bladder = 21; adipose—subcutaneous = 663; artery—aorta = 432; colon—sigmoid = 373; breast—mammary tissue = 459; oesophagus—mucosa = 555; small intestine—terminal ileum = 187; pituitary = 283; adipose—visceral (omentum) = 541; brain—frontal cortex (ba9) = 209; prostate = 245; oesophagus—gastroesophageal junction = 375; minor salivary gland = 162; oesophagus—muscularis = 515; brain—spinal cord (cervical c-1) = 159; colon—transverse = 406; stomach = 359; adrenal gland = 258; brain—nucleus accumbens (basal ganglia) = 246; kidney—medulla = 4; brain—cortex = 255; brain—hypothalamus = 202; brain—caudate (basal ganglia) = 246; muscle—skeletal = 803; brain—anterior cingulate cortex (ba24) = 176; pancreas = 328; brain—putamen (basal ganglia) = 205; brain—hippocampus = 197; brain—substantia nigra = 139; brain—amygdala = 152; liver = 226; heart—atrial appendage = 429; kidney—cortex = 85; whole blood = 755; heart—left ventricle = 432.

---

although NRFs overlap extensively genome-wide, their activity can be distinctly regulated even at shared promoters.

Zbtb11 controls the biogenesis of complex I by promoting the expression of nuclear-encoded complex I subunits with core (Ndufs7), accessory (Ndufc2 and Ndufa12) and assembly (Ndufaf1) roles, Ndufc2 and Ndufaf1 being particularly sensitive to Zbtb11 depletion (Supplementary Data). An abundance of evidence from studies of human disease[35–37] and in vitro experimental approaches[38] show that inactivating either one of these factors individually leads to severe defects in complex I assembly, so their combined downregulation in *Zbtb11* KO cells is expected to lead to the collapse of complex I—an outcome that is reflected in our results. Zbtb11 also directly controls the expression of mitoribosomal protein components, and its depletion eventually caused a broad defect in the biogenesis of respiratory complexes that partly recapitulates the pharmacological inhibition of mitochondrial translation. This effect, however, was not as pronounced as the specific impairment of complex I biogenesis, which may be due to the fact that the majority of the mitoribosome genes were not downregulated to the same extent as the complex I genes *Ndufc2* and *Ndufaf1*.

Zbtb11 therefore maintains the homeostasis of the OXPHOS system primarily by activating transcription at genes of nuclear-encoded complex I subunits, as well as by indirectly facilitating the expression of mitochondria-encoded subunits at the translational level. In its absence, the ETC is disrupted, leading to cell proliferation arrest. The mechanism by which ETC disruption leads to proliferation arrest even in cells with high glycolysis rates was recently revealed to be aspartate restriction caused by depletion of NAD+ (refs. [4,5]). A functional ETC, and complex I activity in particular, is required to maintain the pool of available NAD+, which functions as an essential cofactor in the process of aspartate synthesis in the mitochondrial matrix. In *Zbtb11* KO cells, the activity of complex I gradually decreases until it reaches a critical level that can no longer support proliferation and cells arrest in G1. The G1 arrest is consistent with a metabolic restriction that blocks entry into S phase, in this case underpinned by aspartate depletion. This is evidenced by the fact that provision of aspartate is able to rescue proliferation specifically at

the stage when Zbtb11-deficient cells first show signs of cell cycle arrest (96 h post KO induction) (Fig. 6f).

Experimental modelling of the missense Zbtb11 mutations associated with ID showed that these have a destabilising effect on the Zbtb11 protein. Both mutations lead to the substitution of the second histidine residue in C2H2 zinc-finger motifs, which is predicted to disrupt the folding of the motifs, and protein misfolding may accelerate its degradation. The overall effect of the mutations is therefore equivalent to a Zbtb11 knockdown, which could be further compounded by the fact that the mutant protein may be impaired in its ability to bind and activate target promoters. The complex I genes *Ndufc2* and *Ndufaf1* were particularly sensitive to the reduced Zbtb11 dosage, and consequently, Zbtb11 mutations lead to impaired complex I biogenesis. Our findings may help understand the aetiology of this form of ID, as neuronal development, survival and activity, are heavily reliant on mitochondria for ATP production and for regulation of calcium and redox signalling. Mitochondria play key roles in axonal specification and growth[39], as well as in synaptic plasticity[40], and their importance in neuronal development and survival is underlined by the fact that many mitochondrial disorders are characterised by neurological abnormalities. Of note, cerebellar atrophy and ataxia are prominent features in patients with mtDNA pathogenic mutations[28], and are also evident in patients with *ZBTB11* mutations. Moreover, at least two genes that we find to be directly regulated by Zbtb11, *NDUFA12* and *NDUFAF1*, are known to be mutated in mitochondrial diseases (Leigh syndrome and cardioencephalomyopathy, respectively)[37,38]. We therefore propose that at least some of the phenotypical features reported in patients with *ZBTB11* mutations are manifestations of a mitochondrial disease. Further studies in vivo are required to determine the exact aetiology of this syndrome.

## Methods
**Cells**. Mouse ES cell line E14 (ref. [16]) and its derivatives were maintained on plates pre-coated with 0.2% gelatin (G1393, Sigma). Media were KO DMEM (Dulbecco's modified Eagle's medium) (10829018, Gibco), supplemented with foetal bovine serum (10%) (10270106, Thermo Fisher Scientific), non-essential amino acids (11140035, Gibco), glutamine (2 mM), β-mercaptoethanol, penicillin and

streptomycin, and leukemia inhibitory factor (ESG1106, Merck-Millipore). Proliferation of *Zbtb11* KO cells was rescued by supplementing the media with 1 mM pyruvate and 10 mM aspartate. Proliferation rates were calculated as previously described[5], doublings/day = log$_2$ (cell number assay day/cell number first day)/ number of days in culture.

**CRISPR/Cas9-mediated genome editing.** Guide RNA (gRNA) sequences were synthesised as single-stranded oligos (Sigma-Aldrich), annealed and cloned in the vectors PX458 (pSpCas9(BB)-2A-GFP, Addgene plasmid # 48138) or PX461 (pSpCas9n(BB)-2A-GFP, Addgene plasmid # 48140), both a gift from Feng Zhang[41]. Recombinant plasmids were verified by Sanger sequencing and electroporated into E14 cells using the Nucleofector II platform (Lonza) (programme A-013). At 48 h post electroporation, GFP-expressing cells were FACS-sorted and cloned by serial dilution in 96-well plates. Single-cell colonies were allowed to grow for a week and subsequently screened by PCR. Colonies detected as recombinant were further analysed by Sanger sequencing, by PCR amplifying the targeted locus, cloning the PCR amplicons in the pCR2.1 TOPO plasmid (Thermo Fisher Scientific) and transforming 10-beta competent *Escherichia coli* bacteria (New England Biolabs). Plasmid inserts from 20 bacterial colonies were sequenced for each ES cell colony to verify homozygosity.

**Generation of the *Zbtb11*$^{FLAG/FLAG}$ E14 line.** The *Zbtb11* ATG start codon was targeted using two gRNAs (GCCGGCGGCTATGTCAAGCG and CGCCTGTCAGTGGTAAGGAG) cloned in PX461, which co-expresses the Cas9 nickase and GFP. A donor template for homology-directed repair (HDR), which contained the modified ATG start site with the 3 × FLAG tag flanked by two 800-bp homologous arms, was also electroporated at the same time. Recombinant colonies were screened by PCR using the primers TGGGAGAAAGATGCTCTC CAT and CACCGTCATGGTCTTTGTAGTC. Primers TGGGAGAAAGATG CTCTCCAT and CCTCGGCTTGACATAGCCGCC detect the WT unmodified locus but not the *Zbtb11*$^{FLAG}$ allele.

**Generation of the inducible *Zbtb11* KO line.** In the first step, introns 2 and 3 were targeted using the gRNAs TAGGATTAAGGAAAACATTG and GCTTCTAA TACTCTGTGCAA, respectively, cloned in PX458, which co-expresses the WT Cas9 nuclease and GFP. The plasmids encoding the two gRNAs were co-electroporated with an HDR template containing a 2.8-kb region spanning *Zbtb11* exon 3 and flanking intronic sequences modified to include the loxP sequence alongside ectopic restriction sites at the regions targeted by the gRNAs (Fig. 2a). Successfully targeted colonies were detected by PCR (GATAGGGAGCC CTGCTCTCA and CCACAGCCTCCAAGTCTTCC), followed by restriction digest with *Eco*RI and *Bam*HI (Supplementary Fig.2a). Following the isolation of a homozygous *Zbtb11*$^{lox/lox}$ line, the ERt2-Cre transgene was inserted into the *Rosa26* locus using the gRNA ACTCCAGTCTTTCTAGAAGA cloned into PX458, and the pMB80 plasmid (a gift from Tyler Jacks, Addgene plasmid # 12168)[42] as HDR template. Recombinant colonies were detected by nested PCR with the primers GGCCCAAATGTTGCTGGATA and AGAGCCTGGGCTAGGTAGGG in the first step, and AGGTTCTGCGGGAAACCATT and AGGTAGGGGATCGG GACTCT in the second step.

To induce deletion of *Zbtb11* exon 3, cells were treated with 4-hydroxytamoxifen (T176, Sigma-Aldrich) by adding it to the media from a 1000× stock dissolved in EtOH, to a final concentration of 250 nM.

**Genomic and mitochondrial DNA qPCR.** DNA was extracted using the DNeasy Blood and Tissue kit (Qiagen). Twenty nanograms of total DNA was used in a qPCR reaction containing QuantiNova SYBR Green mix (Qiagen), which was ran on a StepOnePlus Real-Time platform (Thermo Fisher Scientific) running StepOne software (version 2.3). To quantify gDNA, the geometric mean of two primer pairs amplifying different genomic loci was used. The geometric mean of two separate mitochondrial loci was used to quantify the mtDNA. The number of functional *Zbtb11*$^{lox}$ alleles remaining in the cell population after KO induction was quantified using an intron 3-specific primer and a primer that specifically recognises the intron 3 loxP site but not the intron 2 loxP site. Cre-mediated recombination disrupts the binding site of the loxP-specific primer and therefore qPCR signal is no longer generated from the KO alleles. The signal for functional *Zbtb11*$^{lox}$ alleles was normalised to the geometric mean of the two gDNA primer pairs.

**Quantitative reverse transcription-PCR.** RNA was extracted using the RNeasy Mini kit (Qiagen). Exogenous ERCC RNA Spike-In Mix 1 (Thermo Fisher Scientific) was added at the cell lysis stage in a ratio of 1 µl per 9 million cells. One microgram of total RNA was used to synthesise cDNA with the ProtoScript II kit (New England Biolabs) using random primers in a 20 µl reaction. The RT-PCR reaction was diluted to 100 µl with ultrapure water, after which 1 µl was used as template in a qPCR reaction. Because initially it was not known whether Zbtb11 depletion affects the expression of housekeeping genes, gene expression in Fig. 3b was normalised to the RNA spike-in mix using primers for ERCC 002 and ERCC 003. Following the whole-transcriptome analyses at 48 and 72 h post *Zbtb11* KO induction, four suitable internal controls were identified (*Hectd1*, *Actb*, *Sel1*,

*March7*) that consistently remained unchanged between replicates and datasets, and which were subsequently used for normalisation in Figs. 3c and 5f.

**Immunoblotting.** Whole-cell lysates were prepared by heating one million cells at 95 °C in 100 µl 1× Laemmli buffer for 5 min. For immunoblots including Mtco1, lysates were prepared in 1× Laemmli buffer supplemented with Complete protease inhibitors (Roche) by heating at 37 °C for 5 min, sonicated for 30 s in a Bioruptor Pico (Diagenode), followed by 5 more minutes of heating at 37 °C. Following separation by sodium dodecyl sulfate (SDS)-polyacrylamide gel electrophoresis, proteins were transferred onto PVDF membranes, which were subsequently blocked with 5% skimmed milk in [TBS, 0.1% Tween-20]. Primary antibodies were used at a concentration of 1 µg/ml: FLAG M2 (F3165, Sigma), Patz1 (sc-292109, Santa Cruz), Lamin B (sc-6216, Santa Cruz), Mrpl48 (14677-1-AP, Proteintech), Ndufc2 (15573-1-AP, Proteintech), Ndufb11 (ab183716, Abcam), Ndufa9 (ab14713, Abcam), Patz1 (sc-292109, Santa Cruz). Mtco1, Ndufb8, Sdhb, Uqcrc2 and Atp5a are part of an OXPHOS antibody cocktail (ab110413, Abcam). Because Mtco1 detection is not as efficient as the rest of the proteins detected with the antibody cocktail[43], the membranes were incubated with additional Mtco1 antibody (ab14705, Abcam). Secondary antibodies were horse radish peroxidase-coupled: anti-mouse (P026002-2, Dako), anti-rabbit (A16029, Thermo Fisher Scientific) and anti-goat (SAB3700285, Sigma). Chemiluminescence substrates used were WesternSure ECL (926-80100, LI-COR) and SuperSignal West Femto (34094, Fisher Scientific).

**Apoptosis measurement.** For intracellular staining of cleaved caspase-3, cells were fixed in 2% formaldehyde, permeabilised in 90% methanol and stored at −20 °C. Following rehydration and blocking in phosphate-buffered saline (PBS) containing 0.5% (w/v) bovine serum albumin, cells were stained for 30 min at room temperature with Alexa Fluor 647-conjugated anti-cleaved caspase-3 (9602, Cell Signaling Technology) and analysed by flow cytometry. The positive control sample was a 1:1 mixture of cells maintained under normal culture conditions and cells heat-shocked at 45 °C for 30 min, followed by 3 h recovery at 37 °C. For detection of phosphatidylserine externalisation, cells were stained with CF647-coupled Annexin V (4300-0325, Millipore) without fixation, at 37 °C.

**Chromatin immunoprecipitation.** Cultured cells were fixed in media containing 1% formaldehyde (Sigma) for 10 min at room temperature. CD4$^+$CD8$^+$ double-positive thymocytes were FACS-sorted from dissected thymi of 6-week-old mice and fixed in DMEM containing 10% foetal calf serum and 1% formaldehyde, for 10 min at room temperature. Following fixation, cells were resuspended in 10 mM HEPES pH 7.5,10 mM EDTA, 0.5 mM EGTA, 0.75% Triton X-100, Complete protease inhibitors, and incubated mixing end over end for 10 min at 4 °C. Nuclei were pelleted by centrifugation, and then resuspended and incubated as before in 10 mM HEPES pH 7.5, 200 mM NaCl, 1 mM EDTA, 0.5 mM EGTA, Complete protease inhibitors. Subsequently, nuclei were resuspended in sonication buffer [150 mM NaCl, 25 mM Tris pH 7.5, 5 mM EDTA, 1% Triton X-100, 0.1% SDS, 0.5% sodium deoxycholate, Complete protease inhibitors (Roche)], incubated for 30 min on ice and sonicated in a Bioruptor Pico (Diagenode) for 10 cycles (30 s on, 30 s off). Insoluble chromatin was pelleted by high-speed centrifugation and the supernatant was incubated with the ChIP antibody overnight mixing end over end at 4 °C. One microgram of antibody was used per million sonicated ESCs. Antibodies used were: FLAG M2 (F3165, Sigma) and Zbtb11 (NBP1-80327, Bio-Techne). Following overnight incubation, antibody/chromatin complexes were pulled down by mixing 3 h at 4 °C with protein G Dynabeads (Thermo Fisher Scientific) using 5 µl beads per µg antibody. Beads were washed three times and the DNA was eluted by shaking the beads overnight at 65 °C in 1× TE, 1% SDS and 0.5 mg/ml proteinase K (Sigma). Eluted DNA was isolated by phenol/chloroform extraction and ethanol precipitation. Purified DNA was quantified using the Quant-iT PicoGreen kit (Thermo Fisher Scientific).

**Sequential ChIP.** Sonicated chromatin lysates were obtained as described above (see Methods: Chromatin immunoprecipitation) and the protein content was measured using the Bio-Rad RC DC protein measurement kit (cat. no. 5000121). Chromatin (500 µg) was mixed with 5 µg anti-Zbtb11 antibody overnight. Antibody/chromatin complexes were pulled down with 25 µl protein G Dynabeads by mixing for 3 h at 4 °C. The beads were washed three times and the immunoprecipitated chromatin was eluted with 50 µl [1× TE, 10 mM dithiothreitol] by shaking 30 min at 37 °C. Eluted chromatin was diluted 1:20 with 150 mM NaCl, 25 mM Tris pH 7.5, 5 mM EDTA, 1% Triton X-100, 0.5% sodium deoxycholate, Complete protease inhibitors (Roche), and mixed overnight with 8 µg anti-GABPa antibody (Santa Cruz, sc-28312 X) pre-captured on 100 µl sheep anti-mouse IgG Dynabeads (by mixing 3 h at 4 °C). Beads were washed three times and DNA was eluted with 230 µl [1× TE, 1% SDS, 1 mg/ml proteinase K, 0.3 mg/ml RNase A] by shaking overnight at 65 °C. DNA was extracted using a Qiagen PCR purification kit, and 1/40 was used in each qPCR reaction.

**ChIP-sequencing.** Libraries were prepared from 10 ng ChIP DNA using the NEBNext Ultra II DNA Library Prep kit for Illumina (E7645, New England Biolabs). Three biological replicates were obtained for FLAG-Zbtb11 ChIP-seq in

*Zbtb11*[FLAG/FLAG] E14 cells, and four biological replicates for Zbtb11 ChIP-seq in *Zbtb11*[wt] E14 cells. Fifty bases single-read sequencing was performed on the Illumina HiSeq 2500 system, obtaining a minimum of 30 million reads for each library. Reads were trimmed and filtered using Cutadapt[44] applying a Phred quality threshold of 25 and a minimal read length of 25 bases. Filtered reads were aligned to the mouse genome (NCBI37/mm9 assembly) with Bowtie 1.1.2 (ref. [45]) keeping only uniquely aligned reads while allowing a maximum of two mismatches. PCR/optical duplicates were removed with Picard (version 2.6.0) and peaks were called using MACS2 (ref. [46]). MACS2 was used as peak caller within the IDR pipeline[47] as implemented by the ENCODE consortium. The final set of peaks was considered the conservative set for IDR < 0.02. From this set we also removed blacklisted sites[48]. Overlap with UCSC gene annotations and statistical analyses were carried out in R using the Bioconductor packages GenomicRanges (version 1.36.1) and GenomicFeatures (version 1.36.4)[49], considering promoter regions as 2 kb upstream and 500 bp downstream of TSS. Normalised read coverage values for consensus peaks were obtained using the DiffBind Bioconductor package (version 2.12.0)[50].

**Motif analyses**. ChIP-seq peaks were centred around the summit and re-sized to 300 bases. Gene promoters were limited to the region −500 to +500 around the TSS. DNA sequences were obtained using the Bioconductor package BSgenome (version 1.52.0). De novo motif search motif matching were performed using the MEME Suite[51].

**RNA-sequencing**. One microgram of total RNA was used as starting material for each library. rRNA was depleted using the NEBNext rRNA Depletion kit (E6310, New England Biolabs) and subsequently directional RNA-seq libraries were prepared using the NEBNext Ultra Directional RNA Library Prep kit for Illumina (E7420, New England Biolabs). Seventy-five bases paired-end sequencing was performed on the Illumina HiSeq 2500 system, obtaining a minimum of 31 million reads for each library. Adaptor sequences and bases with a Phred quality score lower than 25 were trimmed from read ends with TrimGalore[44], and polyA tails were removed using PRINSEQ[52]. Filtered reads were aligned to the mouse transcriptome with STAR[53] using two-pass mapping, and gene-level counts were generated with HTSeq[54]. DGE analyses were performed using the Bioconductor package edgeR (generalised linear model approach, quasi-likelihood *F* test)[55].

*DGE analysis*. Libraries were prepared from three biological replicates for each of the three sample groups. When all three replicates were used to compare 4OHT- to EtOH-treated *Zbtb11*[lox/lox] *R26*[ERt2-Cre] cells, 110 genes were identified as DE (FDR < 0.05), 103 of which were downregulated. Multi-dimensional scaling analysis revealed that two of the three biological replicates were more similar to one another than the remaining third replicate (Supplementary Fig. 8a). The fact that this was observed for all sample groups in the same replicate suggested that these differences may be underpinned by technical variability. Removing the distant replicate decreased the dispersion across the entire gene expression range and the common biological coefficient of variation (BCV) value dropped from 17.2 to 6.7% (Supplementary Fig. 8b). Repeating the DGE analysis without the distant replicate, 109 of the 110 DE genes found with the 3 replicates were detected, plus an additional 45 DE genes, 43 of which were downregulated (FDR < 0.05). The effect size at these additional DE genes was smaller (median absolute fold-change 1.84 vs. 2.92, *p* = 1.063e−13, Supplementary Fig. 8c), which is why they were only detected when the BCV is reduced. However, 44 of these genes were found to have very strong Zbtb11 binding sites at their promoters with enrichment values indistinguishable from the binding sites found at the promoters of the DE genes originally detected (*p* value = 0.974, Supplementary Fig. 8d). This suggested that the additional DE genes identified were indeed genuine Zbtb11 targets. Moreover, functional annotation term enrichment analysis with either the restricted (110) or extended (154) list of DE genes gave the same result, the cluster of terms related to mitochondria being the only significant result (FDR < 0.05), with similar enrichment scores of 6.28 and 6.12, respectively. This indicated that although the extended list of DE genes contained 40% more genes, the additional ones were associated with the same functional annotation terms, among which terms related to mitochondria are enriched. We therefore considered the total of 154 DE genes to be the final list of deregulated genes 48 h post KO induction (Supplementary Data).

**Pathway mapping and functional enrichment analysis**. The Database for Annotation, Visualisation and Integrated Discovery (DAVID)[56,57] was used to obtain enriched annotation terms in comparison to the complete list of mouse genes (https://david.ncifcrf.gov/). Pathway mapping was carried out using the Reactome database (version 65) (https://reactome.org/).

**TMRE staining**. Cells were trypsinised from maintenance cultures and aliquoted in 12-well plates at 0.5 million cells per well in 0.5 ml media. A serial dilution of FCCP was prepared in media containing 400 nM TMRE (2×), and 0.5 ml of these dilutions was added to the cell suspension for a final concentration of 200 nM TMRE and 0, 1, 2, 4 and 8 μM FCCP. Cells were returned to the incubator for 20 min, after which they were pelleted by low-speed centrifugation, washed once and resuspended in [PBS, 2% foetal bovine serum] for FACS analysis.

**Seahorse assay**. Seahorse assay plates were coated with 0.2% gelatin and dried overnight in the cell culture incubator. One hundred thousand cells were plated in each well the same day of the assay and allowed to attach for 5 h in growth media in the presence of 5 μM ROCK inhibitor (13624, Cell Signaling Technology). Subsequently growth media were replaced with Seahorse base media (102353-100, Agilent) containing 2 mM sodium pyruvate and 10 mM glucose, and plates were incubated for an hour at 37 °C and atmospheric $CO_2$ concentration. The Seahorse MitoStress kit (103015-100, Agilent) was used to prepare dilutions of the mitochondrial inhibitors, which had the following final concentrations: 1 μM oligomycin, 0.5 μM FCCP, 0.5 μM rotenone and antimycin A. Oxygen consumption was measured using a Seahorse XF24 Analyser with the following programme: mix 4 min, delay 2 min and measure 3 min.

*Normalisation*. Control and *Zbtb11* KO cells from each biological replicate were always assayed on the same plate. For normalisation between wells on the same plate, the amount of gDNA was used as a proxy for cell number. At the end of the Seahorse assay, media were replaced with 200 μl cell lysis buffer (25 mM Tris pH 7.5, 150 mM NaCl, 5 mM EDTA, 1% Triton X-100, 0.2% SDS, 0.5% deoxycholate, 0.25 mg/ml proteinase K), plates were sealed with adhesive film and incubated overnight at 37 °C. Plates were briefly centrifuged, and lysates were homogenised by pipetting, followed by 1:10 dilution with ultrapure water. Diluted lysates were transferred to PCR plates and sealed. Proteinase K was inactivated by incubating the diluted lysates at 95 °C for 15 min, after which 2 μl were used in a qPCR reaction. The mean qPCR signal from two different genomic primer pairs (see Supplementary information for primer sequences) was used to obtain a size factor for each well.

To average OCRs from all biological replicates, an additional plate-wide adjustment factor was applied. This was calculated based on the expectation that overall, control cells on different plates should have similar maximal OCR after correcting for cell number. Following normalisation to gDNA, maximal respiration values were averaged across all control samples, and subsequently used to calculate an adjustment factor relative to the inter-replicate mean.

*Parameter calculation*. Following normalisation, mitochondrial respiration parameters were calculated as defined previously[24]: non-mitochondrial respiration (respiration detected after complete inhibition of mitochondrial respiration with rotenone and antimycin A); basal respiration (respiration measured before the addition of any mitochondrial poisons, minus non-mitochondrial respiration); leak respiration (respiration measured after inhibition of ATP synthase with oligomycin, minus non-mitochondrial respiration); ATP production (basal respiration minus leak and non-mitochondrial respiration); maximal respiration (the maximal respiration measured after treating cells with a pre-determined optimal FCCP amount (0.5 μM), minus non-mitochondrial respiration); spare capacity (maximal respiration minus basal respiration); coupling efficiency (ATP production as fraction of basal respiration). Statistical tests were performed using the R package *stats*.

**Mitochondria isolation**. Mitochondria were isolated as described[58]. Twenty-five million cells were lysed in 1 ml hypotonic buffer (5 mM MOPS pH 7.25, 180 mM KCl, 1 mM EDTA, EDTA-free protease inhibitors) using a Dounce homogeniser. Nuclei and debris were removed by centrifugation at $1000 \times g$, and the supernatant was subsequently centrifuged at $10,000 \times g$ to pellet mitochondria. Mitochondria were resuspended in the same hypotonic buffer, and following total protein measurement, 50 μg aliquots were pelleted by centrifugation at $10,000 \times g$ and snap-frozen.

**Blue native-polyacrylamide gel electrophoresis**. Fifty micrograms of mitochondria were solubilised in 10 mM bis-Tris pH 7.0, 50 mM 6-aminohexanoic acid, 2% digitonin and incubated 20 min on ice. Insoluble material was removed by centrifugation and Coomassie Brilliant Blue G-250 was added to a final concentration of 0.6%. Gel and electrophoresis buffers used have been previously described[59]. Native complexes were separated for 30 min at 150 V using blue cathode buffer (15 mM bis-Tris pH 7.0, 50 mM tricine, 0.02% (w/v) Comassie Brilliant Blue G-250), after which the blue cathode buffer was replaced with clear cathode buffer (15 mM bis-Tris pH 7.0, 50 mM tricine) and complexes were further separated for 2.5 h at 250 V. Following BN-PAGE, native gels were equilibrated in transfer buffer (48 mM Tris, 39 mM glycine, 0.03% SDS, 20% methanol) for 10 min and subsequently transferred onto PVDF membrane for 1.5 h at 350 mA using a wet transfer system. Blocking and detection was performed as described above.

**Spectrophotometric measurements of complex I**. Fifty micrograms of mitochondria were solubilised in 10 mM bis-Tris pH 7.0, 2% digitonin and incubated 20 min on ice. Twenty-five micrograms of mitochondrial extract was aliquoted in each well of a microtiter plate, and subsequently 200 μl complex I substrate solution [2 mM Tris-HCl pH 7.4, 0.1 mg/ml NADH, 2.5 mg/ml nitro blue tetrazolium chloride] was added to all wells simultaneously with a multichannel pipette, followed by mixing and incubation at room temperature for 2 min. The activity of complex I was quantified by measuring the absorption at 620 nm in a plate reader using the SkanIt software (version 5.0).

**Reporting summary**. Further information on research design is available in the Nature Research Reporting Summary linked to this article.

## Data availability

All high-throughput sequencing data generated during this study have been deposited in the Gene Expression Omnibus (GEO) repository with the accession code GSE125047. Additional published datasets that were used in this study are publicly available from GEO, as follows: GABPa ChIP-seq (GSE116704); NRF-1 ChIP-seq (GSE67867). These data are featured in Figs. 1, 3a, c, d, 4a–d and Supplementary Fig. 1. Databases used were as follows: Reactome, DAVID, GTEx. Source data are provided with this paper.

## Code availability

The code used in this study can be made available on request.

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

## Acknowledgements
We thank our colleagues Marika Charalambous, Reiner Schulz, Cameron Osborne and Ferdinand von Meyenn for discussions and constructive comments. We would also like to thank Matthew Vander Heiden (MIT) and Lucas Sullivan (Fred Hutch) for experimental advice and discussions. We would also like to thank staff at Guy's and St. Thomas' NIHR Biomedical Research Centre Genomics and Flow Cytometry facilities for technical assistance, as well as staff maintaining the Rosalind High Performance Compute Cluster. V.C.S., B.C.W. and A.A. were supported by a Medical Research Council (MRC) Career Development Award (MR/M009343/1). L.B. was supported by the King's Bioscience Institute and the Guy's and St. Thomas' Charity Prize Ph.D. Programme in Biomedical and Translational Science (MAJ110901). A.H. holds a Medical Research Council (MRC) eMedLab Medical Bioinformatics Career Development Fellowship, funded from award MR/L016311/1.

## Author contributions
Conceptualisation, V.C.S; methodology, V.C.S; investigation, B.C.W., L.B., A.A. and V.C.S; software, A.H., T.S.C. and V.C.S.; supervision, R.J.O. and V.C.S; funding acquisition, V.C.S.; writing, V.C.S.

## Competing interests
The authors declare no competing interests.
