## [Peer Review File · Nature Communications]

Reviewers' comments:

Reviewer #1 (Remarks to the Author):

This work provides data that suggest that a zinc-finger transcriptional activator, *Zbtb11*, mutated in two families with intellectual disability, co-operates with NRF2/GABPa, and its controlled ablation, which is constitutively embryonic lethal, is associated with decrease of some subunits of complex I, particularly *Ndufc2*, decreased respiration, and reduced amount of some mitochondrial riboproteins, particularly MRPL48. As the Authors report, *Zbtb11* has very many potential binding sites, indicating a substantial pleiotropic effect on transcription. Nevertheless, their bioinformatic analysis indicates that the highest "hits" scored by *Zbtb11* are, at least in part related with mitochondrial metabolism and bioenergetics. The results presented to support this *in silico* result are based on protein and biochemical analysis of respiratory chain in cells in which *Zbtb11* expression is switched off by tamoxifen. The first result presented in the experimental section based on CHIP analysis is a potential co-operation with GABPa, an essential component of NRF2, since *Zbtb11* and GABPa appear to bind

simultaneously to shared genomic sites. This effect seems specific, since other similar factors, for instance NRF1, do not respond to *Zbtb11*. The next logical step that would be expected by this result would have been to verify whether NRF2 regulated genes, for instance TFAM, are differentially expressed in the absence or presence of *Zbtb11*. Nevertheless, this result is totally missing in the paper, therefore the idea that the effect of NRF2 may be suppressed, downregulated, or in any case influenced by the absence or by the presence of *Zbtb11* remains an unanswered question. The Authors then show a set of experiments based essentially on Western-blot analysis, that demonstrate that some specific subunits of complex I (particularly *Ndufc* and possibly *Ndufb8*, but neither NDUFA9 nor NDUFB11, figure 6d) or complex I assembly factors (*Ndufaf1*) are decreased in a time-dependent way as long as the expression of *Zbtb11* is abolished. It is unclear to me whether this effect is detected in all mitochondrially related genes that seem to be regulated by *Zbtb11*, and what is the hypothesis proposed to explain why only complex I seems to be affected. In addition to the oxygenographic measurements by Seahorse that show a reduction of oxygen consumption rate (particularly under FCCP), I wonder why the Authors have not measured complex I activity directly, by spectrophotometric assays. IGA is a non-quantitative means and the results shown in figure 6c are of poor quality. According to figure 6 b, BNGE of complex I visualized using an anti-NDUFB8 antibody, is completely absent in *Zbtb11*-less cells, suggesting that the activity of complex I should be nearly zero. Mitochondrial protein synthesis was measured in cells previously treated with chloramphenicol, by measuring *Mtco1*, as the only mtDNA-encoded subunit. I wonder why this was the only protein the Authors considered to test a possible defect of mtDNA-specific translation, and whether a standard experiment of mitochondrial in organello translation could also be performed in order to have a complete analysis of all the mtDNA encoded proteins.

Finally, in my opinion the possible effect of *Zbtb11* on a relatively specific set of genes related to mitochondrial bioenergetics does not necessarily implies that this is the cause of the intellectual disability of the patients. To test this hypothesis, biochemical analysis of the respiratory chain should be performed in the patients and rescue experiments be done by expressing wt *Zbtb11* in supposedly complex I deficient cells derived from the patients.

Reviewer #2 (Remarks to the Author):

Wilson et al analyzed the role of *Zbtb11* by the ablation of this protein in mouse ES cells and found that it regulate the expression (Transcription) of critical structural and assembly factors of complex I as well as proteins of the mitochondrial ribosomes. They show that this regulation is excepted in cooperation with NRF-2/GABP transcription factor.

All the experimental work was done in one cell line E14 derived from an inbred mouse. This call to caution about the generalization of the observations regarding the molecular mechanism of the diseases caused by the *Zbtb11* mutations in humans. This is true since the lack of function of *Zbtb11*

in ES14 cells is dramatic in terms of cell growth and complex I function. This is more relevant because ZBTB11 ChIP-seq data obtained from three different cell lines, K562, HEK293 also found the enrichment in mitoribosome genes but fail to detect the enrichment in complex I genes (PMID 29893856). Wilson manuscript goes much further in experimentally addressing the role of Zbtb11 however, since they only work with a single mouse cell line and the previous report used data from several human cell lines it may be relevant to determine if the direct implication of complex I genes is specie specific. As it may be the case of the implication of P53 in zebrafish but not in E14 ES mouse lines as explained in the manuscript.

By the same token, the experiment of the inducible elimination of Zbtb11 is clever but it lacks on of the necessary control a 4OHT inducible ERT2-Cre cell line with Zbtb11 Wt (no-foxed) to discard that the expression of Cre does not induce any phenotypic consequence. The authors discard that 4OHT induce any significant phenotype but not that Crea could do it.

The lost of viability of the cell lines may be due to the mitochondrial problems as proposed by the authors or may be due to affectation of other genes not investigated. As presented the data of mitochondrial function impairment correlate with the viability phenotype but the experimental set up do not fully demonstrate causality. Since cultured cells can survive even in the absence of mtDNA and therefore complexes I, III and IV and incomplete V. A definitive experiment capable to demonstrate that the lack of viability is solely due to the mitochondrial respiratory phenotype would be showing that the cells become viable in medium supplemented with pyruvate and uridine (conditions that allows mtDNA-less cells tu survive. In addition, if they become Pyruvate dependent but uridine independent it will restrict the viability phenotype to complex I deficit since complexes III and IV but no complex I are required to synthesize uridine.

The functional analysis of mitochondria require some improvements:

First, it is surprising that the Basal respiration is decreased while the maximal respiration is still well above the basal and fully enhanced by FCCP. It seems that in figure 5 OCR data are expressed pe minute and not per minute per cell. This results suggest that the amount of cells in the Zbtb11KO is lower that that of the control.

The BNGE in gel activity experiment shown is of very poor quality. the signal if very faint and the labelling of putative CI and cupercomplexes incorrect (does not match with their own imunodetection). It is recommended to perform in gel activity and imunodetection of other complexes (IV, V, II) to properly understand and labelling of the different migration position of the complexes in the gel. Equally the experiments addressing the stability of the complexes need to be performed in BNGE and not in SDS-PAGE as it is down in Fig 7 since the authors themselves indicate that SDS is nor informative of the assembly status.

Reviewer #3 (Remarks to the Author):

The manuscript by Brooke C. Wilson and coauthors investigated molecular functions of ZBTB11, a protein implicated in hereditary intellectual disability. Authors performed extensive molecular analyses to conclude that ZBTB11 facilitates recruitment of Nuclear Respiratory Factor 2 (NRF2) to a subset of its target genes, thereby ensuring optimum expression of nuclear DNA encoded factors of mitochondrial electron transport chain complex I leading to optimum OXPHOS. Authors also concluded that ZBTB11 contributes to a mitoribosome defect, leading to an impairment in mitochondrial protein translation. Although the study is nicely designed and used various appropriate experimentation, there are several conclusions that need better justifications. The following suggestions should improve the manuscript.

1. It is surprising that authors chose a mouse embryonic stem cell line to monitor mitochondrial electron transport chain and oxidative phosphorylation. It is well known that pluripotent stem cells, including the E14 ES cells, utilizes dual mode (Glycolytic and oxidative) of energy production and largely rely on glycolytic energy metabolism. Thus, an experimental system with restricted mitochondrial oxidative capacity is not ideal to test electron transport chain function.
2. With respect to the above point, it is surprising that defective ETC function is leading to such a

drastic effect on ESC proliferation, rather than promoting differentiation to a primed ESC state or EpiSC/Differentiation state.

3. Nrf2 gene knockout mice are viable and Nrf1 knockout mice die in late gestation. These phenotypes do not support the conclusion that loss of ZBTB11-mediated facilitation of their transcriptional activity will induce a drastic effect on cell proliferation and induce cell death.

4. As authors are focusing on impairment in mitochondrial function, they should show by electron microscopy whether loss of ZBTB11 induces drastic effect on reduction in transverse cristae containing mitochondria or inducing mitophagy. Especially, a defect in mitoribosome could lead to mitophagy.

5. The section and evidences supporting effect on mitoribosome is really weak (The heading of that section includes "mitoribosome biogenesis", which is confusing). If there is a general defect in mitoribosome and mitochondrial transcription, all of the ETC complexes, rather than only complex I, will show drastic functional impairment in Zbtb11-KO cells. Authors should show more direct experimentation, like pulse chase in mitochondria or mito-ribo-Tag (people used strategy of tagging the mL62 of the large subunit) to make more definitive conclusions.

6. The data with mutant ZBTB11 is also inconclusive. The extent of defect (less rescue) in mRNA and protein expression are not convincing (Fig. 8 a,b, c). It is more concerning due to the fact that mutant protein levels are almost 50% to the wild type protein. The reduced stoichiometry by itself can cause the reduced transcription (As it is known that transcription efficiency by a transcription factor often depends on its expression level in a cell).

7. Also, the experimental approach with mutants seems indirect. A more conclusive approach is to generate mutations at the endogenous locus using CRISPR/Cas9 (similar to the approach of generating a floxed locus).

8. Earlier studies using HEH293 cell indicated that ZBTB11 is mostly localized within the nucleoli of cell and regulates ribosomal RNA expression. Does ZBTB11 protein has a similar localization preference in ESCs? If so, what is the effect on rRNA biosynthesis and translation of nuclear DNA encoded genes? If not? Does ZBTB11 show cell-type specific localization pattern? In that situation authors should show relevance of this study/conclusion in a human cell type, preferably neuronal cells.

9. The impaired mitochondrial ETC function often leads to prominent effect in response to oxidative stress (probably that aspect is more relevant to the hereditary intellectual disability with mutant ZBTB11). Authors should consider to test this aspect.

Reviewer #4 (Remarks to the Author):

Wilson et al identified a novel function of Zbtb11 as a transcription factor activating a series of nuclear-encoded mitochondrial genes with another activator NRF-2. Technically, the study employed an elegant mix of CRISPR, RNA-seq, ChIP-seq, bioinformatics and mitochondrial-related experiments. The author walked the reader through the genomic approaches whereby they utilized the elegant RosaCreERT2 loxP system to deplete ZBTB11 in mESCs and identified its direct targets as mitochondrial genes. They further observed that ZBTB11 can cooperate with a basic leucine zipper (bZIP) transcription factor NRF2, which is one of the most prominent regulators of nuclear-encoded mitochondrial genes. Their data also suggested that the ZBTB11-NRF2 interaction may be essential for mitochondrial function by maintaining the biogenesis of respiratory complex I. Overall, this is a solid comprehensive work that reveals a mechanism underlying the transcriptional regulation of nuclear-encoded mitochondrial genes by ZBTB11. However, some suggestions need to be considered:

By re-analyzing the high-throughput raw data from the authors, I found most of the sequencing and analysis in the study were well performed. The RNA-seq, ChIP-seq data was consistent between different samples. ChIPseq motif calling showed ZBTB11 and NRF2 have similar binding motifs (CCGGAAG). However, this makes me confused a little bit. Because ZBTB11 is a well-conserved protein with 12 canonical C2H2-ZF motifs, which is supposed to contact 36bp DNA theoretically. There is a big gap between experimental data and theoretical data. This requires further experimental verification, such as in vitro luciferase assay or electrophoretic mobility shift assay. This also led me to

speculate whether there is a possibility that ZBTB11 cannot directly bind to DNA but instead binds to DNA through NRF2. If so, this speculation is contradictory to the reChIP experiments in the manuscript.

The mutations H729Y and H880Q are located at the fifth and the tenth C2H2 zinc finger motifs, respectively. The authors might need to construct those mutant mESCs to examine if the mutate ZBTB11 binding sites have changed or not. This might help to verify if ZBTB11 interacts with DNA directly. In general, I suggest the authors emphasize on the ZBTB11 and DNA interaction part, which should be clarified.

If ZBTB11 does interact with 8bp DNA and the binding is necessary for the recruitment of NRF2, then their interaction should be easily detected by using immunoprecipitation assay. Furthermore, more ZBTB11 interacting proteins will be identified by mass spectrometry, this can be done by using Flag-tagged ZBTB11 mESCs already used in the manuscript.

It seems that ZBTB11 and NRF2 can activate mitochondrial related genes by directly targeting their promoters. If so, we probably can see some changes in histone modifications and/or chromatin accessibility in ZBTB11 KO cells. Please clarify this point and identify the chromatin related factors involved in this event if possible.

Minors:

Page 8 "To this end we performed de novo motif discovery using the Zbtb11 ChIP-seq peakset, and separately the promoter regions of the 154 Zbtb11-dependent genes. Both approaches identified as the top hit the same motif, which closely matched that recognized by the ETS-domain protein GABPa (Fig. 4a)" It is better to show two separate motifs for ZBTB11 ChIP-seq and promoter regions.

FigureS7a, I recommend labeling wt control (red dots) with C1/C2/C3, rather than R1/R2/R3, because they were not ZBTB11 floxed mESCs.

Point-by-point answers

Reviewer #1 (Remarks to the Author):

This work provides data that suggest that a zinc-finger transcriptional activator, Zbtb11, mutated in two families with intellectual disability, co-operates with NRF2/GABPa, and its controlled ablation, which is constitutively embryonic lethal, is associated with decrease of some subunits of complex I, particularly Ndufc2, decreased respiration, and reduced amount of some mitochondrial riboproteins, particularly MRPL48.

As the Authors report, Zbtb11 has very many potential binding sites, indicating a substantial pleiotropic effect on transcription. Nevertheless, their bioinformatic analysis indicates that the highest "hits" scored by Zbtb11 are, at least in part related with mitochondrial metabolism and bioenergetics.

Answer 1.1. There is no reason to believe Zbtb11 has a substantial pleiotropic effect on transcription. We can tell exactly what the roles of Zbtb11 are in the transcriptional program because of the transcriptomics analysis we performed in cells acutely depleted of Zbtb11 - namely to regulate the 154 genes listed in Supplementary Table 1. Even though Zbtb11 binds many other promoters across the genome, it is obviously not essential for transcriptional regulation at all these sites, because when Zbtb11 is depleted many of its target genes remain unchanged.

It is very common that ChIP-seq datasets have thousands of binding sites while only a fraction of those prove to be functional when that factor is depleted. This is thought to be caused by several reasons – low signal peaks can reflect weak indirect interactions that are stabilised by the ChIP fixation process but which are not of critical functional importance; at the same time there is also redundancy among transcription factors co-localising to the same promoters and any one individual factor may be the critical regulator only at a subset of its binding sites depending on the local sequence, transcription factor binding or chromatin context at the promoter. It is therefore important that ChIP-seq analyses are not attributing equal importance to all peaks detected, and that the signal at individual peaks is taken into consideration and their functional role is correlated with changes in expression when the transcription factor is removed. This was the approach we took with Zbtb11, and our integration of ChIP-seq and RNA-seq datasets showed that only the strongest binding sites in the dataset mark the promoters of Zbtb11-dependent genes (Fig. 3d in original manuscript; or Supplementary Fig. 3b in revised manuscript). The new data included in the revised manuscript also shows that binding of Zbtb11 at these promoters is remarkably well conserved across cell types and species (Fig. 3c in revised manuscript), indicating that Zbtb11 has largely invariant housekeeping roles in transcriptional regulation.

The results presented to support this *in silico* result are based on protein and biochemical analysis of respiratory chain in cells in which Zbtb11 expression is switched off by tamoxifen. The first result presented in the experimental section based on CHIP analysis is a potential co-operation with GABPa, an essential component of NRF2, since Zbtb11 and GABPa appear to bind simultaneously to shared genomic sites. This effect seems specific, since other similar factors, for instance NRF1, do not respond to Zbtb11. The next logical step that would be expected by this result would have been to verify whether NRF2 regulated genes, for instance TFAM, are differentially expressed in the absence or presence of Zbtb11. Nevertheless, this result is totally missing in the paper, therefore the idea that the effect of NRF2 may be suppressed, downregulated, or in any case influenced by the absence or by the presence of Zbtb11 remains an unanswered question.

Answer 1.2. These questions are in fact addressed head on by the results depicted in Fig. 4g and Supplementary Fig. 4 (Supplementary Fig. 3 in the original manuscript), only that instead of Tfam we analysed Tfb2m and Cox6a1. We explain in the manuscript that our genomics analyses show Zbtb11 and NRF-2 associate only at a subset of NRF-2 binding sites, and these do not include some of the classical NRF-2 targets such as Tfb2m and Cox6a (first paragraph on page 9; or lines 259-262 in revised manuscript). This differential binding was detected in our ChIP-seq analysis (Supplementary Fig. 3a in original manuscript; or Supplementary Fig. 4b in revised manuscript) and was validated by ChIP-qPCR (Supplementary Fig. 3b in original manuscript; or Supplementary Fig. 4c in revised manuscript). The lack of Zbtb11 binding at these promoters automatically implies it does not directly regulate these targets, and the transcriptomics analysis confirms none of these genes change expression when Zbtb11 is depleted, as they are not among differentially expressed genes (Supplementary Table 1). We are now stating this in the revised manuscript to hopefully make this clear (line 261), and for convenience we have also included the RNA-seq results for these genes below (see Referee Table 1).

Referee Table 1. RNA-seq results for known classical NRF-2 targets

Gene	log2(FoldChange)	Adjusted p-value
Tfam	0.0147	0.99
Tfb1m	-0.123	0.99
Tfb2m	0.129	0.99
Cox6a1	0.061	0.99
Cox6b1	-0.036	0.99

NRF-2 does however co-localise with Zbtb11 at the promoters of other genes where Zbtb11 controls its recruitment, and therefore its activity, at those specific loci. NRF-2 and Zbtb11 associate at the promoters of virtually all Zbtb11-dependent genes, where in fact some of its strongest binding sites are found, and where the binding of these two transcription factors is highly correlated (Fig. 4d). Importantly, when Zbtb11 is depleted, NRF-2 binding is specifically lost from promoters where it co-localises with Zbtb11 but not from Zbtb11-independent promoters (Fig. 4g) - this clearly shows that the recruitment of NRF-2 and its ability to activate transcription at these promoters (shared with Zbtb11) is dependent on

Zbtb11. It is just a different set of promoters from the previously known NRF-2 targets. As the Reviewer mentions, this regulatory relationship is specific to Zbtb11 and NRF-2, as NRF-1 binding is not affected by Zbtb11 depletion at the promoters they share. This strongly indicates that Zbtb11 and NRF-2 specifically cooperate to activate their shared target promoters.

This is an interesting aspect of our results, because it shows NRF-2 recruitment/activity is not uniformly regulated across the genome as previously assumed, but it can be regulated in a locus-specific manner. We make this point in the relevant results section (top paragraph on page 10 in original manuscript; lines 297-298 in revised manuscript) and in the Discussion, which also includes a discussion on possible mechanisms to achieve this locus-specific regulation (third paragraph on page 17 in original manuscript; lines 565-570 in revised manuscript). Moreover, the new ChIP-seq data we have now added shows that Zbtb11 also binds to the GABPa recognition motif in human cells (Fig. 4a), indicating this functional association is conserved between mouse and human.

The Authors then show a set of experiments based essentially on Western-blot analysis, that demonstrate that some specific subunits of complex I (particularly Ndufc and possibly Ndufb8, but neither NDUFA9 nor NDUF11, figure 6d) or complex I assembly factors (Ndufaf1) are decreased in a time-dependent way as long as the expression of Zbtb11 is abolished. It is unclear to me whether this effect is detected in all mitochondrially related genes that seem to be regulated by Zbtb11, and what is the hypothesis proposed to explain why only complex I seems to be affected.

Answer 1.3. None of the other OXPHOS complexes are affected in the same manner as complex I, because Zbtb11 specifically controls complex I biogenesis genes, and does not control to any significant extent genes directly responsible for the biogenesis of other OXPHOS complexes.

We explain this in the relevant results section as well as in discussion (top paragraph on page 18 in original manuscript; lines 575-586 in revised manuscript). We have now added additional explanations in the results section (lines 367-369) and in the Fig. 6 legend, which we hope clarifies our hypothesis and results.

Our experimental approach was initially guided by the RNA-seq experiment that was carried out 48 hours post-KO induction. It was important to perform the RNA-seq experiment on these samples because it is the earliest time point at which Zbtb11 is completely depleted from the chromatin, so this experiment revealed to us which genes are directly controlled by Zbtb11. These are the 154 genes in Supplementary Table 1. At later time points, of course these genes are still deregulated but the results will be confounded by secondary transcriptional changes that take place in response to the deregulation of Zbtb11-controlled genes.

Functional annotation and pathway over-representation analyses showed the Zbtb11-dependent genes are highly enriched in genes with mitochondrial function, and within this category, the pathways that are over-represented were “complex I biogenesis” and “mitochondrial translation” (the genes that map to these two pathways are listed in Fig. 6a and in Supplementary Table 3). There were four genes in total that mapped to the complex I biogenesis pathway, and two of them (*Ndufc2* and *Ndufaf1*) were particularly strongly down-

regulated – they are in fact two of the most down-regulated genes in the entire RNA-seq dataset. For *Ndufc2* we also have immunoblots showing that not only the transcript, but the protein is also downregulated (Fig. 3f in original manuscript; Fig. 3e in revised manuscript). There is an abundance of evidence from studies of human disease (References 34–36) and *in vitro* experimental approaches (Reference 37) that show inactivating either one of these two genes individually leads to severe defects in complex I assembly, so their combined down-regulation in *Zbtb11* KO cells was predicted to cause a strong complex I biogenesis defect. Except for a mild down-regulation of *Atpaf1*, a complex V assembly factor, no other OXPHOS complexes biogenesis factors are directly controlled by *Zbtb11*.

The immunoblot in Fig. 6b aimed to establish whether the deregulation of the factors that we are detecting by RNA-seq affects the biogenesis of the respiratory complexes. The immunoblot uses a cocktail of antibodies that detect individual OXPHOS subunits that function as stability markers, i.e. are degraded if their respective holocomplex is unstable. Importantly, none of these stability markers are under direct *Zbtb11* control, i.e. their transcripts are not deregulated in *Zbtb11* KO cells, including the complex I stability marker *Ndufb8*.

In the end the results of the immunoblots in Fig. 6b validate the predictions made by the RNA-seq results – namely that complex I biogenesis is disrupted in *Zbtb11* KO cells. The complex I stability marker *Ndufb8* is down-regulated as a consequence of the fact that complex I holocomplex is no longer synthesised or is unstable due to the down-regulation of the *Zbtb11*-controlled factors *Ndufc2*, *Ndufaf1*, *Ndufa12* and *Ndufs7*.

The complex V assembly factor *Atpaf1* was probably not deregulated strongly enough to significantly disturb the synthesis of complex V and affect the expression of the complex V stability marker *Atp5a*. Also, being the only factor deregulated in this pathway, there may be compensatory mechanisms that kick in, while in the case of complex I biogenesis these would have difficulties compensating for the down-regulation of four different factors. At the same time the experiments in Fig. 7a show that complex V is turned over at a very slow rate, so even significant down-regulations of complex V genes would likely take a long time to affect the amounts of complex V subunits.

In addition to the oxygraphic measurements by Seahorse that show a reduction of oxygen consumption rate (particularly under FCCP), I wonder why the Authors have not measured complex I activity directly, by spectrophotometric assays. IGA is a non-quantitative means and the results shown in figure 6c are of poor quality. According to figure 6 b, BNGE of complex I visualized using an anti-NDUFB8 antibody, is completely absent in *Zbtb11*-less cells, suggesting that the activity of complex I should be nearly zero.

Answer 1.4. We thank the reviewer for this suggestion. We have now replaced the in-gel activity assay with spectrophotometric measurements of complex I activity performed on mitochondrial extracts from control and *Zbtb11* KO cells (Fig. 6e in revised manuscript). We measured complex I activity by quantifying the reduction of nitro blue tetrazolium in mitochondrial extracts supplemented with NADH (see also Methods in revised manuscript). This data confirms the loss of complex I activity following *Zbtb11* depletion.

Mitochondrial protein synthesis was measured in cells previously treated with chloramphenicol, by measuring Mtco1, as the only mtDNA-encoded subunit. I wonder why this was the only protein the Authors considered to test a possible defect of mtDNA-specific translation, and whether a standard experiment of mitochondrial in organello translation could also be performed in order to have a complete analysis of all the mtDNA encoded proteins.

Answer 1.5. To clarify, the experiments in Fig. 7 are not aiming to measure mitochondrial protein synthesis directly, but they are monitoring the amount of OXPHOS holocomplexes being synthesised, by using the same cocktail of antibodies to stability markers that we used in Fig. 6b (explained above in Answer 1.3). Mtco1 is the stability marker for complex IV, and it just happens to be the only one out of the five that is mitochondria-encoded, so it would be expected to provide a more direct measure of the resumption of mitochondrial protein synthesis. This detail in itself is not important for the experiments, and we can now see how the reference to this fact in the text can be confusing - we have therefore now removed it.

We did the experiments in Fig. 7 because we were interested in finding out how the dynamics of mitochondrial translation affects the steady state amounts of respiratory complexes detected in Fig. 6b. The reason behind this was the observation that complexes III and IV also appeared to be down-regulated, but after more prolonged Zbtb11 depletion (at 120h post-KO induction, the latest we can assay Zbtb11 KO cells before they die), and also to a lesser extent than complex I (Fig. 6b). While Zbtb11 controls several complex I biogenesis and mitoribosomal protein genes (Fig. 6a), it does not directly regulate any complex III and IV biogenesis genes, so we reasoned that the down-regulation of complex III and IV may be the result of mitochondrial translation defects. A severe mitochondrial translation defect would also affect complex I biogenesis, as this complex contains mitochondria-encoded subunits, so this raised the question to what extent the complex I biogenesis defect we observed is caused by an assembly defect (see Answer 1.3) versus a mitochondrial translation defect.

Because the down-regulation of complexes III and IV is manifested 48 hours after that of complex I, for all three complexes to be down-regulated as a result of a mitochondrial translation block, the turnover rate of complexes III and IV would have to be significantly slower than that of complex I. To get an idea about the rate at which OXPHOS complexes are turned over, we performed the experiment in Fig. 7a in which we measured the respiratory complexes stability markers following mitochondrial translation inhibition with chloramphenicol. The results showed us that when mitochondrial translation is inhibited, complexes I, III and IV decay rapidly, while complex V seemed remarkably slow to turn over (complex II has no mitochondria-encoded subunits, and it was therefore not expected to be affected by the inhibition of mitochondrial translation). The turnover of complex I was faster than that of complexes III and IV, but not to the extent of being able to explain the difference between complex I and complex III/IV down-regulation in Zbtb11 KO cells.

Because the mitoribosome inhibition by chloramphenicol is reversible, removal of chloramphenicol allows mitochondrial translation to resume, and respiratory holocomplexes are subsequently gradually regenerated as can be seen in Fig. 7c (control samples). This ability to control depletion and synthesis of respiratory complexes I, III and IV, also provides a straightforward assay to determine whether the biogenesis of these three complexes is impaired

to the same extent in *Zbtb11* KO. By synchronising the *Zbtb11* KO induction with the mitochondrial translation block and release, we were able to monitor how the synthesis of respiratory complexes I, III and IV is affected in cells depleted of Zbtb11. Because the synthesis of these complexes is completely dependent on new mitochondrial protein synthesis, deficient mitochondrial translation in *Zbtb11* KO cells would be expected to affect all three complexes to a similar extent. However, the results showed that while the synthesis of complex I was completely blocked, complexes III and IV were still synthesised, albeit to a lesser extent (Fig. 7c).

These results therefore reinforce our conclusion that complex I biogenesis is primarily disrupted in *Zbtb11* KO cells by the down-regulation of complex I-specific factors Ndufc2, Ndufaf1, Ndufa12 and Ndufs7. This is the key message we are trying to convey through these experiments. Our results are consistent with a partial impairment of mitochondrial translation, but this is probably mild/intermediate at most, and our experiments do not interrogate the efficiency of mitochondrial translation directly. For this reason our intention is that these experiments are viewed as “controls” that emphasise the fact the disruption of complex I assembly is what primarily underpins the down-regulation of complex I and loss of its activity. We have now made changes to the results section which we hope will better communicate this message. To avoid suggesting that our results show that Zbtb11 controls complex I biogenesis and mitochondrial translation to an equivalent degree we have also removed the reference to mitochondrial translation from the subheading of this results section, which is now entitled “Zbtb11 controls complex I biogenesis”.

Finally, in my opinion the possible effect of Zbtb11 on a relatively specific set of genes related to mitochondrial bioenergetics does not necessarily implies that this is the cause of the intellectual disability of the patients. To test this hypothesis, biochemical analysis of the respiratory chain should be performed in the patients and rescue experiments be done by expressing wt Zbtb11 in supposedly copmplex I deficient cellks derived from the patients.

Answer 1.6. Our study is primarily focused on determining the cellular and molecular functions of Zbtb11, as these are currently unknown, and we do not claim to provide a comprehensive mechanism for the aetiology of *ZBTB11*-associated intellectual disability. However, to understand what processes are disrupted when Zbtb11 is mutated, it is important to determine what the cellular functions of this factor are, and we believe our study is an important contribution towards this goal. In addition, we provide evidence that the human pathogenic mutations destabilise the protein to a significant extent, reducing its dosage and therefore revealing a molecular mechanism directly relevant for understanding the disease aetiology.

We believe our study on the cellular functions of Zbtb11 is particularly relevant because this factor is essentially a housekeeping protein that regulates other housekeeping genes. Zbtb11 is ubiquitously expressed, and is listed as a housekeeping protein by the Human Protein Atlas (<https://www.proteinatlas.org/humanproteome/tissue/housekeeping>). Zbtb11 is essential in a wide range of cell types, controls virtually no cell type-specific genes, and its targets are

strongly conserved across cell types and species. Altogether these indicate that the disease caused by human *ZBTB11* mutations most likely results from a partial impairment of fundamental cellular processes, and our experiments strongly indicate that mitochondrial dysfunction is an important part of that.

The evidence in support of *ZBTB11*-associated intellectual disability being, at least in part, a mitochondrial disorder are as follows:

- *Zbtb11* functionally associates with NRF-2/GABP, a known regulator of mitochondrial genes, and the genes it regulates are highly enriched in genes with mitochondrial functions.
- While *Zbtb11* also controls non-mitochondrial genes, these are not clustered to any significant extent in specific pathways or cellular compartments, so *Zbtb11* dysfunction is expected to disproportionately affect mitochondrial functions.
- Our results unequivocally show that at least respiratory complex I biogenesis is completely dependent on *Zbtb11*.
- Complex I biogenesis appears to be very sensitive to *Zbtb11* dosage, as destabilisation of *Zbtb11* protein by the human pathogenic mutations led to a down-regulation in the amount of complex I holocomplexes.
- There is a clear rationale for how mitochondrial dysfunction could cause intellectual disability, as outlined in the last paragraph of the Discussion
- Besides intellectual disability, there is additional phenotypical overlap between patients with *ZBTB11* mutations and other mitochondrial disorders - cerebellar atrophy and neuromuscular defects (ataxia, hypotonia) are commonly found in patients with mitochondrial DNA mutations.

We therefore find it considerably more likely than not that the disease is at least in part underpinned by mitochondrial dysfunction. This message should be of interest to scientists studying mitochondrial diseases as well as those interested in transcriptional regulation of mitochondrial activity.

Reviewer #2 (Remarks to the Author):

Wilson et al analyzed the role of *Zbtb11* by the ablation of this protein in mouse ES cells and found that it regulate the expression (Transcription) of critical structural and assembly factors of complex I as well as proteins of the mitochondrial ribosomes. They show that this regulation is excepted in cooperation wit NFRF-2/GABP transcription factor. All the experimental work was done in one cell line E14 derived from an inbred mouse. This call to caution about the generalization of the observations regarding the molecular mechanism of the diseases caused by the *Zbtb11* mutations in humans. This is true since the lack of function of *Zbtb11* in ES14 cells is dramatic in terms of cell growth and complex I function. This is more relevant because *ZBTB11* ChIP-seq data obtained from three different cell lines, K562, HEK293 also found the enrichment in mitoribosome genes but fail to detect the enrichment in complex I genes (PMID 29893856).

Answer 2.1. This comment overlaps to some degree with one from Reviewer #1, so please see also Answer 1.6 above.

To clarify, as mentioned in Answer 1.6, our study was primarily focused on determining the cellular and molecular functions of *Zbtb11*, which are currently unknown, and we do not claim to provide a comprehensive mechanism for the aetiology of *ZBTB11*-associated intellectual disability. However, we do believe that our experiments provide valuable insights highly relevant for understanding the aetiology of this disease.

Zbtb11 is ubiquitously expressed, is listed as a housekeeping protein by the Human Protein Atlas (<https://www.proteinatlas.org/humanproteome/tissue/housekeeping>), and is essential in several cell types. Besides our study in ES cells, *Zbtb11* was also found to be essential in two separate lethality screens performed in three different human cell lines (KBM7, HAP1 and K562)^{1,2}, indicating that *Zbtb11* plays fundamental cellular roles that are conserved between cell types and between mouse and human. Consistent with this, we found that the main role of *Zbtb11* is to activate a select subset of housekeeping genes, and the additional data now included in the revised manuscript shows these targets are remarkably well conserved between cell types and between mouse and human (Fig. 3c in revised manuscript). This strongly argues that the disease caused by *Zbtb11* deficiency is the result of a partial impairment of fundamental cellular processes as opposed to disruption of cell type-specific mechanisms, and therefore a wide range of cell types would make valid experimental systems. The ChIP-seq experiments in HEK293 cells and thymocytes that are now included in the revised manuscript were in fact performed early on in our study, and they emphasised that *Zbtb11* is characterised by a high degree of functional conservation. Given this, and the fact that the essentiality of *Zbtb11* prevented us to obtain constitutive KO cell lines, we chose mouse ES cells as an experimental model because they make possible the multiple homologous recombination events required to construct the inducible KO line. Without this complex genetic system, it would have been impossible to gain the same insights into the functions of *Zbtb11*.

The fundamental nature of the mechanisms we have uncovered confirms the functions of *Zbtb11* are not cell type-specific. In support of this, the analysis we present in Fig. 8c shows there is strong correlation of expression in human primary tissues between *Zbtb11* and the targets we identify in mouse ES cells. The analysis in Fig. 8c is particularly powerful because it is based on the entire GTEx dataset comprised of transcriptomics data from 52 different primary human tissues (instead of immortalized cell lines), collected from 948 different individuals. It shows that *ZBTB11* and the 154 targets we identified in mouse ES cells, while ubiquitously expressed, they also have slight differences of expression in different tissues, and these are highly correlated. This type of correlation of co-expression is widely considered strongly indicative of a regulatory relationship, and is commonly used to establish regulatory networks in systems biology approaches such as weighted gene co-expression network analysis³. The high level of correlation between *Zbtb11* and its targets therefore strongly indicates that the regulatory relationship we identified in mouse ES cells, is conserved in humans and applies to a large number of tissues.

Together with the additional data, we have now made changes to the manuscript to emphasise and hopefully better communicate that *Zbtb11* controls fundamental mechanisms of universal importance across cell types, not only for ES cells. We have also added in the Introduction the

information on the genetic screens conducted in human cells that have identified *ZBTB11* as an essential gene.

Answer 2.2. It is not entirely clear why the Reviewer thinks the severity of complex I and cell growth phenotype in *Zbtb11* KO ES cells would call into question the relevance of our study to the aetiology of the disease - is the Reviewer assuming that the human *ZBTB11* mutations are complete loss-of-function? The KO cells are not meant to model the disease, because the human *ZBTB11* mutations are not complete loss-of-function. As mentioned above *Zbtb11* is essential, and our finding that constitutive KO ES cells are not viable indicates a complete *ZBTB11* KO is going to be embryonic lethal and not compatible with the development of an organism. Consistent with this, the human *ZBTB11* mutations are missense single amino acid changes, and complete loss-of-function variants of *ZBTB11* gene are never found as homozygous in the human variant database gnomAD¹. The experiments in *Zbtb11* KO cells were not intended to model the disease, but to help us determine the cellular functions of *Zbtb11*, as these were completely unknown and it is difficult to understand what processes are disrupted in patients if we do not know what *Zbtb11* actually does. On the other hand, the rescue experiments in Fig. 8 interrogated the effect of the pathogenic mutations on the activity of *Zbtb11* and our results showed they do not completely block *Zbtb11* activity, but instead lead to an intermediate level of activation of *Zbtb11* target genes, and an intermediate level of complex I biogenesis.

Answer 2.3. It is important to clarify that in the study cited by the Reviewer (PMID 29893856), Fattahi et al ⁴ do not conduct any functional experiments in which *Zbtb11* is depleted, knocked down, or knocked out, so they did not know exactly what genes are regulated by *Zbtb11*. The only data they use in order to assign target genes to *Zbtb11* is ChIP-seq data generated by the Encode consortium, but they had no way of knowing at which binding site *Zbtb11* is actually essential for transcriptional regulation, because they did not measure transcriptional changes in mutant or *Zbtb11*-depleted cells. Furthermore, their analysis did not stratify or rank the binding sites based on the ChIP-seq signal strength, so genes with the weakest *Zbtb11* peaks (more likely to be false positives) were given equal weighting as genes with the strongest *Zbtb11* binding (more likely to be real functional targets). The direct consequence of this is that the enrichment of real functional targets ends up being diluted by false positive targets or targets where *Zbtb11* binding is not critical, especially ones that belong to annotation categories with more numerous genes.

By contrast, our approach for the enrichment analysis stratified ChIP-seq peaks based on their strength, and integration with the RNA-seq data confirmed the validity of this approach, as transcriptional changes following *Zbtb11* depletion took place at genes with the strongest *Zbtb11* binding sites. Strength of *Zbtb11* binding is therefore a good predictor of whether binding of *Zbtb11* is critical for the regulation of the promoter or not. As we are now showing

¹ https://gnomad.broadinstitute.org/gene/ENSG00000066422?dataset=gnomad_r2_1

in the revised manuscript, the binding of Zbtb11 to its targets is remarkably well conserved between mouse ES cells, thymocytes or human HEK293 cells, both in terms of location and target preference (Fig. 1g-h), and particularly at the promoters of the Zbtb11-dependent genes we identified (Fig. 3c). Importantly, applying the same functional enrichment analysis to the human HEK293 ChIP-seq data, we obtain similar results to the mouse ES cell analysis (Supplementary Fig. 1g).

Wilson manuscript goes much further in experimentally addressing the role of Zbtb11 however, since they only work with a single mouse cell line and the previous report used data from several human cell lines it may be relevant to determine if the direct implication of complex I genes is specie specific. As it may be the case of the implication of P53 in zebrafish but not in E14 ES mouse lines as explained in the manuscript.

Answer 2.4. As mentioned above in Answer 2.3, strength of Zbtb11 ChIP-seq peaks is a good predictor of whether binding of Zbtb11 at a particular promoter is critical for its regulation or not, and the binding preference of Zbtb11 is very well conserved across cell type and species, particularly at the promoters of Zbtb11-dependent genes (Fig. 3c in revised manuscript). These results therefore indicate that the majority of genes that we identified as Zbtb11-dependent in mouse ES cells, will also be under direct Zbtb11 control in human cells, including complex I genes. In support of this, as shown in Fig. 8c (and discussed above in Answer 2.1), there is strong correlation of expression in human primary tissue between Zbtb11 and the targets we identify.

To highlight this conservation at the Zbtb11-dependent complex I biogenesis genes in particular (Ndufc2, Ndufaf1, Ndufa12 and Ndufs7), below we generated the Referee Fig. 1. It shows that complex I genes have some of the strongest Zbtb11 binding sites in the dataset in both mouse ES cells and human HEK293 cells (Referee Fig. 1a), and also that our results are reproduced by independently generated Zbtb11 ChIP-seq datasets obtained by Encode in different human cells (same ones used by Fattahi et al ⁴) (Referee Fig. 1b). Additional loci can be visualised in UCSC Browser sessions, by following these links:

https://genome.ucsc.edu/s/Vladseitan/WilsonB_SeitanVC_May2020_Mouse

https://genome.ucsc.edu/s/Vladseitan/hg19_WilsonB_SeitanVC_May2020_Human

By the same token, the experiment of the inducible elimination of Zbtb11 is clever but it lacks on of the necessary control a 4OHT inducible ERT2-Cre cell line with Zbtb11 Wt (no-foxed) to discard that the expression of Cre does not induce any phenotypic consequence. The authors discard that 4OHT induce any significant phenotype but not that Crea could do it.

Answer 2.5. Our rescue experiments clearly show the observed transcriptional changes are effectively reverted by ectopically expressing *Zbtb11* in *Zbtb11* KO cells (figures 3c (3b in revised manuscript) and 8a). The effects therefore have to be specific because these cells also express ERT2-Cre. There is no other possible interpretation of our results.

The expression of all differentially expressed genes that we measured by qRT-PCR was effectively rescued by ectopic expression of wild type *Zbtb11* in *Zbtb11* KO cells. In total, a tenth of all differentially expressed genes were assayed in this way and they were all effectively rescued (see Fig. 3b and Fig. 8a). Importantly, we assayed every single differentially expressed gene that maps to complex I and mitoribosome biogenesis (Fig. 8a) - pathways on which we focus our functional studies – and showed that all of them were effectively rescued. Because the rescued *Zbtb11* KO cells also express ERT2-Cre, we can be absolutely confident, the deregulation of these genes is not an artefact of Cre expression.

Moreover, integration of transcriptomics and ChIP-seq data showed that the promoters of all differentially expressed genes are strongly bound by *Zbtb11* (Fig. 3d – now Supplementary Fig. 3b), consistent with a direct role for *Zbtb11* in their regulation, and the changes in their expression being specific to *Zbtb11* depletion.

In conclusion, there is very little doubt about the specificity of our results.

The lost of viability of the cell lines may be due to the mitochondrial problems as proposed by the authors or may be due to affection of other genes not investigated. As presented the data of mitochondrial function impairment correlate with the viability phenotype but the experimental set up do not fully demonstrate causality. Since cultured cells can survive even in the absence of mtDNA and therefore complexes I, III and IV and incomplete V. A definitive experiment capable to demonstrate that the lack of viability is solely due to the mitochondrial respiratory phenotype would be showing that the cells become viable in medium supplemented with pyruvate and uridine (conditions that allows mtDNA-less cells tu survive. In addition, if they become Pyruvate dependent but uridine independent it will restrict the viability phenotype to complex I deficit since complexes III and IV but no complex I are required to synthesize uridine.

Answer 2.6. We thank the Reviewer for this suggestion. We have now included this experiment in the revised manuscript (Fig. 6f). Because complex I is the only severely affected complex in the electron transport chain (Fig. 6b) we attempted to rescue cell proliferation by providing pyruvate or aspartate (10mM) and measured proliferation as originally described by Sullivan et al ⁵. As can be seen in Fig. 6f, at 72h post-KO induction when complex I activity is not yet severely disrupted (Fig. 6e), pyruvate and aspartate addition did not change proliferation. However, at 96h, when complex I activity in *Zbtb11* KO mitochondria is less than half of controls (Fig. 6e), addition of pyruvate and aspartate can partially rescue the proliferation. This indicates that complex I deficiency contributes to a significant extent to the block in proliferation.

**The functional analysis of mitochondria require some improvements:
First, it is surprising that the Basal respiration is decreased while the maximal respiration is**

still well above the basal and fully enhanced by FCCP. It seems that in figure 5 OCR data are expressed per minute and not per minute per cell. This results suggest that the amount of cells in the *Zbtb11*KO is lower than that of the control.

Answer 2.7 As explained in the Methods section the oxygen consumption rate (OCR) data was controlled and corrected for cell number. We had methodologies to ensure that, as much as possible, at the start of the Seahorse assay there were equal number of cells in control and *Zbtb11* KO samples, and that these were evenly distributed across the surface of the well. Moreover, at the end of the assay we measured the number of cells in each well in order to correct the OCR data for minor differences that may have arisen during the plating and attachment stage.

In brief, after titrating the cell number, we determined that the optimal density for our assay was 100,000 ES cells per well. This many cells were plated on gelatinised Seahorse plates on the same day of the assay and allowed to attach for at least 5 hours in the presence of ROCK inhibitor – this approach avoids differences in growth rates during overnight cultures, and ensures more uniform monolayer attachment of the ES cells that otherwise aggregate into three dimensional colonies. To compare cell numbers after completion of the assay, at the end of the Seahorse run the cells were lysed and digested with proteinase K in the same plate to release the genomic DNA, which was then quantified by qPCR as a measure for cell number. Given the potential metabolic disadvantage of *Zbtb11* KO cells, in our case gDNA is a better control for cell number than protein measurement. We used the gDNA measurements to calculate a size factor for each well, which we then applied to the OCR data for normalisation – this normalisation approach applies only the minimal necessary transformation to the data, leaving it otherwise as close as possible to the original raw OCR measurements. As expected from the fact that we always aimed to plate the same number of cells, the size factors were always close to 1, and only minimal differences were observed.

To make sure that our technical approach was sound, during the optimisation stage we also observed the wells of the Seahorse plate under the microscope at the end of the run before lysing them for gDNA extraction. This confirmed that the cells were plated and remained attached at similar density throughout the assay in control and *Zbtb11* KO cells. We have included a representative example of images taken at the end of one experiment in Referee Fig. 2.

In conclusion, our oxygenic measurements were well controlled, which excludes the possibility that the observed differences were the result of measuring different cell numbers in control and *Zbtb11* KO samples. In addition, we argue that if there were significant differences in cell numbers, these would also be reflected in differences of non-mitochondrial respiration, but as can be seen in Fig. 5b this is not the case.

Referee Fig. 2 Representative bright field images of cell density in the Seahorse wells, taken at the end of the assay.

The BNGE in gel activity experiment shown is of very poor quality. the signal is very faint and the labelling of putative CI and supercomplexes incorrect (does not match with their own immunodetection). It is recommended to perform in gel activity and immunodetection of other complexes (IV, V, II) to properly understand and labelling of the different migration position of the complexes in the gel.

Answer 2.8 Please see above Answer 1.4. We have now replaced the in-gel activity assay with spectrophotometric measurements of complex I activity performed on mitochondrial extracts from control and *Zbtb11* KO cells (Fig. 6e in revised manuscript). This data confirms the loss of complex I activity following *Zbtb11* depletion.

Equally the experiments addressing the stability of the complexes need to be performed in BNGE and not in SDS-PAGE as it is down in Fig 7 since the authors themselves indicate that SDS is not informative of the assembly status.

Answer 2.9 To clarify, with respect to respiratory complexes assembly, our aim was to investigate the assembly of complex I, because it was evident *Zbtb11* directly controls transcription of complex I subunit genes (*Ndufc2*, *Ndufaf1*, *Ndufa12* and *Ndufs7*; see Fig. 6a), including assembly factors. We did this in Fig. 6e (original manuscript; Fig. 6d in revised manuscript). We have now added a line in the Results section to better explain why the

assembly of complex I was considered a likely hypothesis (lines 375-376 in revised manuscript).

By contrast *Zbtb11* does not control the transcription of any subunits or assembly factors of the complexes III and IV, so there is no specific hypothesis around the assembly of these complexes to be tested experimentally. Instead, several genes encoding mitochondrial ribosome subunits are down-regulated in *Zbtb11* KO, so the most likely hypothesis was that the down-regulation of complexes III and IV in *Zbtb11* KO cells was caused by an impairment in mitochondrial translation. The aim of our experiments in Fig. 7 was therefore to simply monitor the turnover dynamics of complexes I, III and IV, in order to determine whether they are consistent with a significant impairment of mitochondrial translation. We explain in more detail the rationale, scope and interpretation of the experiments in Fig. 7 in Answer 1.5 (please see above).

Reviewer #3 (Remarks to the Author):

The manuscript by Brooke C. Wilson and coauthors investigated molecular functions of ZBTB11, a protein implicated in hereditary intellectual disability. Authors performed extensive molecular analyses to conclude that ZBTB11 facilitates recruitment of Nuclear Respiratory Factor 2 (NRF2) to a subset of its target genes, thereby ensuring optimum expression of nuclear DNA encoded factors of mitochondrial electron transport chain complex I leading to optimum OXPHOS. Authors also concluded that ZBTB11 contributes to a mitoribosome defect, leading to an impairment in mitochondrial protein translation. Although the study is nicely designed and used various appropriate experimentation, there are several conclusions that need better justifications. The following suggestions should improve the manuscript.

1. It is surprising that authors chose a mouse embryonic stem cell line to monitor mitochondrial electron transport chain and oxidative phosphorylation. It is well known that pluripotent stem cells, including the E14 ES cells, utilizes dual mode (Glycolytic and oxidative) of energy production and largely rely on glycolytic energy metabolism. Thus, an experimental system with restricted mitochondrial oxidative capacity is not ideal to test electron transport chain function.

Answer 3.1. Energy metabolism is not actually the most important function of the electron transport chain (ETC), as ATP in many cell types - not only in ES cells - can be supplied in adequate amounts by glycolysis. An essential role of the ETC is to maintain the cellular pool of NAD^+ , which functions as an electron acceptor in several biosynthetic pathways^{5,6}. Complex I oxidises NADH to NAD^+ to regenerate the pool of available NAD^+ , so if the ETC is disrupted, NAD^+ becomes depleted, blocking the generation of several metabolites and, most critical for proliferation, aspartate synthesis^{5,6}. A functional oxidative phosphorylation (OXPHOS) system is therefore required by all proliferative cells⁵⁻⁷, including cancer cells⁸ despite having high rates of glycolysis.

While certain cell types with increased OXPHOS requirements have extreme rates of ETC activity, those are achieved by boosting mitochondrial biogenesis through cell type-specific

mechanisms. However, this is not the level of regulation that *Zbtb11* is implicated in, as neither itself nor its target genes are cell type-specific factors. Instead, *Zbtb11* is involved in the basic maintenance of the ETC homeostasis which is fundamental for all cells. The basic expression of ETC subunits is mainly driven by a subset of essential transcription factors, including NRF-1 and NRF-2/GABP, and our study now shows that *Zbtb11* is part of this essential regulatory axis, functionally associating with NRF-2/GABP to ensure specific subunits of complex I are expressed. This is a fundamental cellular role, and the new data we have now included in the manuscript strongly supports this, showing that *Zbtb11* targets the same genes in several cell types (Fig. 1g-h and Fig. 3c in revised manuscript), and that *Zbtb11* target genes are ubiquitously expressed (Supplementary Fig. 3c in revised manuscript). The evidence therefore indicates that *Zbtb11* is involved in the basic maintenance of the ETC, and not in regulating cell type-specific aspects of extreme ETC activity.

For these reasons, we considered several cell types would make a suitable experimental system to determine the fundamental cellular roles of *Zbtb11*. The essentiality of *Zbtb11* was a factor in choosing ES cells, as they allow the complex genetic manipulation required to generate an inducible KO line, without which it would have been impossible to obtain the same mechanistic insights. We explain this in a bit more detail above (please see Answer 2.1). Knocking out *Zbtb11* results in a reproducible and measurable mitochondrial phenotype in terms of complex I biogenesis, respiration and mitochondrial membrane potential, which shows that these cells are an adequate experimental system for investigating the fundamental mechanisms of ETC homeostasis.

2. With respect to the above point, it is surprising that defective ETC function is leading to such a drastic effect on ESC proliferation, rather than promoting differentiation to a primed ESC state or EpiSC/Differentiation state.

Answer 3.2. As mentioned in Answer 3.1 above, it is actually expected that a defective ETC leads to a block in proliferation, due to the essential role of the ETC in regenerating the pool of NAD⁺. Consistent with this, the additional data we included in the revised manuscript shows that provision of supra-physiological concentrations of pyruvate or aspartate partially rescues proliferation (Fig. 6f in revised manuscript).

It is possible that a differentiation phenotype would be observed if the ETC was moderately impaired, instead of severely blocked.

3. Nrf2 gene knockout mice are viable and Nrf1 knockout mice die in late gestation. These phenotypes do not support the conclusion that loss of ZBTB11-mediated facilitation of their transcriptional activity will induce a drastic effect on cell proliferation and induce cell death.

Answer 3.3. The Reviewer is confusing Nuclear Respiratory Factor 1 (NRF-1) with Nrf1 (Nfe2l1, Nuclear Factor, erythroid derived 2, like 1), and Nuclear Respiratory Factor 2 (NRF-2) with Nrf2 (Nfe2l2, Nuclear Factor, erythroid derived 2, like 2).

- NRF-1, the transcription factor our study is referring to, is essential for mitochondrial biogenesis in all cell types, and KO mice are peri-implantation lethal ⁹.
- Nrf1 (Nfe2l1), the transcription factor Reviewer #3 is referring to, has no relation to our study, and KO mice die in late gestation due to anemia ¹⁰

- NRF-2 is an essential dimeric transcription factor made of two subunits GABPa and GABPb (as also mentioned above by Reviewer #1), and we have made it clear throughout the manuscript the association of Zbtb11 is with GABPa - this is an essential factor and KO mice are certainly not viable. The early embryonic lethality of GABPa KO mice was shown using at least two different alleles^{11,12}.
- Nrf2 (Nfe2l2) on the other hand is not essential and mainly implicated in response to oxidative stress.

4. As authors are focusing on impairment in mitochondrial function, they should show by electron microscopy whether loss of ZBTB11 induces drastic effect on reduction in transverse cristae containing mitochondria or inducing mitophagy. Especially, a defect in mitoribosome could lead to mitophagy.

Answer 3.4. As explained above in Answer 1.5, our experiments do not support a significant impairment of mitochondrial translation. We have now inserted additional explanations in the manuscript and removed the reference to mitochondrial biogenesis from the subheading of the Results section.

Prevalent mitophagy in *Zbtb11* KO cells would be expected to lead to a blanket decrease in mitochondrial proteins relative to Lamin B, as well as a reduction in mitochondrial DNA (mtDNA) relative to nuclear DNA (gDNA). However, this is not apparent in our results. Although we observe down-regulation of specific complex I subunits, the levels of other mitochondrial proteins such as complex V subunit Atp5a, complex II subunit Sdhb (Fig. 6b), as well as complex I subunits Ndufa9 and Ndufa11 (Fig. 6d), do not change throughout the course of our experiments. Similarly, the ratio of mtDNA to gDNA is not significantly affected (Supplementary Fig. 4a). These results argue against pervasive mitophagy taking place in *Zbtb11* KO cells within the time window of our experiments.

5. The section and evidences supporting effect on mitoribosome is really weak (The heading of that section includes “mitoribosome biogenesis”, which is confusing). If there is a general defect in mitoribosome and mitochondrial transcription, all of the ETC complexes, rather than only complex I, will show drastic functional impairment in *Zbtb11*-KO cells. Authors should show more direct experimentation, like pulse chase in mitochondria or mito-ribo-Tag (people used strategy of tagging the mL62 of the large subunit) to make more definitive conclusions.

Answer 3.5. As explained above in Answer 1.5, our experiments in Fig. 7c do not support a severe impairment of mitochondrial translation. The most important aspect of the results in Fig. 7 is dissecting the contribution of individual pathways to the complex I biogenesis defect, confirming that the defect in complex I biogenesis is a consequence of the down-regulation of complex I genes, and not due to a general defect in mitochondrial translation.

While the results are consistent with a mild impairment of mitochondrial translation, as they partly mirror the mitochondrial translation block with chloramphenicol, we agree the effect is weak (which we state in the manuscript). As the experiments in Fig. 7a show, blocking

mitochondrial translation has different effects on individual ETC complexes, due to their different turnover rates. Consequently, the effects on complex V would become obvious much later than complex I for example, because the former appears to be a lot more stable, especially if the impairment is mild. Our experimental system is not ideal for investigating this type of mild chronic deficiencies that take a long time to take effect because all cells die 3 days after Zbtb11 protein becomes depleted.

We have now inserted additional explanations in the manuscript and in order to avoid suggesting that our results show Zbtb11 controls complex I biogenesis and mitochondrial translation to an equivalent degree, we have also removed the reference to mitochondrial translation from the subheading of this results section, which is now entitled “Zbtb11 controls complex I biogenesis”.

6. The data with mutant ZBTB11 is also inconclusive. The extent of defect (less rescue) in mRNA and protein expression are not convincing (Fig. 8 a,b, c). It is more concerning due to the fact that mutant protein levels are almost 50% to the wild type protein. The reduced stoichiometry by itself can cause the reduced transcription (As it is known that transcription efficiency by a transcription factor often depends on its expression level in a cell).

Answer 3.6. This is exactly our point. The fact that mutant protein levels are reduced by almost 50% of the wild type protein is unequivocal, and it is an important result, because it indicates the mutations destabilise the Zbtb11 protein, so consequently there is less of it around to activate transcription. This indicates a mechanism by which human *ZBTB11* mutations may lead to gene deregulation and disease. We explain this in the relevant results section (final paragraph on page 15 in original manuscript; or lines 486-498 in revised manuscript) and discussion (final paragraph on page 18 in original manuscript; or lines 591-598 in revised manuscript).

As explained in the manuscript (second to last paragraph on page 15 in original manuscript; or lines 477-479 in revised manuscript), our experiments were purposefully designed to achieve similar levels of wild type and mutant Zbtb11 transcription, by expressing them fused to an internal ribosome entry site (IRES) and GFP, and subsequently FACS-sorting cells with the same level of GFP expression. qRT-PCR showed this approach was successful, as wild type and mutant Zbtb11 transcripts were indeed expressed at the same level (as shown in Fig. 8a, left panel). The fact that the protein levels were different was unexpected to us, but this result was consistent in showing mutant protein levels were approximately 50% of wild type (Fig. 8b, left and right panels). Because the transcript levels are the same, the logical explanation is that the mutations destabilise the Zbtb11 protein. We do not know the exact mechanism by which this instability is introduced, but it may well be that the predicted misfolding of the zinc finger motifs results in a faster rate of degradation. Interestingly, both mutations affect the second histidine amino acid in different C2H2 zinc finger motifs, and a recent study found that this position is significantly over-represented among somatic mutations in cancer¹³, suggesting that disrupting zinc finger motifs in this way may in many instances lead to increased protein degradation.

These experiments provide an important insight into how the pathogenic mutations affect Zbtb11 activity, namely by destabilising the protein and reducing its dosage. This has an impact on the expression of some of its target genes involved in complex I biogenesis, as Ndufc2 and Ndufaf1 were consistently expressed at lower levels in cells rescued with mutant Zbtb11, being down-regulated two-fold in at least one of the mutant samples, and these changes were sufficient to have an impact on the amount of complex I holocomplexes synthesised in the cell (Fig. 8b, left and middle panels).

It is clear not all Zbtb11-dependent genes are affected to the same extent by the mutation of Zbtb11, as some of them were rescued with similar efficiency by wild type and mutant Zbtb11. This was particularly the case for the mitoribosome genes, which suggests that mitoribosome deficiency is less likely to be a characteristic of ZBTB11-associated intellectual disability. This is likely to reflect the difference dependency of individual promoters on Zbtb11 dosage, as the genes that are most down-regulated in the KO (*Ndufc2* and *Ndufaf1*) are also the most down-regulated genes in the mutant rescue samples as well.

7. Also, the experimental approach with mutants seems indirect. A more conclusive approach is to generate mutations at the endogenous locus using CRISPR/Cas9 (similar to the approach of generating a floxed locus).

Answer 3.7. Both approaches have their strengths and weaknesses, and we see no objective reason to consider ours inconclusive. Generating individual cell lines by genome editing is not a perfect approach either, as it involves single cell clonal expansion which may generate a cell line that is very different from the parental one, accumulating chromosome abnormalities or other undetected off-target editing effects that confer a proliferative advantage and which are selected for during the clonal expansion. By contrast, by using our approach we are certain the wild type and mutant Zbtb11 are compared against the same genetic background.

There is also no reason to consider our experiments indirect. They deplete cells of endogenous Zbtb11 and replace it with mutant Zbtb11 in a controlled approach as described above (Answer 3.6), and then measure transcription of genes controlled by Zbtb11. They are therefore a direct interrogation of the effects of the pathogenic mutations on the activity of Zbtb11. They do not measure indirect effects.

8. Earlier studies using HEH293 cell indicated that ZBTB11 is mostly localized within the nucleoli of cell and regulates ribosomal RNA expression. Does ZBTB11 protein has a similar localization preference in ESCs? If so, what is the effect on rRNA biosynthesis and translation of nuclear DNA encoded genes? If not? Does ZBTB11 show cell-type specific localization pattern? In that situation authors should show relevance of this study/conclusion in a human cell type, preferably neuronal cells.

Answer 3.8. In the study cited here, Fattahi et al ⁴ never actually show that Zbtb11 regulates rRNA expression. They show that overexpressed GFP-Zbtb11 localises to the nucleoli of HEK293 cells, but strikingly, in the same figure panel they also show that immunofluorescence of the endogenous protein does not actually detect any localisation to the nucleoli. The same

antibody to human Zbtb11 used in this study was also used by the Human Protein Atlas project, in parallel with another independent antibody (a total of two independent Zbtb11 antibodies) to stain endogenous Zbtb11 in three different human cell lines, and all of the images show nuclear staining that is clearly negative for the nucleoli.

<https://www.proteinatlas.org/ENSG00000066422-ZBTB11/antibody#ICC>

Strikingly, this inconsistency is never addressed in the paper, despite the fact that their own images of endogenous Zbtb11 as well as those of the Human Protein Atlas (where they source the antibody from) do not agree with the nucleolar localisation of overexpressed GFP-Zbtb11. The logical conclusion is however, that Zbtb11 does not localise to the nucleoli under physiological conditions.

Importantly, the study includes no functional experiments to show that perturbing the expression of Zbtb11 or mutating it actually leads to abnormal rRNA expression, so there is no result that we could compare an experiment in ES cells to.

We have already addressed the issue of the relevance of our study to human cells several times above (please see Answers 1.6, 2.1, 2.4 and 3.1), namely that Zbtb11 controls fundamental processes that are essential for all cell types and are highly conserved. This is supported both by the new data (Fig. 1g-h and Fig. 3c in revised manuscript), as well as by data in the original manuscript (Fig. 8c). The data in Fig. 8c is particularly relevant for the role of Zbtb11 in humans, as it shows strong correlation of expression in human primary tissues between Zbtb11 and the targets we identify in mouse ES cells. The analysis is particularly powerful because it is based on the entire GTEx dataset, which contains whole transcriptome data from 948 different individuals, and comes from 52 different primary human tissues (instead of immortalized cell lines). It shows that *ZBTB11* and the 154 targets we identified in mouse ES cells, while ubiquitously expressed, they also have slight differences of expression in different tissues, and these differences are highly correlated. The chances of this correlation to happen by chance are infinitesimally small, and this type of correlation is considered highly indicative of a regulatory relationship, and is commonly used to establish regulatory networks in systems biology approaches such as weighted gene co-expression network analysis³. The high level of correlation between Zbtb11 and its targets therefore strongly indicates that the regulatory relationship between Zbtb11 and the targets we identified in mouse ES cells, is conserved in humans.

9. The impaired mitochondrial ETC function often leads to prominent effect in response to oxidative stress (probably that aspect is more relevant to the hereditary intellectual disability with mutant ZBTB11). Authors should consider to test this aspect.

Answer 3.9 We thank the reviewer for this suggestion. It is an aspect we are interested in, but it is currently work in progress.

Reviewer #4 (Remarks to the Author):

Wilson et al identified a novel function of Zbtb11 as a transcription factor activating a series of nuclear-encoded mitochondrial genes with another activator NRF-2. Technically, the study employed an elegant mix of CRISPR, RNA-seq, ChIP-seq, bioinformatics and mitochondrial-related experiments. The author walked the reader through the genomic approaches whereby they utilized the elegant RosaCreERT2 loxP system to deplete ZBTB11 in mESCs and identified its direct targets as mitochondrial genes. They further observed that ZBTB11 can cooperate with a basic leucine zipper (bZIP) transcription factor NRF2, which is one of the most prominent regulators of nuclear-encoded mitochondrial genes. Their data also suggested that the ZBTB11-NRF2 interaction may be essential for mitochondrial function by maintaining the biogenesis of respiratory complex I. Overall, this is a solid comprehensive work that reveals a mechanism underlying the transcriptional regulation of nuclear-encoded mitochondrial genes by ZBTB11. However, some suggestions need to be considered:

By re-analyzing the high-throughput raw data from the authors, I found most of the sequencing and analysis in the study were well performed. The RNA-seq, ChIP-seq data was consistent between different samples. ChIPseq motif calling showed ZBTB11 and NRF2 have similar binding motifs (CCGGAAG). However, this makes me confused a little bit. Because ZBTB11 is a well-conserved protein with 12 canonical C2H2-ZF motifs, which is supposed to contact 36bp DNA theoretically. There is a big gap between experimental data and theoretical data. This requires further experimental verification, such as in vitro luciferase assay or electrophoretic mobility shift assay. This also led me to speculate whether there is a possibility that ZBTB11 cannot directly bind to DNA but instead binds to DNA through NRF2. If so, this speculation is contradictory to the reChIP experiments in the manuscript.

Answer 4.1 It is not necessary that all zinc finger motifs in the protein are concomitantly engaged in the specific recognition of the DNA motif. There are several examples in the literature of transcription factors that only use subsets of their zinc finger motifs to determine the DNA sequence specificity. Recent structural studies found that although CTCF has 11 zinc fingers, only 5 of them participate in sequence-specific interactions with the DNA, while other fingers were found to engage the DNA to stabilise the interaction but in a sequence-nonspecific manner¹⁴. Another example is Zfp335, which has 13 zinc finger motifs, but these organise in two different clusters that recognise distinct consensus sequences¹⁵. It is therefore possible that Zbtb11 uses only a subset of its zinc fingers to specifically engage the CCGGAAG motif, while the other zinc fingers may contribute to the stability of the interaction without any particular sequence requirements.

To assess whether Zbtb11 has the ability to engage the CCGGAAG motif directly, we predicted the sequence expected to be recognised by the Zbtb11 zinc finger motifs, using a previously published model¹⁶ (<http://zf.princeton.edu/index.php>). The resulting 36bp sequence includes at least two different regions of high similarity to the GABPa motif (MA0062.1, JASPAR 2020¹⁷) (see Referee Fig. 3), indicating that Zbtb11 contains within its structure at least two different zinc finger clusters that have the ability to interact directly with the motif sequence.

We believe our experimental results do not support the alternative hypothesis - that Zbtb11 binds chromatin via interactions with NRF-2 - because Zbtb11 depletion results in loss of NRF-2 recruitment (Fig. 4g). If Zbtb11 was recruited to promoters via interactions with NRF-2, the recruitment of NRF-2 would be expected to be independent of Zbtb11 since it would be an

upstream event, but this is not the case. There is still however the possibility that Zbtb11 and NRF-2 bind cooperatively.

Irrespective of the mechanism by which Zbtb11 and NRF-2 associate on the chromatin, we would still expect the same ChIP-reChIP results (Fig. 4f) because both factors would be bound to the same promoters, on the same allele. Because the chromatin is cross-linked, this technique does not differentiate between factors that engage the DNA directly or not. The covalent bonds formed by formaldehyde will make sure that all factors that co-localise with Zbtb11 on the same piece of DNA will be reChIP-ed, whether they form a protein complex, or whether they are bound directly to DNA.

The mutations H729Y and H880Q are located at the fifth and the tenth C2H2 zinc finger motifs, respectively. The authors might need to construct those mutant mESCs to examine if the mutate ZBTB11 binding sites have changed or not. This might help to verify if ZBTB11 interacts with DNA directly. In general, I suggest the authors emphasize on the ZBTB11 and DNA interaction part, which should be clarified.

Answer 4.2 Our experiments suggest that the main mechanism by which the pathogenic mutations affect the activity of Zbtb11 is by destabilising the protein and consequently reducing its dosage. Our experiments indicate the mutations lead to close to 50% protein reduction in both mutants. The destabilisation of the protein provides an explanation for how mutations in two different zinc finger motifs, which are not in close vicinity in the protein sequence, can lead to the same phenotype. By reducing the dosage of Zbtb11 to the same extent, both mutations will lead to the same gene expression changes, and therefore the same phenotype.

By contrast, if the mutations had a significant effect on the binding specificity to DNA, the two mutations would be expected to lead to more locus-specific effects, because they affect different determinants of DNA specificity (different zinc fingers), and because the sequence context (how similar it is to the consensus sequence) would play a more critical role. We would therefore expect the two mutations to lead to different transcriptional changes, and therefore likely to different phenotypes.

Our rescue experiments using the two Zbtb11 mutants (Fig. 8a) show that within the set of Zbtb11-dependent complex I and mitochondriome genes, the mutations have very little locus-specific effects. The degree of rescue when expressing the mutants in Zbtb11 KO cells appears

to be uniform across this set of targets, transcription being well correlated between the KO sample and the mutant rescue samples (Referee Fig. 4) – i.e., genes that are most deregulated in the KO remain most deregulated when rescued with mutants, and vice versa. This suggests that dosage rather than sequence recognition underpins the transcriptional activation defect at these genes.

Referee Fig. 4 Correlation of transcriptional changes (effect size relative to control) in *Zbtb11* KO cells vs cells rescued with *Zbtb11* mutants.

If ZBTB11 does interact with 8bp DNA and the binding is necessary for the recruitment of NRF2, then their interaction should be easily detected by using immunoprecipitation assay. Furthermore, more ZBTB11 interacting proteins will be identified by mass spectrometry, this can be done by using Flag-tagged ZBTB11 mESCs already used in the manuscript.

Answer 4.3 We thank the reviewer for this suggestion. We have actually attempted these experiments, but did not obtain reproducible results. The main problem seems to be obtaining *Zbtb11* protein complexes in native conditions for co-IP, so they require further optimisations.

It seems that ZBTB11 and NRF2 can activate mitochondrial related genes by directly targeting their promoters. If so, we probably can see some changes in histone modifications and/or chromatin accessibility in ZBTB11 KO cells. Please clarify this point and identify the chromatin related factors involved in this event if possible.

Answer 4.4 This is an additional mechanistic layer that we are interested in, but it is currently work in progress.

Minors:

Page 8 “To this end we performed de novo motif discovery using the *Zbtb11* ChIP-seq peakset, and separately the promoter regions of the 154 *Zbtb11*-dependent genes. Both approaches identified as the top hit the same motif, which closely matched that recognized by the ETS-domain protein GABPa (Fig. 4a)” It is better to show two separate motifs for ZBTB11 ChIP-seq and promoter regions.

FigureS7a, I recommend labeling wt control (red dots) with C1/C2/C3, rather than R1/R2/R3, because they were not ZBTB11 floxed mESCs.

We thank the reviewer for pointing these out. We are now showing in the revised manuscript the separate motifs identified using the ChIP-seq dataset and the promoters in Fig. 4a and 4b, respectively.

Labels in Fig. S7a (now Fig. S8a in revised manuscript) have also been changed.

References

1. Blomen, V. A. *et al.* Gene essentiality and synthetic lethality in haploid human cells. *Science* **350**, 1092–1096 (2015).
2. Wang, T. *et al.* Identification and characterization of essential genes in the human genome. *Science* **350**, 1096–1101 (2015).
3. Langfelder, P. & Horvath, S. WGCNA: an R package for weighted correlation network analysis. *BMC Bioinformatics* **9**, 559 (2008).
4. Fattahi, Z. *et al.* Biallelic missense variants in ZBTB11 can cause intellectual disability in humans. *Hum Mol Genet* (2018) doi:10.1093/hmg/ddy220.
5. Sullivan, L. B. *et al.* Supporting Aspartate Biosynthesis Is an Essential Function of Respiration in Proliferating Cells. *Cell* **162**, 552–563 (2015).
6. Birsoy, K. *et al.* An Essential Role of the Mitochondrial Electron Transport Chain in Cell Proliferation Is to Enable Aspartate Synthesis. *Cell* **162**, 540–551 (2015).
7. Yao, C.-H. *et al.* Mitochondrial fusion supports increased oxidative phosphorylation during cell proliferation. *eLife* **8**, e41351 (2019).
8. Garcia-Bermudez, J. *et al.* Aspartate is a limiting metabolite for cancer cell proliferation under hypoxia and in tumours. *Nat. Cell Biol.* **20**, 775–781 (2018).
9. Huo, L. & Scarpulla, R. C. Mitochondrial DNA instability and peri-implantation lethality associated with targeted disruption of nuclear respiratory factor 1 in mice. *Mol. Cell. Biol.* **21**, 644–654 (2001).
10. Chan, J. Y. *et al.* Targeted disruption of the ubiquitous CNC-bZIP transcription factor, Nrf-1, results in anemia and embryonic lethality in mice. *The EMBO Journal* **17**, 1779–1787 (1998).
11. Ristevski, S. *et al.* The ETS Transcription Factor GABP α Is Essential for Early Embryogenesis. *Molecular and Cellular Biology* **24**, 5844–5849 (2004).
12. Jaworski, A., Smith, C. L. & Burden, S. J. GA-Binding Protein Is Dispensable for Neuromuscular Synapse Formation and Synapse-Specific Gene Expression. *Molecular and Cellular Biology* **27**, 5040–5046 (2007).
13. Munro, D., Ghersi, D. & Singh, M. Two critical positions in zinc finger domains are heavily mutated in three human cancer types. *PLOS Computational Biology* **14**, e1006290 (2018).
14. Hashimoto, H. *et al.* Structural Basis for the Versatile and Methylation-Dependent Binding of CTCF to DNA. *Molecular Cell* **66**, 711–720.e3 (2017).
15. Han, B. Y., Foo, C.-S., Wu, S. & Cyster, J. G. The C2H2-ZF transcription factor Zfp335 recognizes two consensus motifs using separate zinc finger arrays. *Genes Dev.* **30**, 1509–1514 (2016).
16. Persikov, A. V. & Singh, M. De novo prediction of DNA-binding specificities for Cys2His2 zinc finger proteins. *Nucleic Acids Res* **42**, 97–108 (2014).
17. Fornes, O. *et al.* JASPAR 2020: update of the open-access database of transcription factor binding profiles. *Nucleic Acids Res.* **48**, D87–D92 (2020).

REVIEWER COMMENTS

Reviewer #1 (Remarks to the Author):

The Authors have convincingly responded to most if not all the comments of the reviewers. This is a relevant, solid and convincing paper on the contribution of a new regulatory mechanism in the biogenesis of complex I and address the still poorly understood field of the nuclear gene control of different aspects of mitochondrial formation and bioenergetics regulation. I think the new version of the paper gives an important contribution to this interesting and growing field.

Reviewer #2 (Remarks to the Author):

The authors have answered experimentally to my major concerns and have also eliminated those results that were experimentally weak. I have no further major issues with the manuscript.

Reviewer #3 (Remarks to the Author):

The revised manuscript and the rebuttal letter address several concerns of this reviewer. The new ChIP-seq data in thymocytes and HEK293 is supportive of a generalized role of the Zbtb11 in different cell types. However, there are still major concerns about the manuscript. Major concerns are:

1. Major experimental analyses are done with curated data with target genes, where Zbtb11 ChIP-seq signal is high. The high ChIP-seq signal does not equate to enhanced function as the ChIP procedure only captures one of the dynamic states. For unbiased analyses, authors should perform parallel analyses with all Zbtb11 target genes and find out whether genes for mitochondrial functions are overrepresented. Also, what is the expression pattern of target genes that are not showing high Zbtb11 ChIP signal in Zbtb11-depleted and Zbtb11-mutant cells?
2. The argument for using mouse ESCs as a model system is still not convincing. It is hard to believe that ESCs are showing the drastic phenotypic effect only due to the lack of maintenance of mitochondrial membrane potential based on the fact that their energy requirement is largely reliant on glycolysis. Authors should test with a chemical reagent that specifically affect MMP and test whether that phenocopies Zbtb11 deletion. Also, they could perform the loss of function analyses in other cell types (hepatocyte and HEK293), which are distinct from ESCs about mitochondrial copy number, energy metabolism etc.
3. One of the major phenotypes in Zbtb11-depleted cells is arrested cell cycle. What is the effect on the expression of cell cycle regulators? How does the manuscript rule out the possibility that the defective cell cycle is contributing to the phenotype in Zbtb11-mutant cells?
4. One of the major concerns of this study is lack of any in-vivo experimentation. Most of the studies are done in cell culture model, including the mutation studies. Thus, it is hard to understand whether the generated data truly represents what is happening in-vivo. This concern reduces the significance of the study and any supporting in-vivo data will enhance the impact of the study.

Reviewer #4 (Remarks to the Author):

I have re-read the manuscript and author response to the second round of reviews. This study has been much improved through the review process. I am satisfied that the authors have fully addressed all of my concerns.

We would like to express our gratitude to the reviewers for their time and contributions to improving the manuscript. We hope the below, answers the remaining questions of Reviewer #3.

Reviewer #3 (Remarks to the Author):

The revised manuscript and the rebuttal letter address several concerns of this reviewer. The new ChIP-seq data in thymocytes and HEK293 is supportive of a generalized role of the Zbtb11 in different cells types. However, there are still major concerns about the manuscript. Major concerns are:

1. Major experimental analyses are done with curated data with target genes, where Zbtb11 ChIP-seq signal is high. The high ChIP-seq signal does not equate to enhanced function as the ChIP procedure only captures one of the dynamic states. For unbiased analyses, authors should perform parallel analyses with all Zbtb11 target genes and find out whether genes for mitochondrial functions are overrepresented. Also, what is the expression pattern of target genes that are not showing high Zbtb11 ChIP signal in Zbtb11-depleted and Zbtb11-mutant cells?

To clarify, it is the transcriptomics data that tells us the functional Zbtb11 binding sites are demarcated by Zbtb11-high peaks.

The 154 differentially expressed genes identified by RNA-seq in Zbtb11 KO cells, and which we used for downstream enrichment analyses are the only deregulated genes, and there are no other differentially expressed genes that were omitted from the analysis based on the binding of Zbtb11 or any other criteria. The transcriptomics analysis was performed completely independently of the ChIP-seq data, and it was not curated in any way. It was only after performing the differential gene expression analysis that we then asked how strong is the Zbtb11 ChIP-seq signal at the differentially expressed genes, and this showed that in their vast majority (95%) the differentially expressed genes have Zbtb11-high peaks at their promoters. The ChIP-seq data had therefore no bearing on the RNA-seq results, but the other way around – the transcriptomics results are telling us that the Zbtb11-high peaks are the functionally relevant ones.

It is therefore only genes with Zbtb11-high peaks at their promoters that change expression in the KO cells, while genes without these strong peaks are not deregulated. So, while for some transcription factors ChIP-seq signal does not always equate with function, in the case of Zbtb11 these measures are well correlated.

In the end, analysing the genomic binding sites of Zbtb11 has to be carried out in the context of the RNA-seq data. It is the genes that change expression when Zbtb11 is depleted that we should be focusing on, because they are the only ones that are demonstrably dependent on Zbtb11, and these are highly enriched in genes with mitochondrial functions. The other genes that do not change expression in *Zbtb11* KO cells despite having Zbtb11 binding sites may reflect redundancy with other transcription factors or non-functional Zbtb11 binding events.

2. The argument for using mouse ESCs as a model system is still not convincing. It is hard to believe that ESCs are showing the drastic phenotypic effect only due to the lack of

maintenance of mitochondrial membrane potential based on the fact that their energy requirement is largely reliant on glycolysis. Authors should test with a chemical reagent that specifically affect MMP and test whether that phenocopies *Zbtb11* deletion. Also, they could perform the loss of function analyses in other cell types (hepatocyte and HEK293), which are distinct from ESCs about mitochondrial copy number, energy metabolism etc.

As explained in the revised manuscript (lines 377-388), and also pointed out by Reviewer #2 in the previous round of review, we do not expect the proliferation arrest is related to energy requirements, but to aspartate restriction.

Two seminal studies published in 2015 (Birsoy *et al.*, *Cell* 2015; Sullivan *et al.*, *Cell* 2015) have shown that the activity of the electron transport chain (ETC) is essential for proliferation, even in cells with high glycolysis rates. Although glycolysis is sufficient to support the ATP requirements, the activity of the ETC (specifically the activity of respiratory complex I) is nevertheless required in order to oxidise NADH to NAD⁺, because NAD⁺ functions as an electron acceptor in several biosynthetic pathways. Disruption of the ETC activity leads to the depletion of NAD⁺, blocking the generation of several metabolites and - most critical for proliferation, aspartate synthesis in the mitochondrial matrix (see Referee Fig. 1). This was shown to be a universal mechanism (Birsoy *et al.*, *Cell* 2015; Sullivan *et al.*, *Cell* 2015), including in cancer cells which are considered the quintessential glycolysis-dependent cells (Garcia-Bermudez, J. *et al.*, *Nat. Cell Biol.*, 2018).

It is therefore expected that loss of complex I activity in *Zbtb11* KO cells does have a profound effect on cell proliferation due to aspartate deficiency. Consistent with this, the data included in the last manuscript revision (Fig. 6f) shows that providing supra-physiological concentrations of pyruvate or aspartate rescues proliferation of *Zbtb11* KO cells.

The primary mitochondrial defect in *Zbtb11* KO cells is the loss of complex I activity, and while this leads to mitochondrial depolarisation, the mitochondrial membrane potential (MMP) is not itself driving aspartate synthesis as it does ATP synthesis (Referee Fig. 1). Collapsing the MMP through pharmacological approaches (e.g. with FCCP) in wild type cells would in fact have the opposite effect on aspartate synthesis, as mitochondrial depolarisation stimulates the activity of respiratory complexes and therefore increases the activity of complex I and consequently the rate of NADH to NAD⁺ conversion. This approach would therefore not phenocopy *Zbtb11* KO cells (which suffer from a loss of complex I activity and decreased NADH oxidation rate). The phenotype of *Zbtb11* KO cells would, however, be expected to be reproduced by treating ESCs with the complex I-specific poison rotenone. We tested this experimentally and as can be seen in Referee Fig. 2, even small concentrations of rotenone have a strong inhibitory effect on the proliferation and survival of ESCs, and this effect can be prevented by supplementing the growth media with additional aspartate. This shows the activity of complex I is critical for the proliferation and survival of ESCs.

3. One of the major phenotypes in *Zbtb11*-depleted cells is arrested cell cycle. What is the effect on the expression of cell cycle regulators? How does the manuscript rule out the possibility that the defective cell cycle is contributing to the phenotype in *Zbtb11*-mutant cells?

The cell cycle arrest is a late event following induction of *Zbtb11* KO and *Zbtb11* protein depletion. While the transcriptional changes induced by *Zbtb11* depletion are detected 48 hours post-KO induction, the cell cycle arrest does not manifest until a further 48 hours later, i.e. 96 hours post-KO induction (Fig. 2c). This means that even after the *Zbtb11*-dependent genes become deregulated, *Zbtb11* KO cells still go through at least two more cell cycles before they arrest (likely more than two, given that ESCs are rapidly cycling cells). This is incompatible with *Zbtb11* depletion having a direct effect on cell cycle regulators, especially considering that proteins like cyclins need to be synthesised anew every cell cycle.

In the sequence of events following *Zbtb11* depletion, the mitochondrial phenotype precedes the cell cycle arrest. The activity of complex I, mitochondrial respiration and the mitochondrial membrane potential (MMP) are all already affected 48 hours post-KO induction, and these deteriorate further 72 hours post-KO induction (Fig. 5 and Fig. 6e). By the time cell cycle arrest becomes evident, 96 hours post-KO induction, complex I activity and the MMP are severely impaired (Fig. 5a and Fig. 6e). This sequence of events excludes the possibility that cell cycle arrest is an immediate consequence of *Zbtb11* depletion, and instead indicate it is a consequence of losing complex I activity. The G1 arrest is consistent with a metabolic restriction, to which a significant contribution is brought by aspartate depletion due to impaired biosynthesis of this amino acid as a result of NAD⁺ depletion, which in turn is caused by the impaired complex I activity. Consistent

with this is the fact that at the timepoint the cell cycle arrest is induced (96 hours post-KO), we can rescue the proliferation of *Zbtb11* KO cells by providing supraphysiological concentrations of aspartate (Fig. 6f).

We have now included a paragraph in the Discussion (lines 536-547 in newly revised manuscript) to emphasise the causal link between loss of complex I activity, aspartate restriction and cell cycle arrest. We thank the Reviewer for raising this point.

4. One of the major concerns of this study is lack of any in-vivo experimentation. Most of the studies are done in cell culture model, including the mutation studies. Thus, it is hard to understand whether the generated data truly represents what is happening in-vivo. This concern reduces the significance of the study and any supporting in-vivo data will enhance the impact of the study.

Our study is primarily focused on determining the cellular and molecular functions of *Zbtb11*, as these are currently unknown, and we do not claim to provide a comprehensive mechanism for the aetiology of *ZBTB11*-associated intellectual disability. However, to understand what processes are disrupted when *Zbtb11* is mutated, it is important to determine what the cellular functions of this factor are, and we believe our study is an important contribution towards this goal. In addition, we provide evidence that the human pathogenic mutations destabilise the protein, reducing its dosage and therefore revealing a molecular mechanism directly relevant for understanding the disease aetiology. We are aware these findings will eventually have to be evaluated in the context of an *in vivo* disease model; nevertheless, we believe our experiments provide some valuable insights at this stage.

To avoid confusion over the scope of our study, we made some changes to the manuscript where the *ZBTB11*-associated human disease is referenced. We hope this will clarify that we are not making categorical claims with regard to the aetiology of the disease, and that this should be further investigated.

Abstract (lines 24-25): replaced “provides a rationale for the aetiology...” with “may help understand the aetiology...”

Discussion (lines 556-557): replaced “Our findings provide a rationale for...” with “Our findings may help understand the aetiology...”

Discussion (lines 565-566): replaced “It is therefore likely that...” with “We therefore propose that...”

Discussion (lines 567-568): added the sentence “Further studies *in vivo* are required to determine the precise aetiology of this syndrome.”

References

Birsoy, Kıvanç, Tim Wang, Walter W. Chen, Elizaveta Freinkman, Monther Abu-Remaileh, and David M. Sabatini. 'An Essential Role of the Mitochondrial Electron Transport Chain in Cell Proliferation Is to Enable Aspartate Synthesis'. *Cell* 162, no. 3 (30 July 2015): 540–51. <https://doi.org/10.1016/j.cell.2015.07.016>.

Garcia-Bermudez, Javier, Lou Baudrier, Konnor La, Xiphias Ge Zhu, Justine Fidelin, Vladislav O. Sviderskiy, Thales Papagiannakopoulos, et al. 'Aspartate Is a Limiting Metabolite for Cancer Cell Proliferation under Hypoxia and in Tumours'. *Nature Cell Biology* 20, no. 7 (2018): 775–81. <https://doi.org/10.1038/s41556-018-0118-z>.

Sullivan, Lucas B., Dan Y. Gui, Aaron M. Hosios, Lauren N. Bush, Elizaveta Freinkman, and Matthew G. Vander Heiden. 'Supporting Aspartate Biosynthesis Is an Essential Function of Respiration in Proliferating Cells'. *Cell* 162, no. 3 (July 2015): 552–63. <https://doi.org/10.1016/j.cell.2015.07.017>.

REVIEWERS' COMMENTS:

Reviewer #3 (Remarks to the Author):

I am satisfied with the clarification from authors and associated modifications in the revised manuscript. I support acceptance of the manuscript for publication.